**Title page**

FES2014 global ocean tide atlas: design and performance

Florent H. Lyard, LEGOS/CNRS, Toulouse, France / florent.lyard@legos.obs-mip.fr

Damien J. Allain, LEGOS/CNES, Toulouse, France / damien.allain@legos.obs-mip.fr

Mathilde Cancet, Noveltis, Toulouse, France / mathilde.cancet@noveltis.fr

Loren Carrère, CLS, Toulouse, France / lcarrere@groupcls.com

Nicolas Picot, CNES, Toulouse, France / nicolas.picot@cnes.fr

florent.lyard@legos.obs-mip.fr

https://orcid.org/0000-0001-6044-2642

**Abstract**

Since the mid-1990's, a series of FES (Finite Element Solution) global ocean tidal atlases have been produced and released with the primary objective to provide altimetry missions with tidal de-aliasing correction at the best possible accuracy. We describe the underlying hydrodynamic and data assimilation design and accuracy assessments for the latest FES2014 release (finalized in early 2016), especially for the altimetry de-aliasing purposes. The FES2014 atlas shows extremely significant improvements compared to the standard FES2004 and 20 (intermediary) FES2012 atlases, in all ocean compartments, especially in shelf and coastal seas, thanks to the unstructured grid flexible resolution, recent progress in the (prior to assimilation) hydrodynamic tidal solutions, and use of ensemble data assimilation technique. Compared to earlier releases, the available tidal constituent's spectrum has been significantly extended, the overall resolution augmented, and additional scientific by-products such as loading and self-attraction, energy diagnostics or lowest astronomical tides have been derived from the 25 atlas and are available. Compared to the other available global ocean tidal atlases, FES2014 clearly shows improved de-aliasing performances in most of the global ocean areas and has consequently been integrated in satellite altimetry Geophysical Data Records (GDRs) and gravimetric data processing, and adopted in recently renewed ITRF standards (International Terrestrial Reference System, 2020). It also provides very accurate open boundary tidal conditions for regional and coastal modelling.

**Keywords**

Global tides; unstructured modelling; data assimilation; satellite altimetry; de-aliasing

**Declarations**

All manuscripts must contain the following sections under the heading 'Declarations'.

If any of the sections are not relevant to your manuscript, please include the heading and write 35 'Not applicable' for that section.

*To be used for non-life science journals*

**Funding**:

FES2014 project has been supported by CNRS and the French space agency (CNES) under grants n°131678, n°160306 and grants n°161431 (general grant framework no14026) and through the OSTST/TOSCA (INSU) scientific fundings.

**Conflicts of interest/Competing interests** (include appropriate disclosures):

none

**Availability of data and material:**

public access through:

AVISO+ CNES/NASA website (https://www.aviso.altimetry.fr/en/data/products/auxiliaryproducts/

global-tide-fes.html)

CTOH products website (https://doi.org/10.6096/CTOH_FES2014_2021)

**Code availability:**

The hydrodynamic code T-UGOm (CECILL licence) is available at https://hg.legos.obs-mip.fr/tugo Mercurial repository

The tidal prediction code is distributed by AVISO+ and it is available on: https://bitbucket.org/cnes_aviso/fes/src/master

**Authors' contributions** (optional: please review the submission guidelines from the journal whether statements are mandatory)

Please see the relevant sections in the submission guidelines for further information as well as various examples of wording. Please revise/customize the sample statements according to your own needs.

# FES2014 global ocean tides atlas: design and performance

Florent H. Lyard[1], Damien J. Allain[1], Mathilde Cancet[2], Loren Carrère[3], Nicolas Picot[4]

[1]LEGOS, Observatoire Midi-Pyrénées, Toulouse, France
[2]Noveltis, Labège, France
[3]CLS, Ramonville, France
[4]CNES, Toulouse, France

*Correspondence to:* F. Lyard (florent.lyard@legos.obs-mip.fr)

**Abstract.** Since the mid-1990's, a series of Finite Element Solution (FES) global ocean tidal atlases has been produced and released with the primary objective to provide altimetry missions with tidal de-aliasing correction at the best possible accuracy. We describe the underlying hydrodynamic and data assimilation designs for the latest FES2014 release (finalized in early 2016), especially for the altimetry de-aliasing purposes. The FES2014 atlas shows extremely significant improvements compared to the FES2004 and (intermediary) FES2012 atlases, in all ocean regions, especially in shelf and coastal seas; these advances are due to the unstructured grid flexible resolution, recent progress in the (prior to assimilation) hydrodynamic tidal solutions, and to the use of an ensemble data assimilation technique. Compared to earlier releases, the FES2014 available tidal constituent's spectrum has been significantly extended, the overall resolution augmented, and additional scientific by-products such as loading and self-attraction, energy diagnostics or lowest astronomical tides have been derived from the atlas and are available. Compared to the other available global ocean tidal atlases, FES2014 clearly shows improved de-aliasing performance in most of the global ocean areas. It has consequently been integrated in satellite altimetry Geophysical Data Records (GDRs) and gravimetry data processing, and adopted in recently renewed ITRF standards (International Terrestrial Reference System, 2020). It also provides very accurate open boundary tidal conditions for regional and coastal modelling.

## 1 Introduction

The FES2014 global ocean atlas is the latest release of a twenty-years-long effort to improve tidal predictions needed in satellite altimetry de-aliasing. It is based on the hydrodynamic modelling of tides (Toulouse Unstructured Grid Ocean model, further denoted T-UGOm) coupled to an ensemble data assimilation code (Spectral Ensemble Optimal Interpolation, denoted SpEnOI). It is a very significant upgrade compared to both FES2004 (Lyard et al., 2006) and FES2012 (Stammer et al., 2014) atlases, thanks to the improvement of the assimilated data accuracy and the model performance. To some extent, FES2014 can be considered as an iterative step of the FES2012 atlas, mostly motivated by the overwhelming progress made in the hydrodynamic solutions accuracy toward the end of the FES2012 project and which could not be incorporated due to the project schedules. As will be further mentioned in this publication, the efficiency of data assimilation increases significantly with prior solutions accuracy, and for two main reasons. First, despite a rigorous theoretical framework, data assimilation relies on strong assumptions in which the choice of the vector norm chosen to build the penalty function is critical (the most commonly used nom is $L_2$-norm, which is consistent with a Gaussian-shaped error probability density assumption and which leads to easily resolved linear systems, but also which tends to over-weight outliers in data or simulation values, see Bennett, 1992 and Tarantola, 2005). Data

assimilation must also be fed with quasi-empirical, partially subjective parameters, such as error covariances assigned to data sets. So while correcting prior (hydrodynamic) solution errors, it can also inject some methodological errors in the assimilation solutions, more or less proportional to the prior distance between the observations and the numerical solutions. Second, as we use an ensemble technique to assess the prior modelling error covariances, and as those covariances will strongly dictate data assimilation innovation in model regions where assimilation data density is very sparse (sparse must be understood as compared to the tidal wavelength, hence being quite different in shallow water seas compared to deep ocean regions), the prior hydrodynamic realism is critical to consistently propagate information from data locations (where data/prior model trade-off is actually solved) toward "remote" model regions. Therefore, considering the significant potential improvements and thanks to the financial support of CNES (Centre National d'Etudes Spatiales), the decision was made to rapidly upgrade the FES2012 atlas toward the FES2014 atlas.

The FES atlas series started with the FES94 release, quickly followed with the FES95 one (Le Provost et al., 1998), which included some upgrades and fixes for various issues detected after the FES94 official release. A similar scenario occurred for the FES98 and FES99 (Lefevre et al., 2002), FES2002 and FES2004, FES2012 and FES2014 atlases production. Despite intensive quality checking during the production phase, any new major version of FES atlas release is followed by an extended verification/validation phase from the FES team and other worldwide specialists through the science applications that use the new atlas. The upgrading/fixing step is limited to issues that do not demand any major changes in the production process (such as unstructured grid modifications) but still will bring valuable improvements for the final user. The FES2014 atlas denomination is quite misleading, as its final version has been delivered in early 2016. This has left time to the project team to precisely assess the FES2014 accuracy and performance in altimetry data de-aliasing correction, and to make some final adjustments to guarantee the best possible quality at that time. It results in 3 available FES2014 releases. FES2014a is the first guess based on a data assimilation set where altimetry data were corrected from tidal loading provided by the GOTv8 model (Desai and Ray, 2014). Its production allowed for internal verification checks and data assimilation adjustments, and finally the production of the self-consistent FES2014a tidal loading atlas used within the FES2014b altimetry assimilation data processing. The FES2014a atlas was not intended to be widely distributed or advertised. FES2014b was the first official release and, after re-gridding from the native unstructured grid onto a regular 1/16$^{th}$ degree resolution grid, it has been made available on the AVISO+ website (https://www.aviso.altimetry.fr/en/data/products/auxiliary-products/global-tide-fes.html). To provide a more comprehensive, coherent tidal spectrum for tidal predictions particularly for the geodetic community, several long period tide constituents were explicitly added in 2019 (computed from the usual mass-conservative equilibrium approximation) to the FES2014b atlas. It must be noticed that similar long period constituents are implicitly added in tidal prediction if no corresponding external solution file is provided. To avoid confusion in public releases, the extended FES2014b atlas has received the FES2014c denomination.

The objectives of our communication are to concisely present the FES2014 atlas main construction details, the validation diagnostics and the available by-products, and not to propose a dissertation about tidal science findings based on this atlas which would lead us much too far. Consequently, in the following sections, we intend to provide to the reader with information on the major ingredients of the FES2014 atlas production (hydrodynamic modelling, data processing and data selection for assimilation and validation, assimilation processing), and a basic accuracy assessment overview. Complementary to the present publication, some additional

	information on present and earlier FES atlases and a link to the associated prediction software can be found on the AVISO+ website.

## 2 Hydrodynamic prior solutions

One primary objective in the FES2014 atlas production is to dynamically model the ocean tides with the best possible accuracy, and to keep the data assimilation correction as limited as feasible, hence limiting the atlas
	dependence upon altimetry-derived data and altimetry errors (Zawadzki et al., 2018).

### 2.1 T-UGOm time-stepping and frequency-domain solvers

T-UGOm (Toulouse Unstructured Grid Ocean model, Mercurial repository at https://hg.legos.obs-mip.fr/tugo/) is a 2D/3D unstructured grid model developed at the Laboratoire d'Etudes en Géophysique et Océanographie Spatiales (LEGOS). It can accommodate a variety of numerical discretizations (continuous and dis-continuous
	finite elements, finite volumes) on triangle or quadrangle elements, based on the usual Navier-Stokes equation in the Boussinesq approximation, with a non-hydrostatic pressure solver available. It can be used in time-stepping (TS) or frequency-domain (FD) mode. In 2005, based on FES2004 experience, an internal tide wave drag parameterization (ITWD) has been implemented for 2D shallow-water simulations (characterizing the energy transfer from the barotropic tides to the internal, baroclinic tides). The ITWD parameterization, originally
	developed from the pioneering work of Bell (1975) and Baines (1982) proved to be essential in tidal and storm surges simulation accuracy, as tidal energy conversion accounts for a significant portion of the total barotropic energy dissipation. Most of the critical dynamical parameters (such as bottom roughness, internal tide drag coefficient, etc...) can be non-uniformly prescribed inside the domain. Initially, the frequency-domain mode has been integrated in the original T-UGOm time-stepping code to dynamically and consistently downscale tidal
	boundary conditions for domain-limited, time-stepping simulations (actually, some classes of open boundary condition time-stepping schemes, such as Riemann invariants, require prescribing tidal velocities in conjunction with tidal elevations. Contrary to elevations, velocities are very sensitive to bathymetry and grid resolution, and a simple interpolation from a global atlas, with different bathymetry and resolution, may not meet the necessary consistency with the domain-limited configuration. A FD simulation, where only elevations are prescribed at
	open boundaries, will produce a properly downscaled velocity field over the domain-limited grid, including open boundaries). The FD solver is run for each tidal component separately. It basically assembles a frequency-domain wave equation and the solution is obtained by a simple inversion of the system. Naturally, the FD solver is based upon linearized equations, and subsequently non-linear processes require an iterative approach to converge toward the fully non-linear solutions. The number of iterations is rather limited for the major
	astronomical tidal components; it tends to increase when addressing compound tides and overtides. In any case, the numerical cost of the FD solver is extremely small compared to the TS solver cost (more than 1000 times smaller). In terms of solution accuracy, FD and TS solvers are quite equivalent, with of course a limited advantage to the TS solver in non-linear tides cases. Therefore, in the perspective of data assimilation using ensembles for the major ocean tides components, the ensemble members have been computed in the FD mode
	(details of data assimilation are described in a dedicated section of the article). Another major advantage of the FD solver's reduced numerical cost is the possibility to conduct a wide range of experiments in order to (globally or regionally) test numerical developments, calibrate the model parameters such as bottom friction and internal

tide drag coefficients, verify bathymetry improvements, or examine loading and self-attraction consistency. It must be noticed that the optimal parameter set for the FD mode will also meet the TS mode requirements. Both solvers are discretized through the standard finite element, variational (weak) formulation. Consequently, solutions must be handled in a consistent manner, especially when expressing conservation laws (which hold in a "weak sense") or estimating energy budgets.

**2.2 FD discrete equations**

The T-UGOm FD solver is originally inspired from the "Code aux Eléments Finis pour la Marée Océanique" (CEFMO model, Le Provost and Vincent, 1997; Lyard et al., 2006) frequency-domain tidal model that was previously used for the FES atlases (such as FES2004). The frequency-domain tidal equations and wave equation construction have been extensively described in the literature. Consequently, we will confine ourselves

to the main differences between the CEFMO and T-UGOm formulations. The FES2014 mesh is built on triangle elements. Various numerical discretizations for elevations and currents can be defined on triangle elements, i.e. continuous or discontinuous, high or low order. Since its early releases, the FES tidal atlases mesh has been designed in terms of spatial resolution for continuous LGP2 discretization (quadratic Lagrange Polynomials basis functions, allowing for about 4 times more numerical nodes compared to linear Lagrange Polynomials,

denoted LGP1). Among other available options, tidal velocity discretization is element-wise, discontinuous non-conforming linear interpolation function (NCP1). This choice has two major advantages: the elevation gradient discrete space is identical to the tidal currents space, and the discrete momentum equation system is diagonal, easing the construction and solving of the wave equation. Non-diagonal terms, such as horizontal momentum diffusion, must be left in the right-hand side vector and converged in an iterative manner, or simply dismissed (in

time-stepping codes, momentum diffusion acts mostly as a temporal scheme stabilizer, which is not needed in the frequency-domain solver). Tidal currents are expressed under a standard Galerkin procedure and this is one of the major differences with the CEFMO model where currents were estimated at numerical integration nodes (Gauss quadrature).

**2.3 TS discrete equations**

Quite similarly to the FD equation, the TS 2D shallow-water equations in T-UGOm are based on the so-called generalized wave equation. Inspired by Lynch and Gray (1977), and continuously developed since, the approach has evolved from application to the global ocean, now up to the inclusion of near-shore and estuarine numerical applications, with wetting/drying and non-hydrostatic (surface wave dynamics) capabilities. Although it allows for pressure instability modes, the discretization used in FES2014 simulations is (linear) LGP1 both for

elevations and currents, for its numerical efficiency. As a matter of fact, the potential pressure instabilities will appear only in some peculiar local mesh geometry and are easily avoided by precisely controlling the mesh construction (Leroux et al., 2007). From its earlier versions, T-UGOm includes an embedded, multi-level, time sub-cycling that allows for locally modifying the numerical time step. It is coupled to a simulation stability control procedure, and sub-cycling is locally triggered and disabled following the need to control this stability on

the fly. This turns out to be a very efficient way to relax time step limitation due to the Courant–Friedrichs–Lewy (CFL) stability condition (already eased by T-UGOm semi-implicit time scheme) and therefore to profit from the natural flexibility of unstructured triangle grids. Contrary to the FD solver, horizontal momentum

diffusion is needed to fully stabilize the temporal, centred-in-time leapfrog-like scheme, and is provided by a
       Laplacian operator with Smagorinsky's diffusion coefficient scheme.

**2.4 Model grid settings**

       Since the first truly global ocean atlas (FES2004), the unstructured FES model mesh has been upgraded by using
       regional patches. The main meshing difficulty consists in dealing with the shoreline details. Present databases
contain a high level of small scale coastal details, much more than needed for a global ocean mesh. These small
       scale details consequently need to be filtered out according to the targeted coastal resolution. Conversely, it is
       necessary to maintain and assemble together some packets of micro-islands that will form a macro-obstacle to
       the tidal propagation. Considering the tedious task of re-meshing most of the ocean shorelines, automated tools
       have been developed to optimize the meshing operation. The targeted resolution for coastal areas is typically 10
kilometres or less in terms of triangle side-length (shown in Figure 1; the mesh details will not be visible on a
       printed global ocean figure, the authors have provided a zoomable supplementary pdf file available on the Ocean
       Science website https://www.ocean-science.net). The resolution has been augmented to about 1.5 km in some
       specific places where coastal geometry is more challenging (such as fjords, estuaries, straits, etc…). Special
       attention was paid to regions where the accuracy and the precision of the available bathymetry are known to be
adequate with higher mesh resolution, i.e. where mesh details will truly reflect the bottom topography
       complexity. On the other hand, only minor upgrades were made in regions where the bathymetry remains poorly
       known (such as the Patagonian and Siberian shelves). As a matter of experience, increasing resolution in those
       regions would likely have a model accuracy worsening effect. An additional constraint was to limit the
       hydrodynamic solver memory use to 30 Gbytes in order to keep computation load at a tractable level (at the time
of production). Despite the large increase in resolution compared to FES2004, the FES2014 mesh resolution is
       still clearly not sufficient in some highly complex coastlines, with narrow channels of dynamical significance, or
       topographically trapped wave generation sites, and it could result in a loss of details/accuracy in such regions.
       This is for instance the case of the western Canadian and Alaska coastal regions (where the project failed to
       access any accurate bathymetry database at the time of production and so left resolution at a standard level), and
it has resulted in a loss of details/accuracy in all of this area, especially away from assimilated data. Following
       the FES2014 atlas release and thanks to our collaboration with the Canadian tidal research community, this issue
       has been identified as quite damaging. This issue and similar ones such as around the Tierra del Fuego
       (Argentina and Chile) will be fixed in a future FES atlas release, where the number of computational nodes
       should be increased at least by a factor five compared to the FES2014 grid.

**2.5 Model bathymetry**

       When dealing with tides, bathymetry remains one of the most critical parameters. Several global ocean databases
       were available at the FES2014 production time: General Bathymetric Chart of the Oceans (GEBCO, GEBCO
       compilation group, 2020), Earth Topography (ETOPO, Amante and Eakins; 2009), Smith&Sandwell (Smith and
       Sandwell, 1997), etc... Their successive releases have shown tremendous improvements during the last ten years.
Unfortunately, none of those global databases have the effective resolution nor the accuracy needed to be used
       directly in our global ocean tides modelling. For example, satellite inverted bathymetry accuracy is very limited
       on shelves and in coastal regions (Gibb's effect due to the spherical harmonic technique, uncertainties arising

from sediment density, etc) , and consequently should not be used in such locations except in some specific areas, namely in the absence of any other more accurate bathymetry. It must be noticed that the latest GEBCO distributions now include patches derived from inverted bathymetries, which is a serious issue for using recent GEBCO distributions in FES model bathymetry. Consequently, as for the earlier FES atlases, a composite bathymetry has been built from available global and regional databases. In some cases, a regional digital terrain model (DTM) has been specifically constructed from depth sounding and/or multi-beam data. A special treatment is applied to the Ross and Weddell Seas, where the free water column depth must be processed by substracting ice-shelf immersion to the bottom topography, using the RTopo-1 dataset (Timmermann et al., 2010). Many regions of the world ocean are now quite well documented in terms of bathymetry, however two major continental shelves, namely the Patagonian shelf and the Siberian shelf, do not match modern standards in any publicly available database. Bathymetry selection, reconstruction and merging is a tedious task, and quite uncertain because of the lack of independent validation data. Finally, the most practical way to assess bathymetry changes remains the examination of the tidal solutions computed from the candidate bathymetry. Naturally this is not a perfect measure of accuracy, as errors in bathymetry can compensate some other modelling errors, but so far we have always found consistent results between improvements in bathymetry and tidal solutions. Thanks to the FD solver, extensive simulation testing can be performed, including the necessary re-calibration loop needed when modifying significantly the model bathymetry, even on a regional level, as earlier calibration settings would be biased to compensate errors due to the former model bathymetry. Despite those efforts, bathymetry still remains unfortunately the limiting error to our prior hydrodynamic solutions in most of the global ocean, and also impacts the data assimilation accuracy in shallow waters regions. For most of Northern American, European and Japanese waters, bathymetry-linked errors are reducing with time, allowing for distinguishing more subtle error sources. For instance, thanks to the impressively accurate new bathymetry of the European shelf (as available through the European Marine Observation and Data Network (EMODnet) website, https://emodnet.eu), most of errors due to bathymetry have dramatically reduced, so we could clearly demonstrate (in a regional configuration) that a wetting/drying time-stepping scheme is necessary to reach the best tidal accuracy in the North Sea. Using older bathymetry would have totally blurred this point, making any conclusions uncertain. But in most of the global ocean, improving the model bathymetry remains the first and overwhelming priority, and enormous efforts have been dedicated to this in FES2014 hydrodynamic configuration settings.

**2.6 Loading and self-attraction effects**

Geometrical loading and gravitational self-attraction terms (LSA) are essential in tidal simulations, especially in global ocean tidal modelling (Hendershott, 1972). They can be implicitly accounted for in the hydrodynamic tidal equations, but at a totally prohibitive computational cost. As rather accurate LSA atlases are available since the early 2010's, it is much more efficient to use explicit LSA in the simulations, not only for computational cost reasons (non-sparse dynamical matrices in FD, expensive convolutions in LSA computation), but also because it tends to provide a relaxation toward the tidal atlases from which the LSA have been computed (actually, this is the only model ingredient which depends upon pre-existing ocean tide information in our hydrodynamic simulations). As some anomalies were detected in the LSA atlases deduced from FES2004, we used instead the FES99-derived LSA atlases to produce a first version of FES2014 (FES2014a), from which a new LSA atlas was

computed. As it will be mentioned in the following sections, this new LSA atlas was used in the final FES2014b release production.

**2.7 FES2014 hydrodynamic (assimilation-free) solutions**

Some parameters of the T-UGOm hydrodynamic model need to be calibrated in order to obtain the most accurate hydrodynamic solution, either to improve model realism or provide useful error compensation. The two main parameters to which the model is the most sensitive are the bottom friction coefficient and the internal tide drag coefficient. Most of T-UGOm model parameters can optionally be tuned locally using various methods (pre-defined regions, polygons inclusion, or by mesh node or element vectors). In the FES2014 atlas simulations, internal wave drag coefficients are tuned using a global ocean regional partition (distinguishing north, tropical, and south basins in the various oceans plus Arctic Sea and Mediterranean Sea), and bottom frictions coefficients are tuned by using polygons covering the large bottom friction dissipation areas. A global default value is locally used in regions not being targeted by the user-defined partition/polygons tuning list. Several simulations of the main tidal components (limited to M2, K1, S2 and O1 constituents) have been performed by extensively varying these two parameters (mostly globally except in a few regions for the internal tide drag coefficient), and each resulting simulation was compared to the altimetry and tide gauge (later denoted TG) validation databases. Figure 2 and Figure 3 show the vector differences between the TP/J1/J2 (deep ocean) crossover point database and the hydrodynamic simulations of the FES2012 and FES2014 tidal models, for the M2 and K1 tidal components, respectively. Global values of vector differences are given in Table 1, for the same two hydrodynamic simulations plus FES2004. These results clearly point out the improvement that has been achieved from the FES2004 to the FES2014 free simulations on the global ocean, with a global vector difference RMS reduced by nearly a factor of three from FES2004 to FES2014 (M2 tidal component) in the deep ocean. The improvements are also very strong in the shelf regions, and for the other main tidal components. Moreover the histograms displayed in the "5.2 Validation " section indicate that the FES2014 hydrodynamic solution reaches an unprecedented accuracy level, close to other global ocean model performance as such GOT4.8/10 (Ray, 2013), EOT11a (Savcenko and Bosch, 2012), DTU10 (Yongcun and Andersen, 2010) or TPXO9v2 (Egbert and Erofeeva, 2002), which are all empirical or assimilated models.

The case of the S2 tidal components was specifically addressed, as it derives both from atmospheric and gravitational forcing. It is even more the case for the S1 tide, which originates mostly from atmospheric forcing, but because of the intrinsic variability of the atmosphere we consider that it must be dealt with in the storm surge correction (DAC), and not in ocean tidal corrections. Some other tidal constituents have a clearly atmospherically-forced component (such as K2 and even M2 tides), but at a much lower level. Consequently, to ensure the best possible prior solution, the S2 wave was computed in the spectral domain using atmospheric pressure forcing at S2 frequency, based on ERA-interim 3-hour data (Berrisford et al., 2011). There are numerous difficulties arising from the atmospheric pressure forcing at tidal frequencies (impacting tidal hydrodynamic solutions, de-aliasing corrections and data processing), so additional discussions on S1 and S2 constituent issues are given in the following sections.

## 3 Tidal harmonic constant data processing

TG and altimetry-derived harmonic constant data have been used in validation of simulations and data assimilation steps. Concerning the TG data, preference was given to TGs for which the original time series were available and documented, hence for which basic quality control could be performed by means of harmonic analysis and/or operational reports. In most cases, the time series were long enough so that a wide tidal spectrum could be analyzed with the best possible accuracy. To some extent, TG selection (either for validation or data assimilation purposes) is more a question of how representative are the tides captured by the instruments (especially in coastal seas) and keeping a balanced distribution all over the ocean regions. Several tidal gauge databases have been used within the FES2014 project: a harmonic analysis was performed on time series from GLOSS (Holgate and al., 2013) and SONEL (Wöppelmannn and Marcos, 2016) databases, GLOSS being a global observation network and SONEL providing measurements on all French territories; then three validated databases provided by R. Ray have been used (Ray et al., 2013), named Deep_BPR (Bottom pressure recorders), Shallow and Coastal hereafter and dedicated to deep ocean, shallow waters and coastal regions respectively.

The altimetry-derived time series raise more processing and accuracy issues, with a strong dependence on the mission orbit and duration (which firstly determine the level of contamination of the tidal analysis by non-tidal ocean signals). Clearly, the twenty years and more duration of the Topex-Poseidon and Jason series on a nearly ten-day repeat orbit allows for deriving outstandingly high-quality along-track and cross-over datasets of tidal harmonic constants (Topex-Poseidon, Jason-1 and Jason-2 are three CNES/NASA satellites, successively launched, having exactly the same ground-track and repetitivity, and similar on-board instruments and radar technologies; since the FES2014 release, the series has been continued with the Jason-3 and Jason-CS satellites; at the end of their nominal missions, a satellite's orbit is changed toward an exactly interleaved ground-track, hence doubling the mission spatial sampling until a possible move to a geodetic orbit or final decommissioning; interleaved tracks observations are not continuous in time and thus have shorter records compared to the nominal track records). Moreover the altimetry dataset benefits from new altimeter standards, which allow a better observation of the tidal signals: GDR-D and REAPER orbits, ERA-INTERIM Dynamic Atmospheric Correction for ERS and TOPEX missions, improved wet tropospheric, sea sate bias and ionospheric corrections, and new mean sea surface profiles computed over a 20-year period (Carrère and Lyard, 2003; Carrere et al, 2016). TOPEX-Interleaved and Jason-1-interleaved track (denoted TPN-J1N) also provides an accurate crossover dataset, but with larger uncertainties than the 20 years of TP-Jason series, due to the shorter, cumulative period of 6 years available. ERS/Envisat series and Geosat Follow On (GFO) series do not have the same level of accuracy, as their orbits offer higher spatial coverage at the price of a lower temporal coverage (time sampling of 35 days for ERS/Envisat and 17 days for GFO). The temporal under-sampling of tidal observations affects the apparent tidal periods (aliasing effect) which depend on the true tidal periods and on the mission temporal repetitivity. Because of the red nature of the ocean energy spectra, the longer the aliased period, the larger the contamination of the tidal signal by non-tidal signals. The TP/Jason orbit was deliberately chosen to maintain the aliased period in a reasonable range. Conversely, sun-synchronous orbits (such as ERS/Envisat/Altika) are disadvantageous in that matter: not only are the S1 and S2 tides projected on an infinite period (mean state), but many other tidal constituents show a rather large aliased period (cf Table 2). This would prevent us to use ERS/Envisat derived data, and concentrate only on the Topex/Jason dataset, however the inclination of Topex/Jason is rather low and ERS/Envisat remains the only choice for very high latitudes and polar seas. Thus

for the purpose of the FES2014 tide model, crossovers and along-track data from TOPEX/Jason-1/Jason-2 were preferred and were complemented with some crossover data from TPN-J1N and ERS-Envisat series in some shallow water regions and at high latitudes respectively. Table 3 presents the altimeter dataset used for the estimation of the harmonic constants within the FES2014 project.

**3.1 Tidal loading effect**

As the standard tidal atlases are targeted on the ocean tide component, a tidal loading correction needs to be applied to the altimeter measurements (in addition to the so-called solid earth deformation correction). In a first step, the GOT4v8ac tidal loading model was applied (Ray 2013), taking into account the recent correction of the tidal geo-center motion proposed by Desai and Ray (2014). These data have been used in the data assimilation process for the preliminary version of the ocean tide model, denoted FES2014a. In a second step, a new tidal loading atlas was computed from this FES2014a ocean solution, denoted "FES2014a tidal loading" (cf. section 6.3). Then, this FES2014a tidal loading solution was used to produce a second version of the altimeter dataset, which was assimilated into the final version of the tide model named FES2014b.

**3.2 Non-tidal signal at K1 aliased period prior removal**

Due to the aliasing effect, the K1 diurnal frequency is aliased to the semi-annual frequency with the TOPEX/Jason sampling and to the annual frequency with the ERS/Envisat orbit (cf. Table 2). Annual and semi-annual signals are quite large in the ocean, and contamination of tidal analysis by the non-tidal signal is severe. By virtue of the Parseval Identity (the identity asserts that the sum of the squares of the Fourier coefficients of a function is equal to the integral of the square of the function, see Johnson and Riess, 1982), this contamination decreases with time as the square root of the recording duration. The present reference TOPEX-Jason time series benefits from 20 years of continuous measurements and allows a very accurate estimation of all tidal components including K1. However, for the TPN interleaved and the ERS orbits, the available time series are not long enough to guarantee an accurate separation of the K1 tidal signal from the semi-annual (resp. annual) ocean variability. A large portion of the annual and semi-annual ocean surface signal is due to the low frequency atmospheric surface pressure, and therefore is removed by applying a storm surge or inverted barometer correction. However, the ocean circulation contributes also to this signal, and so to tidal harmonic contamination. To tackle this issue, and then improve the K1 tidal signal observation in the TPN and ERS/Envisat records, a specific processing has been applied, consisting in removing an estimation of the ocean annual (Sa) and semi-annual (Ssa) non-tidal signals prior to the analysis. This estimation is computed from the GLORYS2-V1 global ocean reanalysis provided by Mercator-Ocean (Ferry et al., 2012). GLORYS produces and distributes global ocean reanalyses at eddy-permitting (1/4°) resolution that aim to describe the mean and time-varying state of the ocean circulation, including a part of the mesoscale eddy field, over recent past decades with a focus on the period since when satellite altimetry measurements of sea level provide reliable information on ocean eddies (i.e. from 1993 to present). The numerical model used is the NEMO OGCM in the ORCA025 configuration developed within the DRAKKAR consortium (global with sea-ice, 1/4° Mercator grid). Assimilated observations are in-situ temperature and salinity profiles, satellite sea surface salinity (SST) and along track sea-level anomalies obtained from satellite altimetry. GLORYS2v1 products are free of atmospheric surface pressure effects (i.e. they are not taken into account in the NEMO model forcing and are corrected for in the assimilated SSH data). Consequently, they are comparable to IB-corrected sea level (at Sa and Ssa frequencies) in altimetry and tide gauge observations. GLORYS2-V1 sea

surface height (SSH) has been harmonically analyzed at semi-annual and annual frequency, predicted at observation location and time and removed from altimetric SSH measurements. The efficiency of the non-tidal ocean signal contamination has been assessed at TP/Jason cross-overs, where the K1 harmonic constant misfits between ascending track and descending track analysis are diminished. As shown in Figure 4, the amplitude of the correction is well above a few centimeters in some large ocean regions. A specific study (Gulf of Tonkin) was performed by examining the K1 analyzed tidal constant misfit at cross-overs (ascending track versus descending track). The ocean circulation contamination will appear as an incoherent contribution to K1, and will be different for ascending and descending tracks. Such differences were found to be consistently reduced when applying the GLORYS correction, hence demonstrating the benefits of the model-based correction for the tidal analysis accuracy.

**3.3 S2 tidal constituent processing**

S2 tide harmonic analysis needs a special attention both in TG and altimetry time series. Because of the significant S2 atmospheric tide, especially in the tropics, bottom pressure records must be precisely corrected from air pressure contribution to retrieve the S2 ocean signal. This is easily done for coastal TGs, from dedicated or neighbouring atmospheric pressure records. Deep moorings in remote ocean regions are more problematic, especially for records made before the quite recent availability of hourly pressure fields in operational atmospheric products. In altimetry mission observations, the S2 tidal constituent is challenging as it is aliased to infinite period and thus is not observable by the ERS/EnviSat sun-synchronous orbit as mentioned before. The TP-Jason orbit is adequate for the observation of most of the main tidal constituents. However, because of its 58.74-day aliased period, the S2 tide sea surface signal is mixed with the residual Mean Sea Level (MSL) signal visible at the same frequency in the TP-Jason time series, which in turn is linked to inaccuracy in the β' angle in MSL computations (Ablain et al. 2010; Zawadzki et al. 2016). Consequently, S2 harmonic analysis will be contaminated by this GDR processing-dependent signal (with a possible feed-back through the tidal corrections in the GDRs, making this issue even more complicated). As this problem is larger for the TOPEX-POSEIDON mission GDRs (as reported in Zawadzki et al., 2016), several analyses have been performed using either the entire TOPEX-Jason time series or only the Jason-1/Jason-2 relatively recent records. But due to the much shorter duration of the latter, the estimation error is larger for the J1-J2 only analysis, and the assimilated solution proved finally to be more accurate (using TG data as sea truth) using the analysis from the entire altimeter series. Notice that thanks to its primary emphasis on accurate hydrodynamic modelling, further moderately tuned by data assimilation (thus allowing a reduced weight of the data and data errors in the global FES solution), the FES2014 S2 solution is less affected by this residual GDR processing signal than empirical models, with in addition a beneficial effect on reducing the residual MSL error if used for tidal corrections in GDR processing (Zawadzki et al., 2016).

**3.4 Numerical Rayleigh criterion**

When extracting a comprehensive tidal spectrum from a sea level time series, the question of frequency separation must be examined carefully (Cherniawsky et al., 2001). Not only can the contamination by non-tidal signals at the aliased frequency be comparable to a given constituent amplitude (especially the minor constituents), but also the minimum observation duration for a proper separation is greatly increased in the

aliased frequency space. For instance the N2 and T2 pair needs about a minimum of seven years duration (in TP-Jason observations) for a proper separation instead of the usual ten days in the non-aliased frequency space. The data assimilation spectrum in FES2014 is a mitigation between the objectives of extending the tidal correction spectrum and limiting data assimilation to accurately observable tidal constituents, and we have developed a new numerical approach to address the frequency separation issue. In the case of a continuous (i.e. uninterrupted or sparsely interrupted) time series, the Rayleigh criterion is classically used to determine frequency separation and some additional parameterization (based on the smoothness credo, or admittances) can be implemented to ease the harmonic system solving. For TGs as well as for most of the altimetry-derived time series, the Rayleigh criterion will be appropriate to predict rather accurately the harmonic separation performance. However, in the case of high-latitude altimetric time series, the seasonal sea ice cover is responsible for annually unbalanced observations, with data gaps duration that can be comparable to the aliased wave frequency. In that case, it has been observed that the Rayleigh criterion will return over-optimistic diagnostics. This turns into an ill-defined harmonic system, and consequently larger errors in the harmonic constants deduced from its solving. Neither high-latitude data set manual editing nor entire data set rejection were options, the former being a gigantic task and the latter an extremely damaging loss of data in already poorly documented regions. Instead, we directly examined the ratio between the diagonal and extra-diagonal terms in the numerical harmonic matrix, and we used an analogy with the Rayleigh criterion for continuous time series (and the corresponding harmonic matrix) to decide on a maximum ratio (extra-diagonal/diagonal) above which the frequency separation was considered deficient. The maximum ratio is set by analogy with the Rayleigh criterion. Ideally, i.e. in case of quasi-infinite time series, the harmonic matrix will be quasi-diagonal. The shorter the time series, the larger the cross-terms/diagonal-terms ratio in the matrix, which reflects the loss in separation efficiency. In the case of a regularly sampled, continuous time series (no data missing), the usual Rayleigh criterion (at least 1 period difference between two different constituents over the time series duration) is equivalent to a maximum ratio of ~0.15 in any row of the harmonic matrix. In the case when 2 constituents show a ratio larger than 0.15, we check whether admittance can be used to infer the one with the lowest astronomical potential or not. If not the case or if at least one is a non-astronomical constituent, it is dismissed from the harmonic analysis spectrum.

### 3.5 Filtering internal tide signatures

FES2014 is a barotropic tide model and it is not aimed to include the small scales of the internal tide signals by definition. Thus internal tide surface signatures have to be removed from the altimeter data prior to data assimilation and validation processes. Internal tides have much shorter wavelength (and much lower phase speed) than barotropic tides, and their juxtaposing with barotropic tides creates well known ripples in the along-track harmonic analysis due to in-phase/out-of-phase changes (Egbert and Ray, 2001). So low-pass filtering is a convenient way (still imperfect as it is vulnerable to the baroclinic waves propagation angle with respect to the ground-track) to separate barotropic and baroclinic tide components for each frequency. Based on gravity wave vertical modes theory (Gill, 1982), new estimates of the first vertical mode, baroclinic wavelengths have been computed for the main waves M2, N2, S2, K1 and O1, using WOA2009 climatology (Locarnini et al., 2010, Antonov et al., 2010). First mode baroclinic tides show the largest wavelengths, which are roughly in the 100 to 150km range in the deep ocean and much shorter on shelf seas. Still, barotropic tides have short wavelength in their amplitude and phase distribution, for instance close to amphidromic points or at shelf edge crossing, that

should not be filtered out. The barotropic tide wavelength has been numerically computed from the FES2012 atlas (by estimating the local wavenumbers from the ratio of the complex Laplacian of the tidal elevation field and the tidal elevation field itself) and both barotropic and baroclinic estimates were then used to compute the along-track low-pass filtering cutting length scale, which is the minimum between twice the baroclinic wavelength and $1/15^{th}$ of the barotropic one. Figure 5 shows the filtering cut-off length scale in km: it goes to zero in near-amphidromic point areas and in shallow waters where the wavelength of the barotropic tide becomes shorter.

## 4 Data assimilation

The data assimilation method used in FES2014 is quite similar to the one used in FES2004, with the notable exception that the ensemble approach has been substituted by the variational one. This change in our approach, initiated after FES2004 completion, is motivated by the difficulty to prescribe bathymetry errors as right handside, forcing terms errors, as a variational technique would ask for. More generally, the ensemble technique is much more flexible and natural, especially when dealing with highly inhomogeneous error sources, in nature and magnitude, as is the case for shelf and coastal tides.

### 4.1 SpEnOI assimilation code

The SpEnOI (Spectral Ensemble Optimal Interpolation) data assimilation code is an evolution of the Code d'Assimilation Océanique par la méthode des Représenteurs (CADOR) data assimilation code (Lyard 1997, used up to FES2004), based on a variational approach using representer method, originally inspired from Bennett and MacIntosh (1982). The main difference lies in the fact that CADOR uses a variational formulation to infer the tidal elevation error covariance matrix, using an adjoint system. Although the variational approach is quite well designed to capture model errors arising from the right-hand side of the tidal equations (linear forcing terms), it turns to be poorly able to account for bathymetry-derived and non-linear terms (bottom friction) errors that usually dominate modelling errors in coastal and shelf seas. For this reason, an ensemble approach has been constructed to improve the realism and flexibility of the modelling error prescriptions. The optimal interpolation denomination is a mis-nomer as the error covariances of the state vector are not idealized covariances (such as Gaussian-shaped distribution), but are justified by the non-incremental nature of the data assimilation due to the frequency-domain space where it applies.

### 4.2 Ensembles construction

In the ensemble assimilation approach, a large number of simulations are run in order to describe the model errors. This ensemble of simulations is generated by varying the parameters and input datasets to which the model is the most sensitive. In the case of the FES2014 tidal model, the perturbations were made on the bottom friction coefficient, the internal tide drag coefficient, the bathymetry and the LSA. All the simulations were validated against the altimetry and the TG databases, in order to identify potential outliers. In addition, the dispersion of the ensembles and the distance of the ensemble mean to the reference hydrodynamic simulation were computed, in order to verify that the ensembles were centered on the reference. In total, the whole ensemble contains 432 simulation members for each tidal constituent, built by following the methodology described in the next sections.

**Perturbation of the loading tide**: Numerical experiments have shown that the model is very sensitive to the explicit LSA forcing, with tidal species dependence. Namely, the diurnal tidal components (K1, O1) are improved when using the FES2012-derived LSA, while the semi-diurnal tidal components (M2, S2) are better resolved when using the FES99-derived LSA. The latter result needs some explanations: first the FES2014 hydrodynamic configuration has been adjusted (i.e. bottom friction and internal wave drag due to barotropic to baroclinic energy conversion, denoted IWD) in simulations using the FES99 LSA, and including clearly an error compensation contribution, i.e. configuration adjustments compensate for the FES99 LSA defects. Consequently, considering the high level of accuracy of the hydrodynamic solutions and thus the sensitivity to any minor changes, they are not fully appropriate for a simulation forced with another LSA atlas; second, the most sensitive component in the adjustment process is clearly M2, as bottom friction is truly non-linear for M2, as it has the strongest currents and the dominates the velocity amplitude in the non-linear friction term, and as the other constituents have consequently a kind of quasi-linear friction in the presence of M2 dominant velocities. So using a more modern and more accurate LSA, will usually profit all constituents but M2, as it would require re-processing the adjustment steps to get back at least to a similar or improved accuracy. In order to obtain a thorough description of the model errors, all the simulations based on perturbations were done twice, using the FES99 and the FES2012 loading tides as input, respectively. This doubled the number of members in the ensembles described hereinafter.

**Perturbation of the bottom friction roughness**: Figure 6 shows the energy dissipated by the bottom friction in the FES2014 hydrodynamic model, for the M2 tidal component. As expected, the areas where the dissipation is the largest correspond to the shelves and coastal seas. The model is consequently more sensitive to the bottom friction coefficient in these areas. Following this map, thirteen polygons, highlighted in red in Figure 6, were defined in order to generate local perturbations of the bottom friction coefficient in significant bottom friction tidal dissipation regions. The definition of tuning polygons is a compromise to include the most significant sites for tidal dissipation and to limit the number of polygons (to avoid too many members in our ensembles). For each of these polygons, the bottom friction roughness was assigned eight different values ranging around the global-average value set for the reference hydrodynamic simulation ($10^{-3}$m). As presented above, all the simulations were done twice, with the FES99 and the FES2012 loading tides respectively as input, and the ensemble of bottom friction perturbations finally contains 208 members.

**Perturbation of the wave drag coefficient**: Contrary to the bottom friction, the energy dissipation due to the energy transfer from the barotropic tides to the baroclinic tides (internal tide drag) does not happen in very specific and local regions, but in various, dispersed, sloping bottom topography regions (shelf edges, ocean ridges) where the tidal currents cross the bathymetry gradients, making it difficult to isolate each single active site. In addition, energy transfer efficiency strongly depends on local ocean stratification, which is not precisely known in standard climatology or Ocean General Circulation Models (OGCMs). The perturbations of the wave drag coefficient were consequently done at the sub-divided basin scale (equatorial/tropical, mid-latitudes and high latitudes sub-divisions), shown in Figure 7. For each of these ten regions, the non-dimensional wave drag coefficient was locally varied over seven values ranging around the global-average value set for the reference hydrodynamic simulation (i.e. 75). The wave drag perturbations ensemble finally contains 140 members (70 perturbations run with each of the FES99 and FES2012 LSA).

**Perturbation of the model bathymetry**: Several approaches are possible for the hydrodynamic model bathymetry perturbations such as linear combinations of various datasets or modifications in specific regions using either synthetic or heterogeneous bathymetry dataset. The latter was used in the case of the FES2014 model, as it enables to better control the perturbations and to choose the most responsive regions. The reference hydrodynamic model bathymetry is replaced by depths extracted from what we call "gridone", 1 minute

resolution from GEBCO, and Smith&Sandwell, 15.1 release, in each of the 19 regions displayed in Figure 8 and chosen either for their dynamical impact on tidal solutions or for the large uncertainties of the reference bathymetry quality (such as the Patagonian shelf). However, the construction of the ensemble simulations has highlighted that the two bathymetry perturbations in the Weddell Sea (Southern Atlantic Ocean) resulted in solutions showing errors in semi-diurnal tides up to two to four times larger than the average simulations, with a

large increase of errors in the whole Atlantic Ocean, in the Indian Ocean and in the Southern Pacific Ocean. This comes from the free water depth reduction due to the Weddell Sea ice-shelf immersion, which has been corrected in our reference bathymetry, but not in the gridone and Smith&Sandwell patches because of project schedule constraints. Despite being considered as potentially critical for the model error space, the Weddell Sea region was discarded from the bathymetry patches ensemble construction, which effective set contains 36

members.

A few additional members have been added from the perturbations of the model minimal depth threshold. It is usually set to 10 m in the TUGO-m hydrodynamic, global ocean model. Depth threshold aims to minimize frequency-domain modelling validity limitations in very shallow waters (T-UGOm has the ability to modulate the threshold as a function of local tidal range, it was not used in FES2014 to avoid additional complexity in the

model configuration setting), but more importantly to deal with the existence of unrealistically shallow depths in most bathymetry datasets. The depths found in most bathymetry databases in the 0-10m (and probably 0-20m) range is anything but reliable. In most places, the depths are linearly varying with distance from 0m at coastline to the 10m isobaths, which is not the usual morphology one will find in the true ocean. Such artificial, very shallow water patches can have a damaging impact on the  bottom friction budget in coastal areas. The 10 m

limitation has been verified to be quite reasonable by experiments in the last 2 decades of tidal modelling. Of course, in regions where bathymetry databases are highly accurate, it is preferable to keep the true depths (and use a wetting/drying scheme if running the time-stepping mode). But it represents only a tiny portion of the global ocean coastal regions. Potential errors arising from this parameter have been taken into account by producing six members with global values centered around the standard value (10 m). In total, the ensemble of

bathymetry perturbations contains 84 members (42 perturbations run with each of the FES99 and FES2012 LSA).

**4.3 Data selection**

As described in section 3, the TG and altimetry sea surface height observations were processed with a harmonic analysis in order to retrieve the tidal harmonic constituents (amplitude and phase lag) for about fifteen tidal

components (M2, K1, S2, O1, etc…) and the associated error estimates. The altimetry data were processed at the crossover points for the TP/J1/J2, TPN/J1N and E1/E2/EN series, and along the tracks for the TP/J1/J2 series. This means a large amount of data, with more than 9 000 crossover points for each of the TP/J1/J2 and TPN/J1N series, about 64 000 crossover points for the E1/E2/EN series, and many more points along the TP/J1/J2 tracks.

In addition to severe computational cost (SpEnOI code is solving an assimilation problem in the data space), using data of the entire dataset is not optimal. First TPN/J1N and E1/E2/EN data can contain errors larger than the prior solutions' ones and associated error bars are not fully reliable, so their inclusion can degrade the resulting data assimilation accuracy. Second, previous studies have shown that a limited subset of high-quality data can perform as well as the full dataset. Thirdly, it is the long-going objectives of FES atlases to keep the weight of data assimilation at the lowest possible level and preserve as feasible of the hydrodynamic properties of the solutions (needed for instance to perform energy budgets). So the selection of the observations for the data assimilation process is driven by the following general guidelines: keep the overall assimilation dataset as limited as feasible; giving priority to the TP/J1/J2 cross-over data, with a partial decimation especially at high latitudes to favour the homogeneous repartition of the assimilated observations all over the global ocean; and add (possibly decimated) along-track TP/J1/J2 data to constrain more closely the model with the observations in problematic regions, i.e. where problems have been identified in the hydrodynamic solution, mostly linked with deficient bathymetry. Those regions lie mostly in shelf and coastal seas, where data could also be taken from TPN/J1N and E1/E2/EN along-track/cross-over datasets. However, due to the 20 years of data available during the TP/J1/J2 orbit at FES2014 production era, the tidal constituents' retrievals at the TP/J1/J2 crossover points and along the tracks are more accurate than the tidal retrievals at the TPN/J1N and E1/E2/EN crossover points. Consequently, TPN/J1N and E1/E2/EN along-track/cross-over datasets were not used, except in some rare exceptions. Of course, the Topex/Jason orbit being limited to 66° in latitude, the E1/E2/EN data are definitely needed as a complement in the northern high latitudes (E1/E2/EN data were not considered as being accurate enough in southern high latitudes).

The altimetry assimilation dataset was built in two steps. First, a systematic decimation was performed, following the criteria detailed in Table 4. A threshold on the error estimate of the M2 tidal constituents was also used as a selection criterion. As some observations provide accurate estimates for some given tidal components and show strong errors for other ones, data were decimated specifically by applying a threshold value to the error estimate associated with the considered tidal component. In particular, regarding the S2 tidal constituent, no E1/E2/EN data were selected, because of its infinite aliasing period (sun-synchronous orbit).

The second step of the construction of the altimetry assimilation dataset consisted of re-ingesting TP/J1/J2 crossover and along-track data that were discarded by the spatial decimation in regions where the model needed more close constraints, using an empirical, iterative procedure. The final dataset of altimetry crossover points selected for the data assimilation process is presented in Figure 9, with a specific color for each altimetry mission. One can notice there are fewer observations in the major ocean surface circulation areas (Gulf Stream, Kuroshio, Agulhas Current), because of the potentially large contamination by meso-scale dynamics (the non-tidal ocean dynamic contamination is estimated by looking at sea surface signal spectral energy close to the considered constituent's aliased frequency, and data showing values higher than the thresholds given in Table 4 are dismissed from the assimilation dataset). In the sub-Antarctic region, the seasonal presence of sea ice limits the availability of usable E1/E2/EN altimetry data and will be rejected by the numerical Rayleigh criterion at harmonic analysis step. The TP/J1/J2 along-track data, shown in Figure 10, clearly enable us to densify the assimilation dataset on the shelves and near the coasts, where the amplitude of the tide and the errors of the model are the largest and tidal wavelength the shortest.

The TG dataset for the data assimilation process was obtained from several tidal data sets: the WOCE/GLOSS coastal database, open ocean BPR database provided by R. Ray (and used as validation dataset in Stammer et al, 2014), an open ocean BPR database in Antarctica compiled by LEGOS, an Arctic database from Kowalik and Proshutinsky (1994), the International Hydrographic Office (IHO) data set, the research of Gjevik and Straume (1994), and some additional data compiled by LEGOS, and four TG stations of R. Ray's shelf database, located North of Florida. Any inevitable redundancy due to neighbouring observations were identified and the consistency between the neighbouring stations was systematically verified. In total, the TG database contains 600 stations (Figure 11) with a relatively homogeneous geographical distribution. For efficiency reasons, and because the TG time series needed to compute numerical error estimates were not available for the full dataset, the TG data error estimates were fixed arbitrarily and empirically to 3 mm for the deep ocean stations and to 1 cm for the shelf and coastal stations. The idea was to limit the constraints on the model at the tide gauge stations on the shelf and close to the coast in order to avoid drawing the solution to fit some very local tide features observed by the coastal stations that may be inconsistent with the larger scale tidal patterns that can be accurately solved at the resolution and/or bathymetry of the model.

Finally, iterative data assimilation experiments proved the need for some additional observations in particular regions, where neither TG nor standard altimetry data were available. Dedicated coastal altimetry-derived tidal observations provided by the French Observation Service dedicated to **satellite altimetry** studies (Centre of Topography of the Oceans and the Hydrosphere, denoted CTOH, ctoh.legos.obs-mip.fr), based on the harmonic analysis of TP/J1/J2 GDRs, were used to better constrain the model in these specific cases: 1 point North of Tierra del Fuego, 1 point in the Pamlico Bay (North Carolina) and 2 points between the Southern islands of Japan. The total assimilation dataset contains 12 622 observations for the M2 tidal component and slightly less for the other components, depending on the error estimates associated with the tidal constituents or because of constituent-specific aliasing issues.

It should also be noticed that the M4 tidal component received a special treatment for the construction of the assimilation dataset. Indeed, the non-linear M4 tidal component mostly develops on the continental shelves. Because of its small amplitude in the open ocean, it is difficult to separate the M4 signal from the other ocean signals with similar space and temporal scales, and the noise-to-signal ratio in the M4 analysis is much too large to provide appropriate data to the assimilation. Consequently, only shelves and coastal seas data have been kept in the M4 assimilation dataset. The complete M4 assimilation dataset contains altimetry crossover points from TP/J1/J2, TPN/J1N and E1/E2/EN, along track data from TP/J1/J2, the four CTOH TP/J1/J2 coastal points previously mentioned and only one TG, the Avonmouth station, in the Bristol Channel(UK).

**5 Atlas assessment and validation**

The validation of the FES2014 tidal atlas is based on a frequency-domain (harmonic) validation of the ocean tide components plus a temporal validation of the total geocentric tide components (i.e. ocean tide plus loading tide). The FES2014b performance is compared to state-of-the-art global tidal models available at the time of the study, namely GOT4v8/GOT4v10, DTU10, TPXO9v2, EOT11A and FES2012 (please note that FES2014c and FES2014b have identical main long period, diurnal, semi-diurnal and sub-harmonics solutions, and the FES2014c long period extension is identical to the one implicitly made inside the prediction software, so the following validations will mention FES2014b only and will hold for FES2014c as well). The FES2012 and

FES2014a atlases have been included in performance inter-comparison assessments to demonstrate the beneficial impact of the following evolutions: FES2012/FES2014a differences mostly illustrate the improvement coming from the significantly higher accuracy of the FES2014a prior hydrodynamic solution in the assimilated solutions, while FES2014a/b differences mostly illustrate the improvement coming from the FES2014a-derived LSA forcing in the hydrodynamic model and in the assimilated altimetry data processing (through the tidal loading correction applied in GDRs).

The prediction code used for the time domain validations presented in 5.3 is the operational TP-Jason GDRs processing code. The tidal prediction software is available on a bitbucket deposit: https://bitbucket.org/cnes_aviso/fes/src/master/. It is appropriate not only to be used with FES atlases, but also with other popular atlases, such as GOT or TPXO releases. Without getting into deep details, the tidal prediction includes all tidal constituents provided by an atlas, but it also uses inference technique to add some significant,

missing astronomical constituents when not given as a prediction input file (namely $\mu2$, $\nu2$, L2, T2, $\lambda2$, 2N2, $\varepsilon2$, $\eta2$, 2Q1, $\sigma1$, $\rho1$, M1-1, M1-2, $\chi1$, $\pi1$, $\varphi1$, $\theta1$, J1, OO1) and a quite comprehensive number of long period constituents (up to 106 from second degree terms and 17 from third degree terms of the gravitational tidal potential) are added using the tidal equilibrium approximation. The inference formulae and the extensive list of the tidal constituents that are computed using inference/equilibrium approximations are listed in the code.

Contrary to the tidal models inter-comparison exercise in Stammer et al. (2014), we made the choice of not restraining the tidal prediction to a common set of constituents when making comparisons with the GOT4v10 atlas, in order to take into account both the accuracy and the omission error of the models, thus keeping close to real life performance. However, the implementation, inside the prediction software, of the inference method to increase the prediction spectrum efficiently compensates for the impact of missing astronomical constituents in

the GOT4v10 atlas, so most of the differences in the actual prediction spectrum will be limited to the differences in the availability of compound tide and overtide constituents.

### 5.1 Description of FES2014 tidal spectrum

FES2014b is the only global tidal atlas that offers a comprehensive tidal spectrum of 34 tidal components, including linear components (K1, M2, N2, O1, P1, Q1, S1, S2, K2, 2N2, EPS2, J1, L2, T2, La2, Mu, Nu2, R2),

non-linear components (M3, M4, M6, M8, MKS2, MN4, MS4, N4, S4) and long-period components (MSf, Mf, Mm, MSqm, Mtm, Sa, Ssa). Late extension to eight additional equilibrium, mass conservative long-period tides, The Doodson numbers of which are 0555555 (M0), 0565545, 0585545 (Sta), 0655556, 0656553, 0735555 (MSf), 0753555 and 0754556, has been recently made to FES2014b and, as previously mentioned, the extended atlas is denoted as FES2014c. The MSf tide was already present in FES2014b, but resulted from non-linear

dynamics (M2/S2 interaction) only because its astronomical potential was accidentally dismissed at production time. In the FES2014c atlas, the corresponding complementary equilibrium solution file is denoted MSf.FES2014c.LPequi_only.nc, and should be used in conjunction with the FES2014b solution file.

Despite providing the S1 tide, we discourage its use in tidal corrections when storm surges or DAC corrections that include diurnal atmospheric effects (i.e. not filtered out) are available. Actually, the astronomical part of S1

is rather negligible, and it is mostly forced by the atmospheric surface pressure, which shows significant seasonal and inter-annual variability. So any harmonic S1 solution will be the reflection of the mean of the S1 tide over a given time period (of simulation and/or data assimilation), and would ideally need to be completed with a consistent residual S1 DAC correction to account for its intrinsic variability, which would be technically quite

tedious to perform in a fully consistent way. Up to now, the accuracy of the S1 DAC solution has been quite limited by the temporal resolution of the available atmospheric pressure forcing products. At present, the operational processing of GDRs data is based on a DAC filtered from which the mean S1 atmospheric components have been filtered out, and the S1 tide (both atmospherically and gravitationally forced) is then removed by the S1 tidal solution in the tidal prediction. However, because of the recent improvements in the

atmospheric products (notably in their time sampling), the FES group is in favour to revise the present operational data processing paradigm by leaving S1 correction to be accounted for in the high frequency storm surge correction (DAC) instead of in the tidal correction for the next generation of altimetry products.

FES2014 contains either free hydrodynamic solutions or data assimilation results. The choice of the tidal components that benefited from data assimilation was made upon two criteria. First, the accuracy of the non-

assimilated tidal component with regards to its amplitude: the smallest tidal components were not assimilated. Second, the capability to separate the tidal components in the altimetry and TG observations, in terms of signal to noise ratio: the long-period tidal components were not assimilated. Finally, the following 15 tidal components benefited from data assimilation: K1, M2, N2, O1, P1, Q1, S2, K2, 2N2, EPS2, L2, La2, Mu2, Nu2 and M4. Most of the diurnal, semi-diurnal and non-linear tides were computed using the frequency-domain solver,

especially the assimilated ones (for ensemble computational cost reasons). The smaller linear and non-linear tidal constituents (not targeted by the data assimilation) J1, M3, M8, MKS2, N4, R2, S1, S4 and T2 were computed in time-stepping simulations, with atmospheric forcing (ERA-INTERIM) in addition to the usual tidal potential forcing. As earlier mentioned, the correction of S1 tide must be consistent with DAC correction content, and this is presently the case in the operational altimetric data processing. The addition of the atmospheric forcing not

only provides an S1 solution, but also guarantees a more accurate S2 tide representation and consequently a more accurate modelling of non-linear processes. This leads to a modelling strategy dilemma, as the use of the sufficient high-frequency atmospheric forcing (1-hour sampling) also raises a potential risk of partially duplicated correction with DAC, especially for the R2, T2 and S4 constituents, which should ideally be removed from the DAC correction. Let us also mention the K2 and M2 tides that have a small but significant atmospheric

contribution, which is already taken into account in the tidal solutions through the data assimilation. Definitely, the possible overlapping between tidal and DAC corrections (and dynamical coupling) is a serious issue that should be addressed in the future altimetric data processing. No admittance relationship was used for these minor waves. The long-period components (Mf, Mm, Mtm, MSqm, MSf, Sa, Ssa) were computed in time-stepping mode without atmospheric forcing.

A major, novel interest of the FES2014 tidal atlas is the availability of many non-linear tidal constituents. These components are generally not provided by other models although their amplitudes can reach several centimetres in shallow seas and even 1 cm in the deep ocean in the case of the M4 wave. The FES2014 atlas is originally designed for the tidal de-aliasing correction of the altimetry sea surface height observations, for which the mission accuracy requirements are set to 2 cm in the open ocean, so each (accurate) contribution to the tidal

spectrum is of importance. Another asset of the FES2014 atlas is the supplying of six long-period tidal components (Mf, Mm, Mtm, MSqm, Sa and Ssa; see previous comment about MSf) computed from the dynamical model forced with gravitational forces. These long-period components are generally approximated by the equilibrium solution in the other global ocean tidal models. At least for the constituents of period shorter than one month, the overall ocean (dynamical) tide shows significant differences with equilibrium approximations. In

addition, these dynamical solutions can show regional, fully unbalanced, enhancement due to topography trapped waves (for example in the Southeast Pacific). To compute the total geocentric tide, needed for altimetry observations correction, the FES2014a loading tide must be added to the FES2014b ocean tide, both being consistent as the FES2014a loading tide was removed from the altimetry data used in data assimilation step (cf section 2.6).

**5.2 Validation in the frequency domain**

The validation in the frequency domain (i.e. of constituent harmonic constants) enables to easily identify and locate potential deficiencies in tidal atlases. The performance of the tidal model can be quite different from one region to another, but also from one tidal component to another. As for the hydrodynamic simulations, the optimal tidal atlas (i.e. with data assimilation) has been validated by computing the vector differences between the observations (altimetry and TGs) for each tidal component. Figure 12 shows the vector differences between

the TG databases provided by R. Ray (and used as validation databases in Stammer et al, 2014) and the most recent global tidal models, for four main tidal components (M2, K1, S2 and O1). Here, it must be reminded that most of the deep ocean TG database was included in the assimilation dataset for FES2014b. As a consequence, it is expected that the vector difference between this database and the FES2014b/FES2014c tidal model should be

very low (still, it indicates that this dataset was found to be self-consistent in the data assimilation process). To a lesser extent, shelf and coastal datasets inevitably contain some assimilated data. Actually, the amount of quality TG data is not large enough to allow for a distinct data assimilation and well-balanced validation datasets. Moreover, if a quality-checked, validation dataset's data demonstrates some divergence with the atlas solution, it would be then extremely useful to include it in the final dataset for assimilation. In addition, because of the

ensemble/data representers approach, the assimilation solution will not easily fit the TG data if not consistent with model error covariance and other data, including altimetry data. Internally, we also made consistency checks by assimilating altimetry data only, then compared solutions with TG data and reached very close numbers. So there is a favourable but moderate bias in terms of accuracy when comparing our final solution with the validation dataset. Still, the comparison to the other databases (shelf and coastal) shows the overall excellent

performance of the FES2014b tidal atlas, whatever the considered tidal component. This highlights the rather uniform accuracy of the FES2014b atlas, compared to some other competing atlases that sometimes show uneven accuracy estimates, also strongly depending on the tidal constituent.

In this validation against TG data, FES2014b and TPXO9v2 (recently released, April 2020, Egbert and Erofeeva, 2002) show the best agreement with data. The TPXO9-atlas is a 1/30 degree resolution fully global solution,

obtained by combining the 1/6 degree base global solution TPXO9-atlas and thirty 1/30 degree resolution local solutions for all coastal areas. To some extent, the regional patches in the TPXO9v2 reproduce the (seamless) FES unstructured grid flexible resolution, and therefore explain the similarities in terms of performance in shelf and coastal seas. In these comparisons, we have chosen to display the latest release of TPXO atlases, and not the release available at FES2014 production time (TPXO8), which proved to be significantly less accurate than

FES2014 in a similar diagnostic. The gap in accuracy is much reduced with the TPXO9v2, which has probably taken advantage of longer time series for altimetry data, and possibly improved bathymetry for its prior simulations (or any other improvements in the regional hydrodynamic models configurations).

## 5.3 Variance reduction in satellite altimetry observations and in tidal gauges

A complementary validation consists in estimating the variance reduction obtained for altimeter observations or tidal gauges measurements, when using the FES2014b tidal atlas as a correction for the barotropic tide sea surface height and comparing with other tidal atlases (including FES2012 and FES2014a atlases to demonstrate the gradual progression, coming from improved prior solutions and loading, toward the final FES2014 atlas). This temporal approach allows taking into account the solution error as well as the omission error for the missing tidal constituents. Notice that, while the geocentric tide solution (i.e. ocean tide solution plus earth tide and ocean tidal loading solution) is used for correcting altimeter data, only the oceanic solution is used for tidal gauge corrections as gauges follow the tidally induced bottom motion. Because of data assimilation, errors in the LSA atlas used in altimetry assimilation data corrections will reflect in the ocean assimilated solution (and will participate to the misfits with TG data), while they will cancel if performing altimetry validation data correction with the same LSA atlas as used in data assimilation processing (as recommended for GDRs operational processing).

Figure 13 shows the maps of variance reduction at tidal gauge sites from the GLOSS network, when using the new FES2014b tidal model and compared to the GOT4v10 solution; although some of these tidal gauges have been assimilated within the FES2014b model, this diagnostic still permits to give information about the quality of the solution in coastal regions, particularly on the French coasts where no data has been assimilated. Results indicate a significant variance reduction when using the new FES2014b solution compared to the GOT model for nearly all sites. A few tidal gauge sites show an increased variance but these TGs are located in very complex or enclosed regions and are thus not representative of the coastal ocean variability observable with a global ocean tide model. A complementary validation was performed using some independent TG information along the Canadian Atlantic coasts (cf. Figure 14); it shows an important mean variance reduction of -17 cm² for the 10 TG when using the FES2014a solution instead of the GOT4v10 one.

The impact of using the FES2014 tidal corrections in the global ocean is estimated by computing the altimeter SSH differences between ascending and descending tracks at crossovers, using either the new correction or a reference one. Crossover points with time lags shorter than 10 days within one cycle are selected in order to minimize the contribution of the ocean variability at each crossover location. This diagnostic allows an accurate estimation of the impact of the tide correction on the high-frequency part of the altimeter SSH. This diagnostic gives information on the temporal variance of the SSH differences in the small boxes of 4°x4° used for the computation. The analysis has been performed using several missions and many different global tidal atlases, but we will only present the results for Jason and AltiKa missions: the Jason is the reference and very accurate mission and AltiKa is independent of all the models tested. Figure 15 and Figure 16 show the maps of SSH variance differences when comparing FES2014b with GOT4v10 and FES2012 tidal models respectively. Results demonstrate a very good performances of the FES2014b tidal solution compared to the other models, with a strong variance reduction noted in all shallow water regions (more than 10 cm² when comparing to both FES2012 and GOT4v10) and also in some deep ocean areas. Statistics for AltiKa are a bit noisier compared to Jason ones due to the shorter time series available, but they give valuable information for high latitudes: FES2014a in particular shows a strong improvement compared to FES2012 in most of the Arctic Ocean region, except in the Laptev and Kara Seas. It is difficult to estimate how significant is this local deterioration in

variance reduction, just it must be remembered that the Altika mission suffers from shorter (in duration) and fewer (in repetitivity) exact repeat observations compared to Jason time series. In consequence, the variance reduction diagnostic is therefore made on a less significant statistical basis, and the overall variance reduction map shows many local, "noisy" outliers (compared to the surrounding general tendency). In addition, seasonal ice in the Arctic Sea is furthermore diminishing the number of available valid observations, hence potentially

increasing the uncertainty on the variance reduction estimates. Some independent validation was performed to compare FES2012 and FES2014 (see R. Ray et al., 2019) and showed the clear improvement in FES2014, except for M2 in the Kara and Laptev Sea, where comparison with Altika measurements shows a slightly weaker performance of FES2014. Unfortunately, the lack of a more comprehensive TG dataset makes any stronger conclusions quite difficult to draw. FES2014b also strongly reduces the variance compared to GOT4v10 in

northern high latitudes, except for a slight rise of variance noted north of Baffin Bay when comparing to the GOT model.

To pursue the analysis further to the coast, we consider along-track sea level anomalies (denoted SLA) calculated from 1 Hz altimetric measurements. Although high-frequency signals are aliased in the lower-frequency band following the Nyquist theory as appropriate to each altimeter sampling, SLA time series contain

the entire ocean variability spectrum. Figure 17 shows the difference of SLA variance when using the FES2014a tide model instead of FES2012 (resp. GOT4v10) model, for the AltiKa mission and as a function of distance to the coast. This diagnostic shows the very strong improvement of the new tidal solution within the first 60 km from the coast compared to that for the global ocean, with a mean variance reduction reaching more than 20 cm² within the first 30 km from the coast when comparing FES2014a and GOT models. Surprisingly, FES2014a

improvement versus FES2012 reaches its maximum at some distance from the coast (about 15km). The near-shore performance, both in FES2012 and FES2014a, is probably limited by local bathymetry accuracy and coastal detail discretization, and ensemble/representers being less able to properly describe local error statistics, so data assimilation improvements in FES2014a propagate only partially toward near-shore zones.

**6 FES2014 atlas additional derived products**

The primary objective of the FES2014 project is to improve the tidal elevation prediction used in satellite altimetry data de-aliasing. However, additional tidal estimates are available from the modeling and data assimilation outputs. Particularly, new global tidal current maps, estimations of tidal energy budgets in the global ocean and loading and self-attraction components are presented here.

**6.1 Tidal currents**

Tidal currents have been estimated on the finite element mesh with the element-wise discontinuous non-conforming P1 discretization (one estimate in the middle of each element, estimated separately for each triangle). The FES2014b tidal currents benefited from the data assimilation of the tidal elevations data through the dynamical correlation computed from the assimilation ensemble. The tidal currents are provided on a 1/16° grid like the elevations.

Contrary to sea surface elevation where tides are the major contributor to variability in most ocean regions, the validation of tidal currents is quite challenging as it requires long-enough (several months to years) accurate current meter time series to accurately extract current harmonic constants from the tidal harmonic analysis. In

addition, to be useful for consistent comparisons, the current meter gauges must be moored in sites that are representative of the surrounding tidal dynamics. The main resulting constraint is to discard areas showing pronounced uneven bottom topography, as currents are highly sensitive to local bathymetry which cannot be captured properly by the model grid resolution. All these constraints (together with the fact that the access to the data is often restricted) imply that very few observations are finally available for the tidal velocity validation.

Luckily, for more than 10 years Australia has been maintaining a network of 48 ADCP instruments all around the continent, principally through its government-supported Integrated Marine Observing System (IMOS). The Australian continental shelf has a wide range of tidal regimes ranging from macro-tidal to micro-tidal, thus providing ideal conditions to thoroughly test a model. The ADCP observations are accessible via the IMOS portal (https://imos.org.au). An additional issue is that FES2014b tidal currents are representative of depth-averaged currents (as they are based on the shallow water 2D equations), and vertical profiles of tidal currents will potentially contain some baroclinic tidal current signal, those currents being possibly one order of magnitude larger in the vicinity of intense internal tide generation sites. In this case, predominance of internal tides currents makes the barotropic current retrieval quite difficult, and would require full water-column auxiliary data (potential density at least) to be conducted precisely by using the vertical mode decomposition approach (Cao et al., 2015; Nugroho, 2017). In this paper, the ADCP time series were specifically processed by CSIRO with regard to the computation of the depth-averaged currents which comparable to currents computed with a barotropic, shallow-water model. Then, a harmonic analysis was performed in each current direction separately (U eastward and V northward) for five main tidal components (M2, K1, S2, O1, N2).

These in situ tidal harmonic constituents are compared to the FES2014b model tidal currents in terms of vector differences and tidal current ellipse characteristic differences. The latter gives a synthetic description of the tidal current for a given tidal component. The length of the semi-major axis gives the maximum amplitude of the tidal current and the orientation of the ellipse gives the angle between the main current direction and the eastward direction. The parameters of the ellipse (orientation and lengths of the minor and major axes) are computed from the tidal velocity harmonic constituents estimated in both directions (eastward and northward). The tidal current ellipses computed from the current meter observations (in red) and from the FES2014 model (in blue) are displayed for the M2 and K1 tidal components in Figure 18 and Figure 19 respectively. The green dots show the positions of the current meter moorings. For some moorings, the ellipses are not visible on the figures due to the very low amplitudes of the tidal currents in these micro-tidal sites. Overall, there is a very good agreement between the FES2014b model and the observations, at most of the macro-tidal sites. At some specific moorings (Darwin station and some stations inside the Great Barrier Reef), some large discrepancies are observed, that are due to the fact that these stations are very close to the coast, in very shallow areas where the resolution and/or bathymetry of the FES2014 global tidal mesh is too coarse to accurately solve the currents. At some other stations (Coffs Harbour mooring), located in the open ocean, the model shows very strong unrealistic eastward components. This is due to a lack of resolution in the model grid, in this case at the shelf break (the Coffs Harbour station is located close to a steep bathymetry slope). This is a well-known numerical artefact of the discontinuous numerical discretization of the tidal currents appearing where the model grid has accidentally insufficient resolution over steep bottom topography, despite all the care taken in the mesh construction (in which the built-in constraint for slope in topography imposes resolution to be proportional to H/grad(H)). After appropriate verification, it appears that this issue occurs only in a few locations of the FES2014 mesh. The

validation of the FES2014b tidal currents not only depicts the overall fit with observations, but can also suggest a careful additional screening for future FES grid design, complementary to diagnostics made from the tidal elevation validation.

### 6.2 Energy budget

Barotropic tides energy budget is a valuable diagnostic to examine the model performance and accuracy, and to understand more precisely how tidal dynamics works as an energy generation, transport and dissipation mechanism. It can also be a proxy for the interactions of ocean tides with ocean circulation and stratification (bottom friction and internal tide drag rates of work) and be a feeding parameter to general ocean circulation models that do not solve explicitly for the tides and need to parameterize their effects, mostly on mixing. Energy budget has been estimated both from the prior, dynamically balanced tidal solutions (thanks to their unprecedented accuracy), and from the data assimilation solutions. The latter are of course more accurate in elevation and currents, but are not perfectly balanced (dynamically consistent). However the limited action of the data assimilation due to the prior solutions accuracy, and the (somehow) dynamical properties of the model error covariance, computed from the ensembles' dynamical members, allow for meaningful energy budget estimates.

Among other possible energy estimates (bottom friction, potential forces, etc... rate of work), the energy conversion rate from barotropic tides toward baroclinic internal tides (Figure 20) is very valuable diagnostic to identify and quantify internal tides generation. For instance, it can be used to provide additional vertical diffusion information in ocean circulation models where tides are not explicitly resolved. Nowadays, some global ocean circulation models explicitly resolve tides (Maraldi et al., 2012; Kodaira et al., 2016; Arbic et al.,2018), and can produce a similar conversion rate estimate, but with a lower accuracy in terms of barotropic tide solutions. In FES2014 estimates, uncertainties come from the parameterization used in the dynamical model. In other words, our energy diagnostics bring a complementary, independent information to a still evolving and uncertain knowledge.

### 6.3 Loading/self-attraction atlases

New maps of the loading and self-attraction effects have been estimated taking into account the preliminary FES2014a tidal elevations. In the pre-FES2014 era, LSA atlases were computed from the projection of the native finite element tidal elevation upon a high resolution regular grid, either using spherical harmonics/Love numbers approach or an equivalent Green's function convolution. However, T-UGOm tidal models needs the gradient of LSA, obtained first through a projection back to finite element grid, followed by a numerical derivation. The two-ways projection can trigger some undesirable numerical effects, and a new software has been developed to directly derive the LSA atlases on the finite element grid, using Green's functions convolution (Lyard et al., 2020, in preparation). Figure 21 shows the amplitude of the resulting M2 LSA computed from the FES2014a atlas, and the differences with the GOT4v8ac loading effects.

As the computation of a tide model is an iterative process, these FES2014a LSA maps have been used to compute the final tidal model versions FES2014b/c, showing an improvement of the global performances in terms of tidal correction as shown in Figure 22.

## 6.4 Lowest/Highest Astronomical Tides (LAT, HAT)

Lowest astronomical tides are commonly used in hydrographic services as the reference level for nautical charts and terrain models. It is also a valuable parameter in maritime engineering and risk assessment studies. The FES2014b LAT (and HAT, highest astronomical tides) chart has been computed from an twenty-years tidal prediction (to account for nodal fluctuation in tidal amplitudes) based on all available tidal constituents in the FES2014b atlas (Figure 23). Mean Lower Low Water tides (MLLW) and Mean Higher High Water tides (MHHW) tide levels (as used by NOAA and some others) could be obtained in a similar way, as well as some additional ancient hydrographic datum, as mentioned in Pugh and Woodworth (2014). FES2014b LAT is routinely used at LEGOS to convert bathymetry from hydrographic services into ocean mean-level bathymetry as needed in numerical ocean modelling, especially in coastal and near-shore configurations.

## 7 Conclusions

Despite the tremendous efforts devoted worldwide to improve tidal corrections for altimetry during the last two decades, we still face challenging issues in shelf and coastal seas, as well as in high latitude oceans, where the accuracy of tidal atlases remains too limited for precise altimetry data processing. Considering this matter, the FES2014 atlas can be considered as a very significant step forward, keeping close to other atlases in the deep ocean but showing a lot of improvements in shallow water seas, and some significant ones in the high latitude seas.

After competitive evaluation procedures (mostly based on variance reduction of altimetry time series when applying tidal corrections predicted from various tidal atlas candidates), it has been selected for the CNES/NASA/ESA/EUMETSAT operational and re-processing altimetry data de-aliasing correction, and more recently as the standard correction in ITRF2020 conventions by the International Earth Rotation and Reference Systems Service (IERS). Thanks to the (accidental) unusual delay between the FES2014 atlas release and this publication, the project team and the user community were able to accumulate extensive experience on FES2014 atlas performance in the tidal prediction/correction domain. Namely, besides space-borne applications, it is now widely and successfully used for regional modelling and in situ data processing applications, supporting our confidence in its remarkable accuracy. As a matter of fact, one can consider that, even five years after its release, FES2014 is well placed in the most useful global ocean tide atlases shortlist because of its extended tidal spectrum (34 constituents, among which 15 were optimally adjusted by data assimilation), its unprecedented accuracy in shelf and coastal seas and its detailed coastal grid.

The forthcoming Surface Water Ocean Topography (SWOT) altimetry mission will especially profit from these specific characteristics as it will offer coastal and near-shore nearly continuous, high resolution coverage (Morrow et al., 2019). However the FES project team is already making plans to design the next FES atlas, with emphasis on SWOT mission requirements and needs, which should be available within three years or so. Special attention will be paid to complex coastal regions (such as fjords, narrow channels and straits) with a 1km to 4km overall coastal resolution, and to polar seas where the tidal atlases accuracy is weakened by the difficulty of gathering quality tidal data for data assimilation and model validation. Thinking about more detailed shallow water observation de-tiding, the improvement of the hydrodynamic model will be one of the critical issues, and will need to aggregate further accurate world-wide bathymetry, which is a tedious and complicated task as the

attempt to access national hydrographic services data is often frustrating, especially when existing public data release is limited by non-scientific considerations. To some extent, we foresee that future atlas improvements and overall accuracy will be correlated with the level of cooperation of national services in this matter. New or improved space-borne bathymetry estimates (gravimetry/sea surface inversion, ICESAT-2 laser processing, surface wave wavelength inversion from optical data) might hopefully ease the issue, especially in remote or poorly accessible ocean parts, but open-minded international cooperation and open public data access remain a key factor for next generation tidal products. Meanwhile, we believe that the FES2014 tidal atlas will remain a useful base for tidal prediction and correction, in terms of surface elevation as well as tidal currents, in present or future altimetric or gravimetric satellite observations and in many maritime applications.

Among the new challenges that will be faced for the future is the question of ocean tides non-stationarity, as possibly induced by the time-varying ice-cover friction (Kowalik, 1981, Godin, 1986, Lyard, 1997), the barotropic to baroclinic tides energy conversion, storm surges or estuarine river discharge interactions with tides. As it is the case for existing global tidal atlases, the stationarity of barotropic tides is not questioned, neither in hydrodynamic modelling nor data processing. However it is quite a challenge for the future atlases in the context of SWOT, which will provide data in estuaries and deltas, and in very high latitude regions. Also tides/storm surge interactions need to be considered in altimetry high frequency corrections in shelf and coastal seas, but will require renewing the present correction paradigm (separate tides and storm surge corrections) in the operational data processing. Last but not least, the pole tides (Wahr, 1985; Carton and Wahr, 1986) have not been targeted in the FES tidal atlases as they are already corrected in GDRs by a specific correction based on Desai et al. (2015). However, further investigations are planned in the frame of the FES2022 project, especially for the non-equilibrium response of the ocean to the pole tide forcing. As a conclusion, we should insist on the fact that the accuracy of the tidal correction in altimetry products is far from being a resolved problem and further improvements will need to tackle details and issues that have been usually left aside in the past decades, and that of course the FES atlas production efforts will be continued.

**Acknowledgements**

The FES2014 project has been supported by the French space agency (CNES) under grants n°131678, n°160306 and grants n°161431 (general grant framework n°14026) and through the OSTST/TOSCA (INSU) scientific fundings. The authors warmly thank David Griffin from CSIRO for providing the depth-averaged ADCP current time series at the IMOS stations around Australia and for all the fruitful discussions about the comparison with the FES2014 tidal currents. They thank P. Gégout (GET, Toulouse) and J.P. Boy (IGSP, Strasbourg) for their assistance in developing the unstructured grid loading/self-attraction software, and D. Greenberg (BIO, Canada) for his long-lasting support in providing expertise and data for the Hudson Bay and Canadian Archipelago modelling. They also thank their colleagues of the OSTST tidal community for their long-lasting support and collaborations, with a special thought for R. Ray.

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

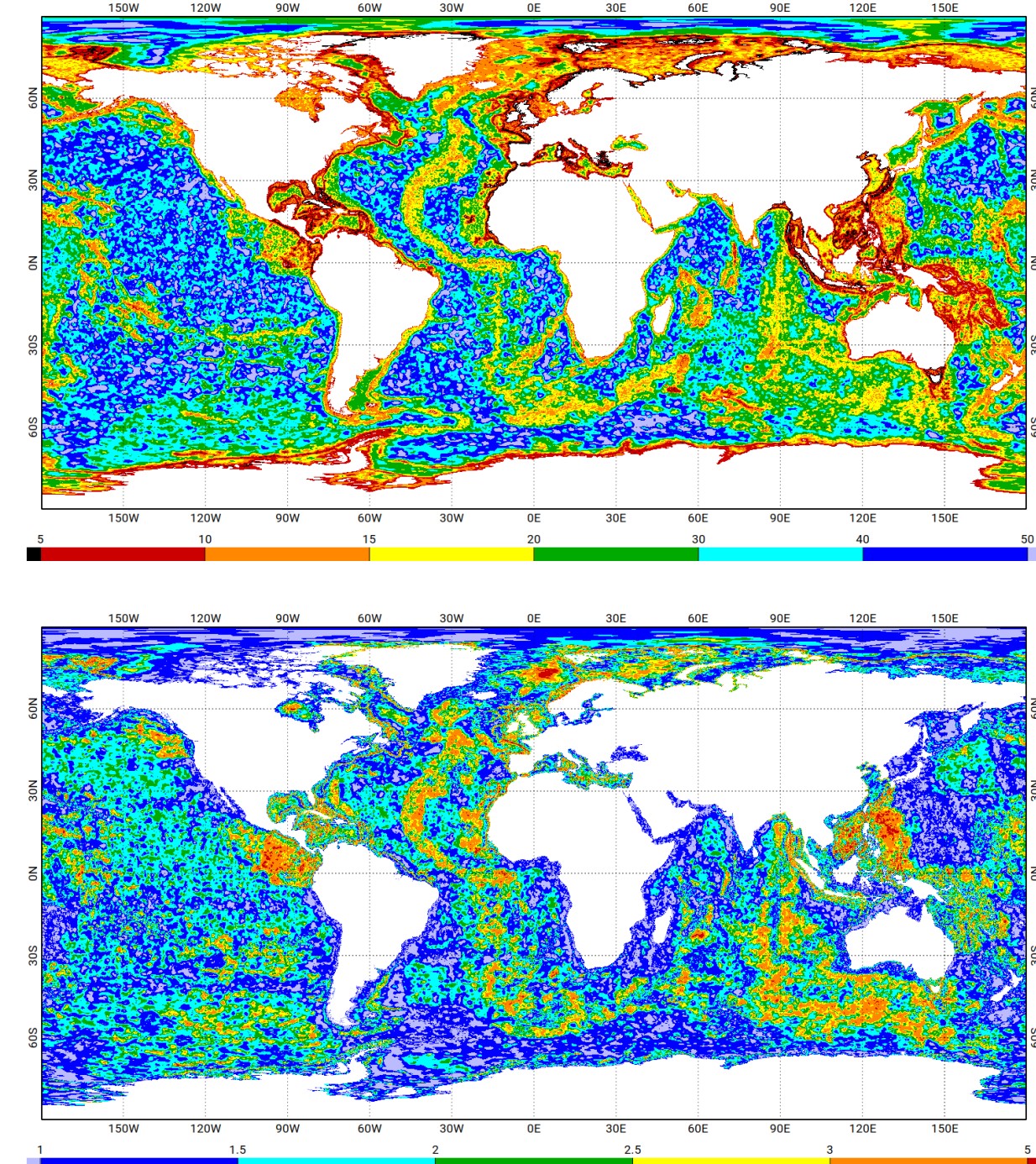

**Figure 1: Element-wise resolution (in km) of the FES2014 unstructured grids (upper panel) and the FES2014 resolution divided by FES2004 resolution ratio (lower panel). Resolution increase has been mostly focused on ocean ridges, shelves and shores (wherever reasonably accurate bathymetry was made available to the project). The numerical resolution of the frequency domain solutions is half the element-wise resolution due to second order basis functions (Lagrange P2).**

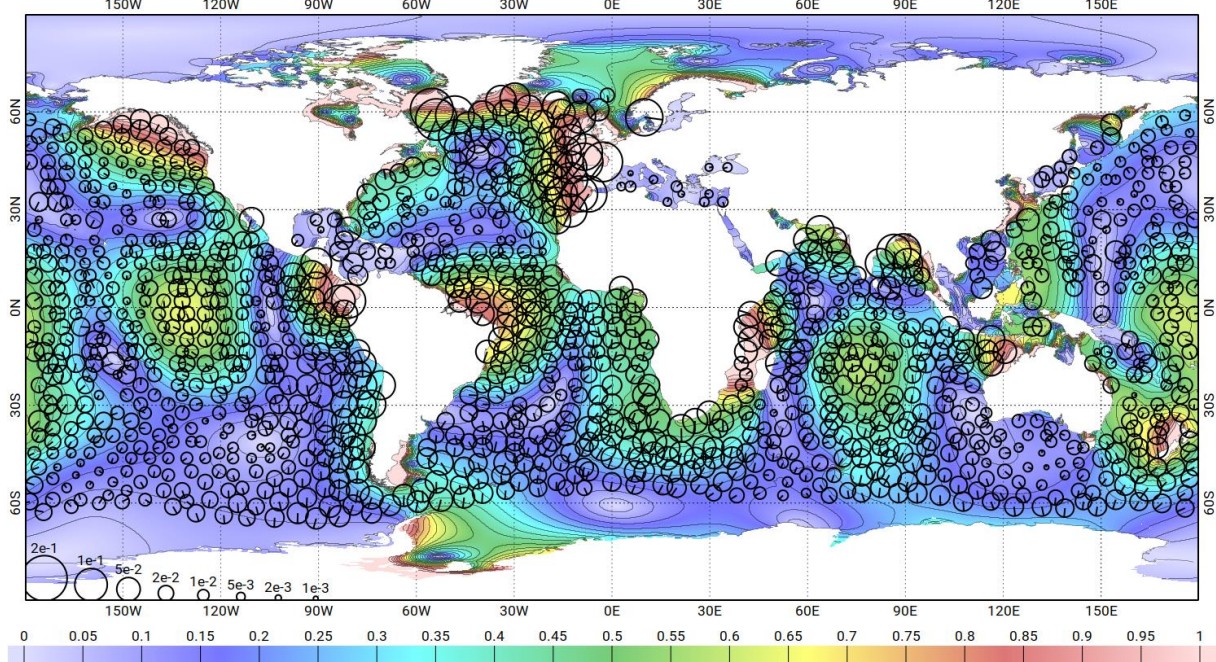

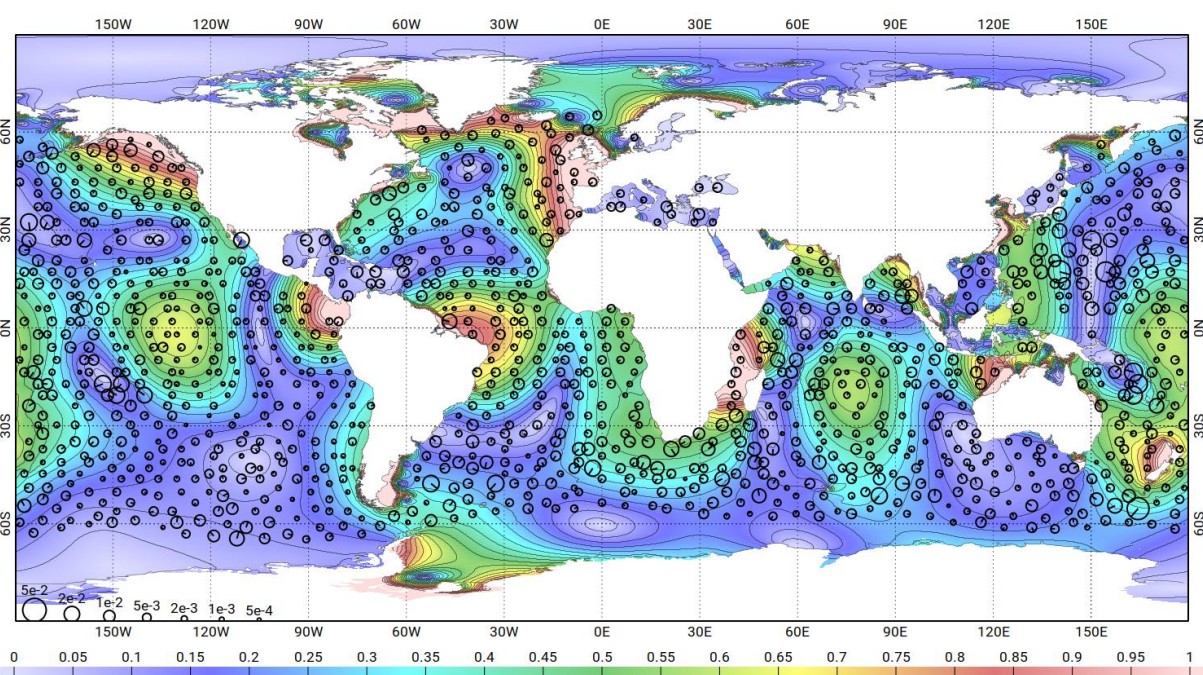

**Figure 2: Vector differences (black circles) between the purely hydrodynamic solutions of FES2012 (upper panel) and FES2014 (lower panel), and the deep TPJ1J2 altimeter crossover points, for the M2 tidal component. The accuracy improvement between the FES2012 and FES2014 prior solutions is a key ingredient in the accuracy improvement between the FES2012 and FES2014a/b/c assimilated solutions. The size of the black circles is proportional to the square root of the amplitude of the vector difference between the solutions and the observations (see bottom left normalized symbols, units in metres). The line inside circles shows the vector difference phase. The background colour shows the amplitude of the M2 tidal component from the model (in metres).**

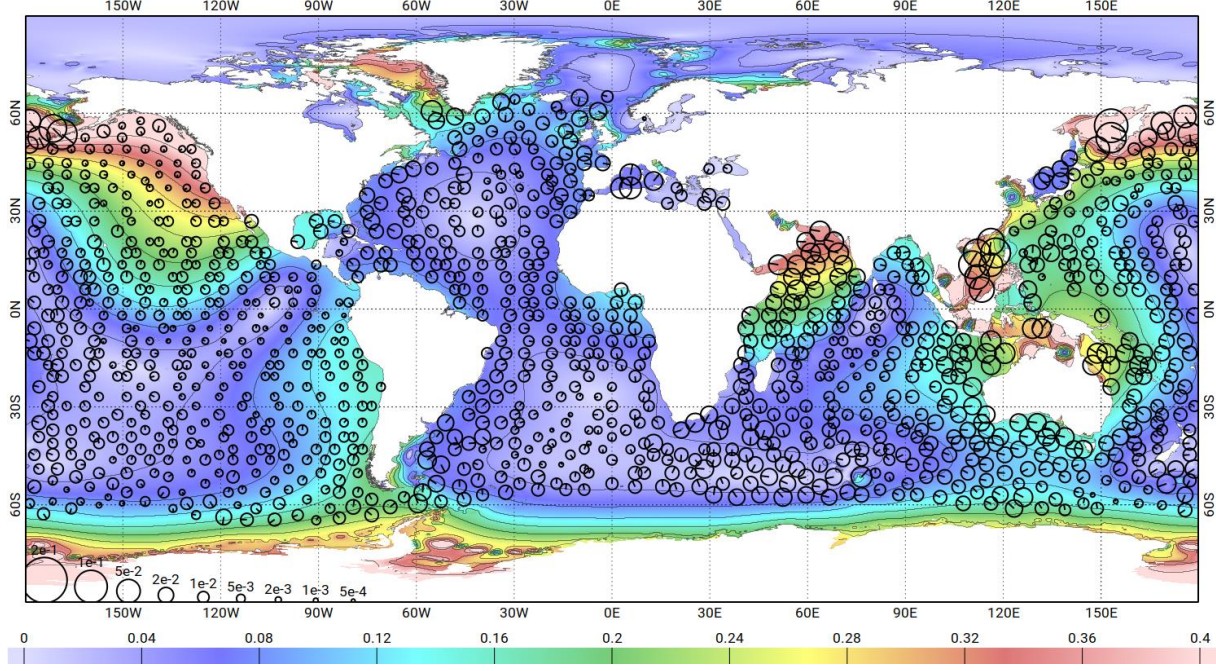

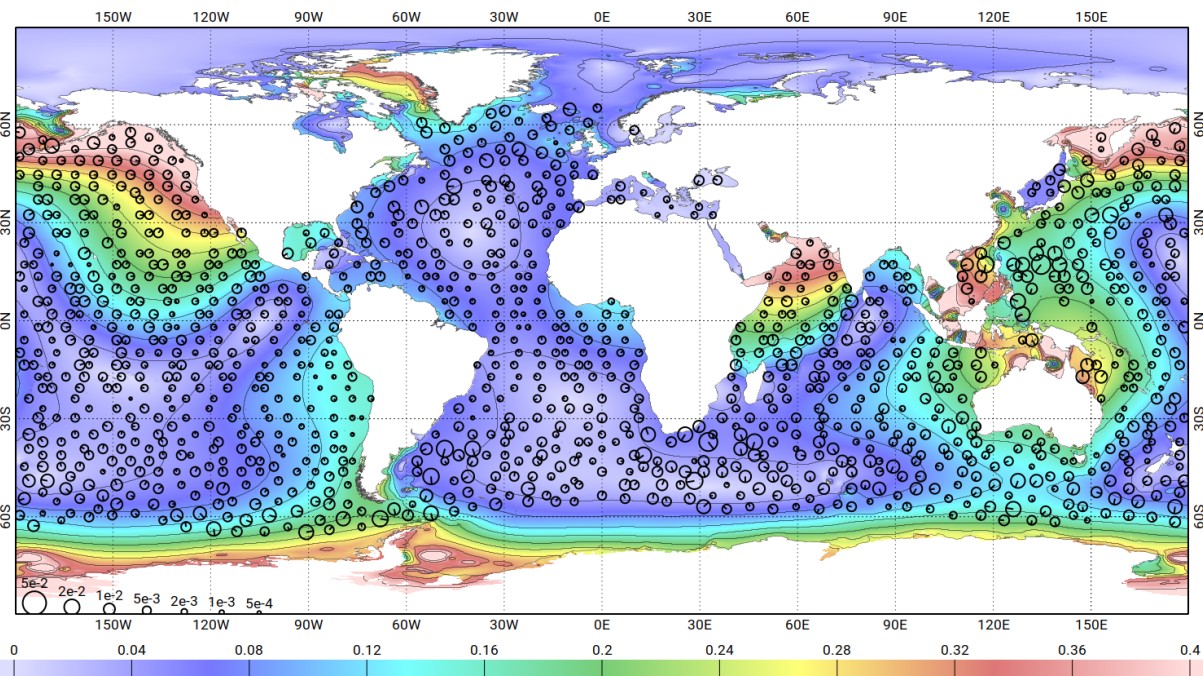

**Figure 3 : Vector differences (black circles) between the purely hydrodynamic solutions of FES2012 (upper panel) and FES2014 (lower panel), and the deep TPJ1J2 altimeter crossover points, for the K1 tidal component The accuracy improvement between the FES2012 and FES2014 prior solutions is a key ingredient in the accuracy improvement between the FES2012 and FES2014a/b/c assimilated solutions. The size of the black circles is proportional to the square root of the amplitude of the vector difference between the solutions and the observations (see bottom left normalized symbols, units in metres). The line inside circles shows the vector difference phase. The background colour shows the amplitude of the M2 tidal component from the model (in metres).**

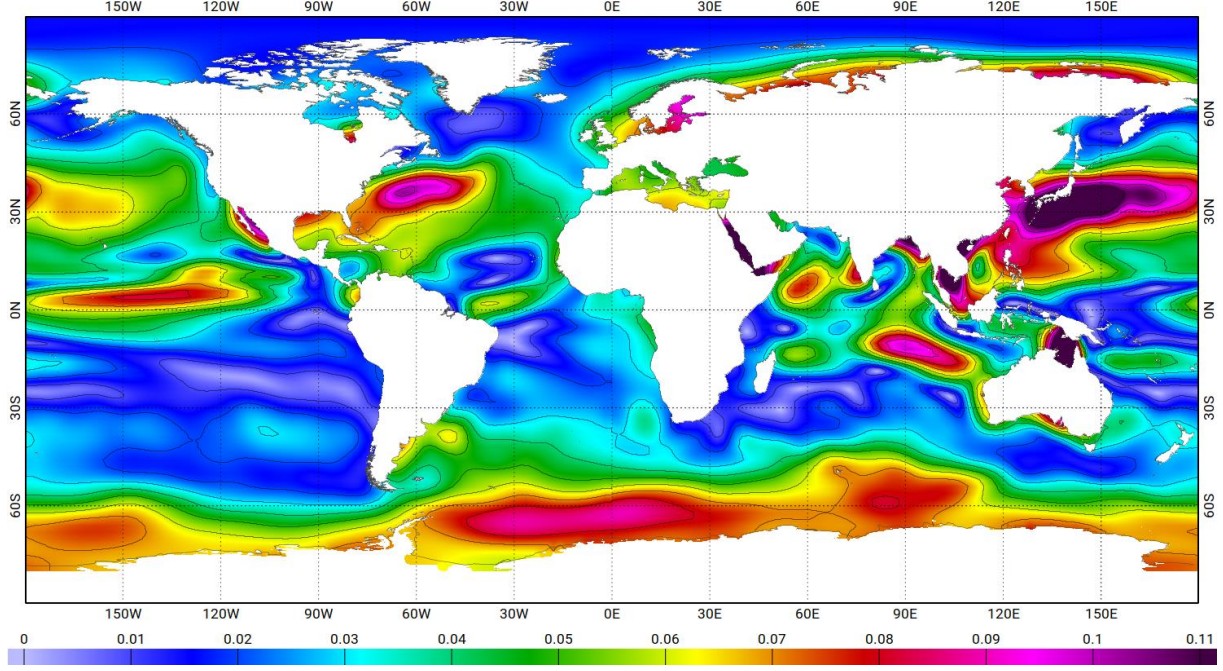

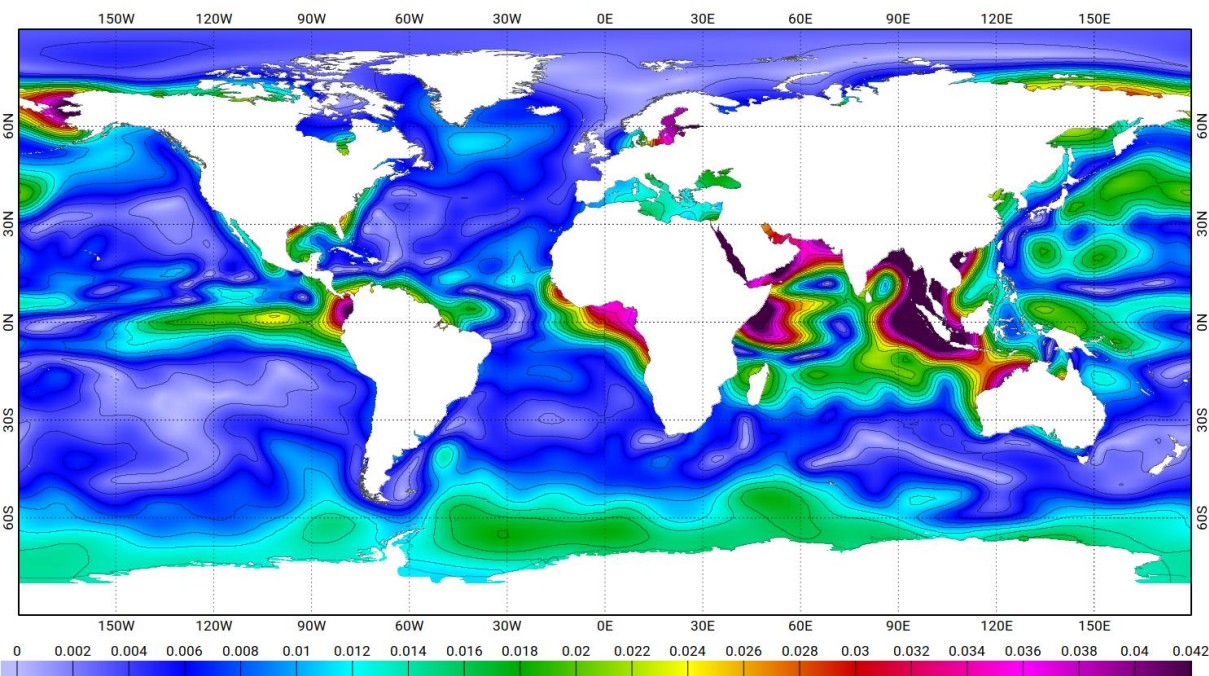

**Figure 4 : Maps of amplitude in metres of Sa (upper panel) and Ssa (lower panel) ocean signals estimated from GLORYS2v1 reanalysis. GLORYS2v1 products are free of atmospheric surface pressure effects (i.e. they are not taken into account in the NEMO model forcing and are corrected for in the assimilated SSH data). Consequently, they are comparable to IB-corrected sea level (at Sa and Ssa frequencies) in altimetry and tide gauge observations.**

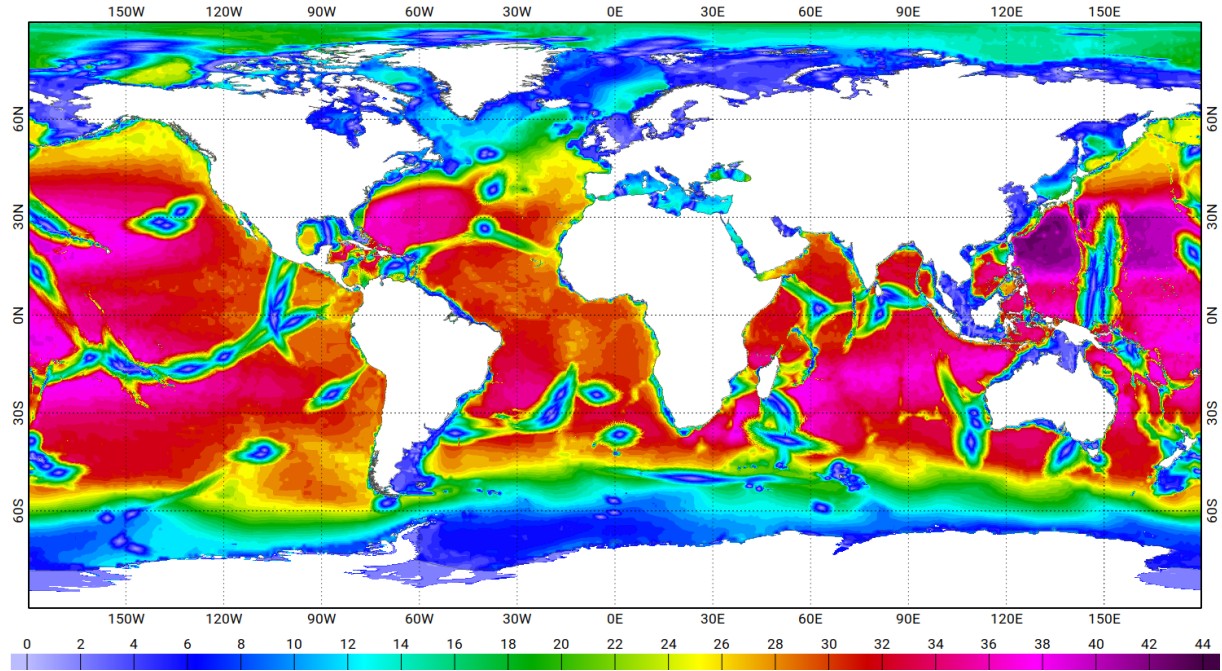

**Figure 5 : Along-track filtering wavelength used to remove internal tides surface signatures (expressed in number of 1Hz along-track points, to be multiplied by a factor six to retrieve the equivalent wavelength in kilometre)**

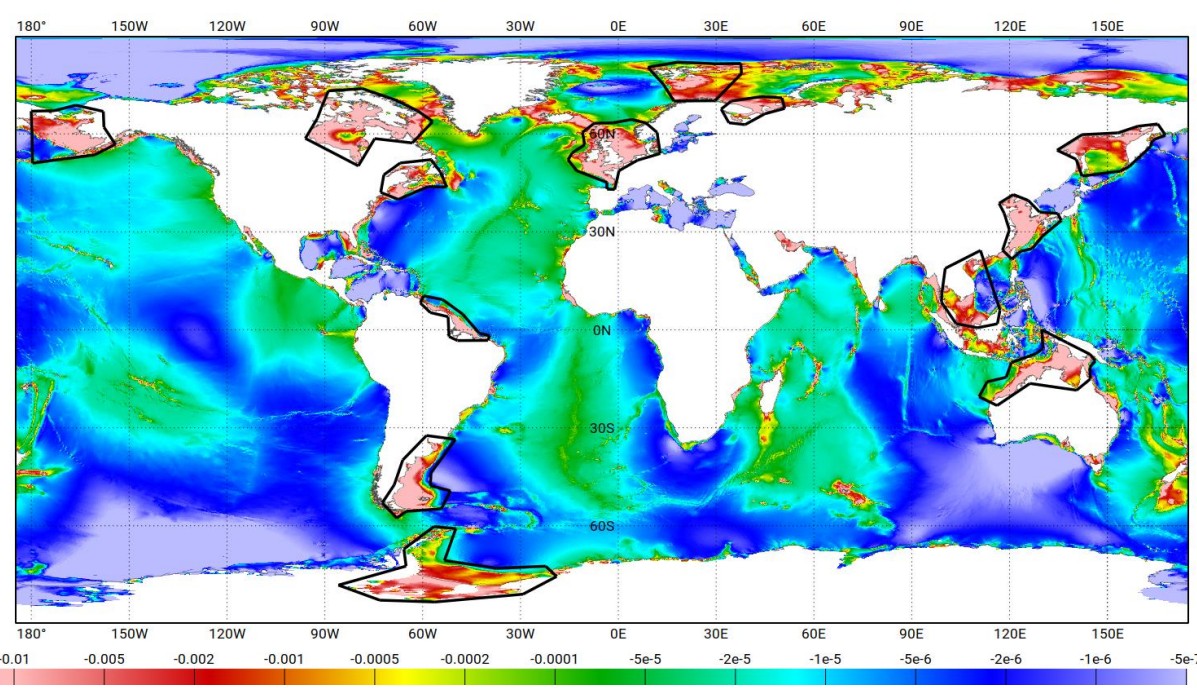

**Figure 6: Energy (W/m²) dissipated by bottom friction in the FES2014 hydrodynamic model, for the M2 wave, and polygons used for the perturbations of the bottom friction coefficient.**

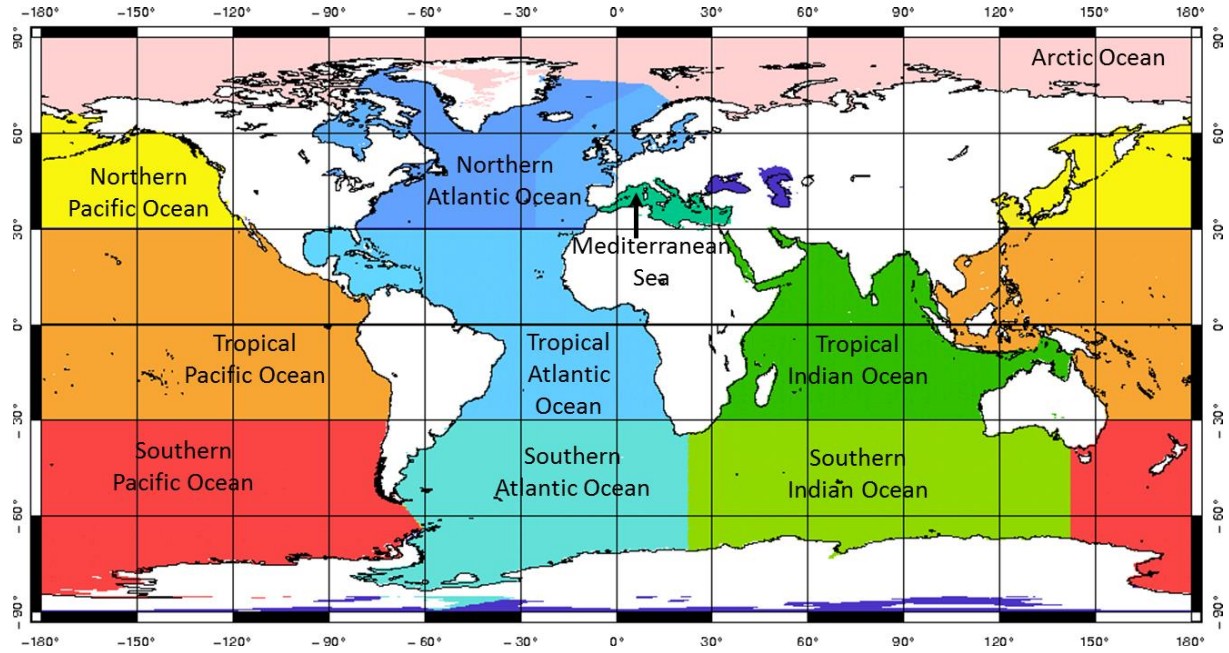

**Figure 7: Divisions used for the perturbations of the wave drag coefficient.**

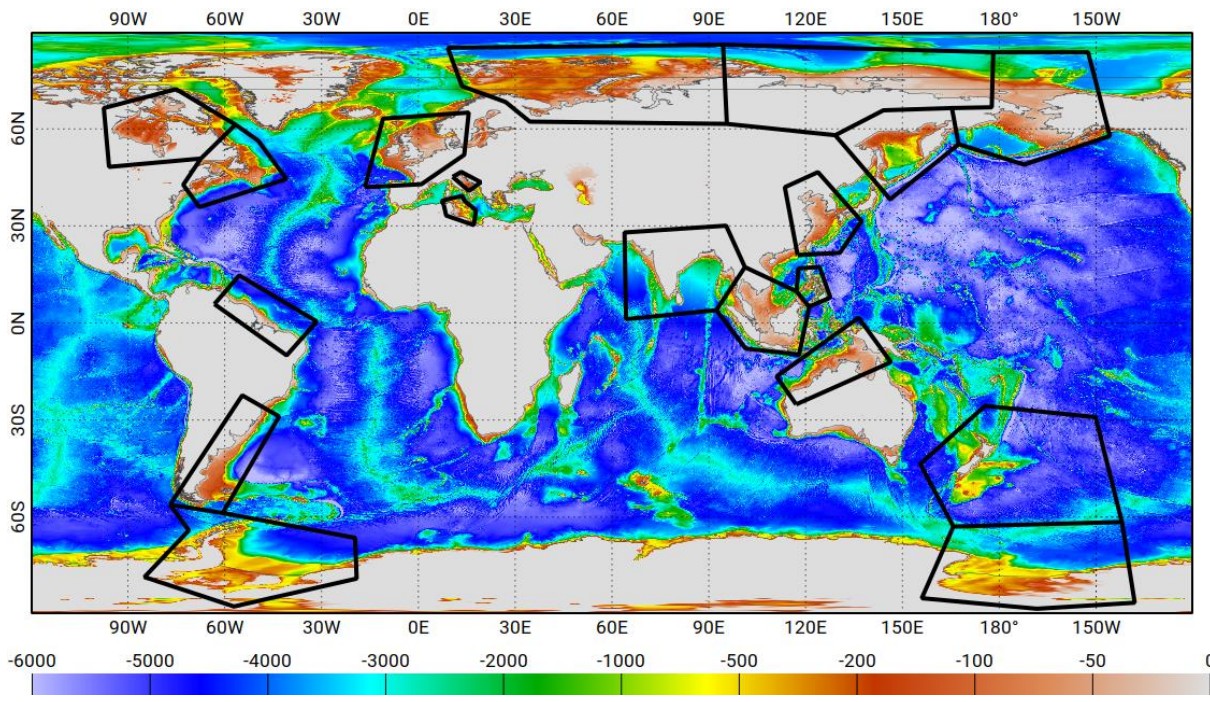

**Figure 8: Bathymetry (in metres) used as input in the FES2014 hydrodynamic simulation and polygons where the bathymetry perturbations were implemented for the bathymetry ensemble (Nota: the members related to the perturbation of bathymetry in the Weddell Sea have been discarded from the final data assimilation ensemble, see the data assimilation section for comments).**

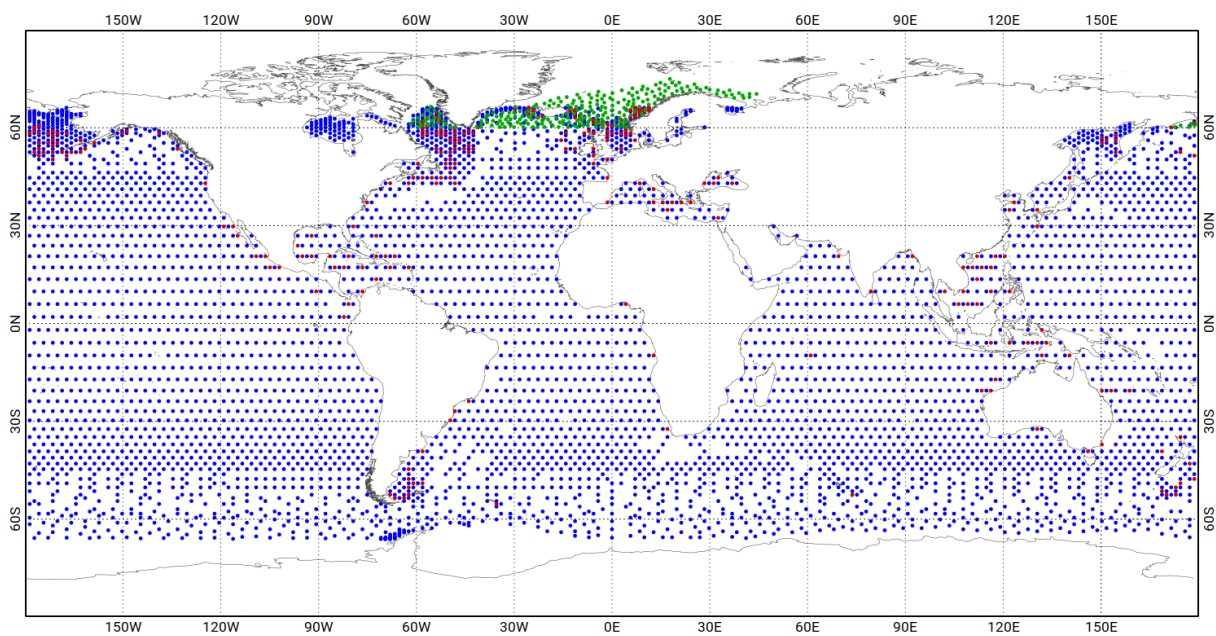

**Figure 9: Altimetry crossover points selected for the data assimilation: TPJ1J2 in blue, TPNJ1N in red, E1E2EN in green.**

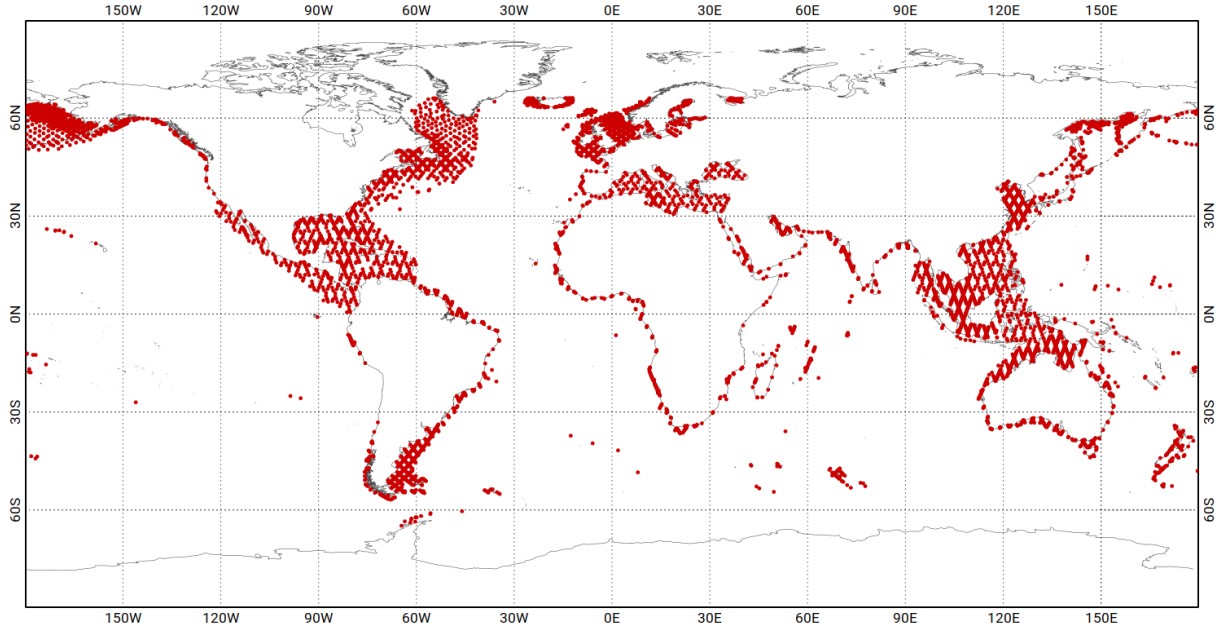

**Figure 10: TP/J1/J2 along-track data selected for the data assimilation.**

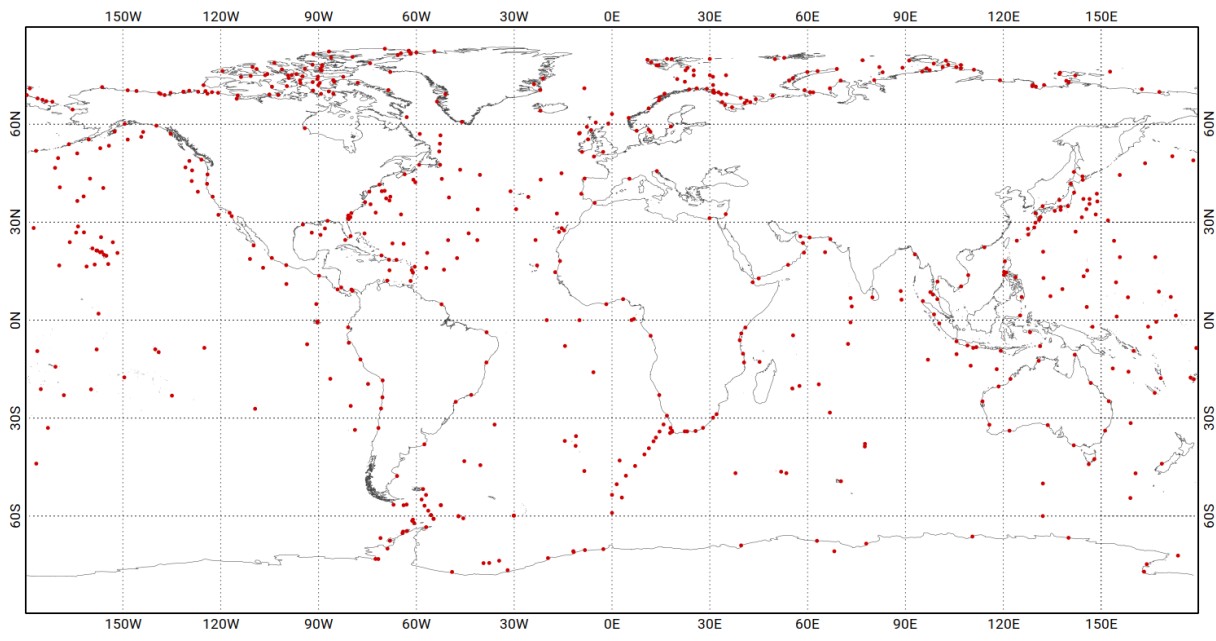

**Figure 11: the 600 TG stations selected for the data assimilation. It includes: 151 BPR deep ocean TG from R. Ray, 249 GLOSS coastal TG, 33 Antarctica BPR deep ocean TG, 164 TG from LEGOS composite database (including 15 TG in the Canadian Archipelago and 13 in the Baffin Sea), 4 TG from R. Ray shelf database north of Florida (Gray's Reef, Georgia, US, denoted GR; R2, offshore GR; R5, offshore GR, R6, offshore GR), 1 TG from British Oceanographic Data Centre (BODC) at Avonmouth**

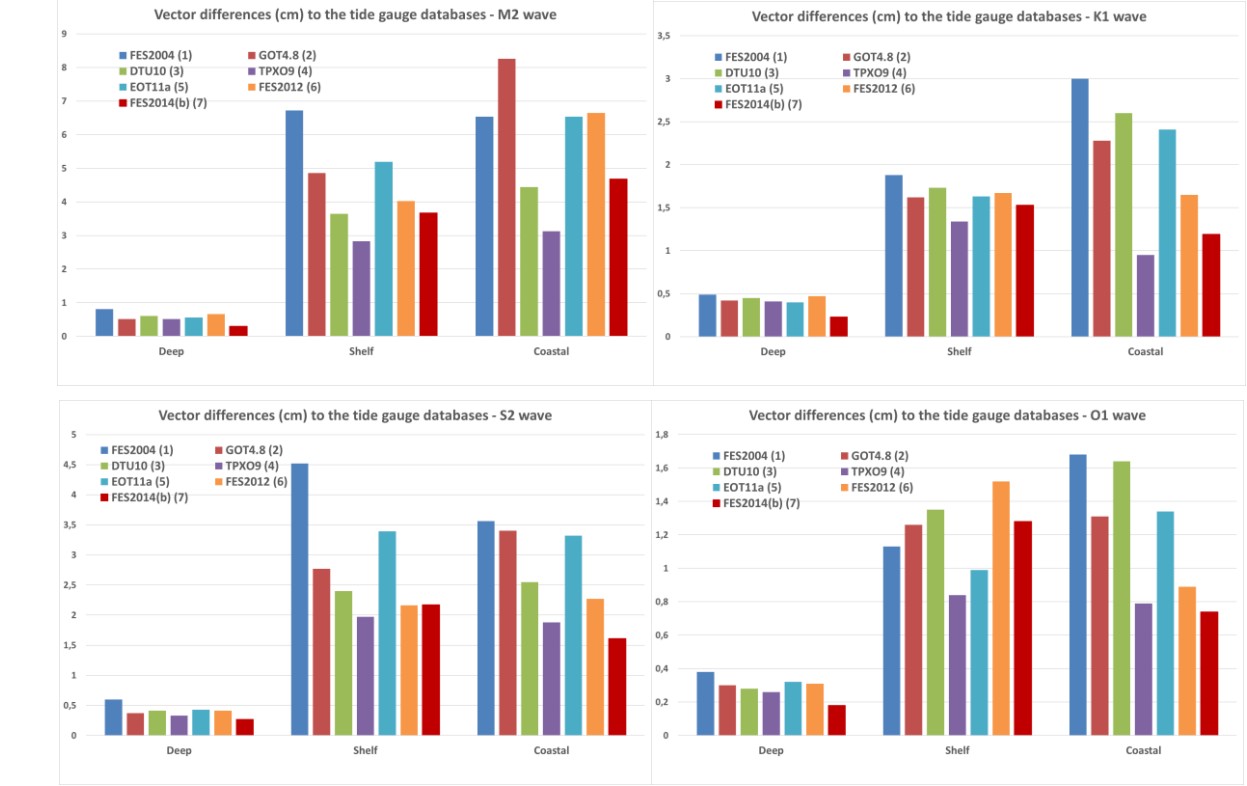

**Figure 12: Vector differences (cm) between the TG databases and the global tidal models, for M2, K1, S2 and O1. Deep group is made of abyssal plain TGs, shelf made of tides gauges located in upper 500m depth limit. Coastal group is made of TGs collected in coastal data databases such as GLOSS.**

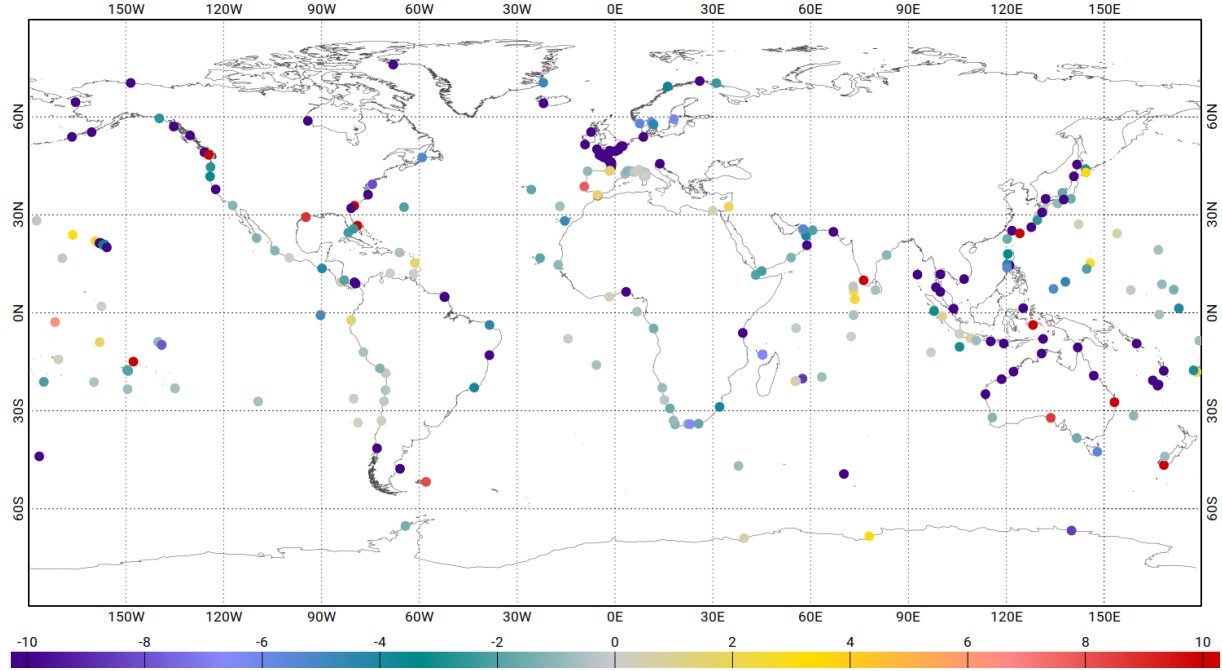

**Figure 13 : Variance reduction differences (cm²) at tidal gauge sites from GLOSS network, when using the FES2014b atlas versus the GOT4v10 atlas. Analysis computed over the 2007 to 2011 time period. Blue colours indicate a higher variance reduction when using FES2014b tidal correction. Tidal corrections made for both models with their native constituents spectrum (i.e. not restricted to their common constituents).**

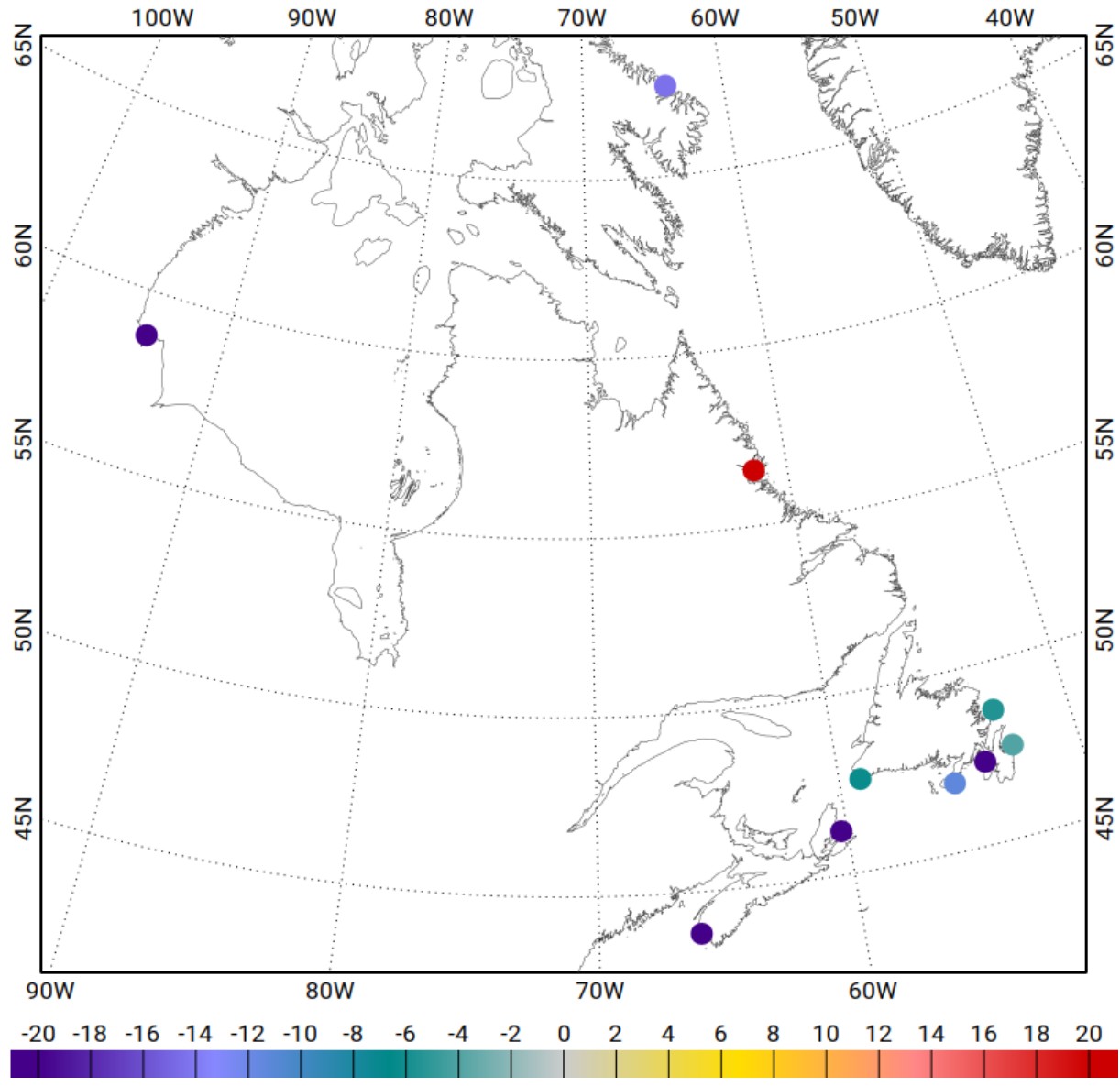

**Figure 14: Variance reduction differences (cm²) at Canadian tidal gauge sites, when using the FES2014b atlas versus the GOT4v10 atlas. Analysis computed over the 2007 to 2011 time period. Blue colours indicate a higher variance reduction when using FES2014b tidal correction. Tidal corrections made for both models with their native constituents spectrum (i.e. not restricted to their common constituents).**

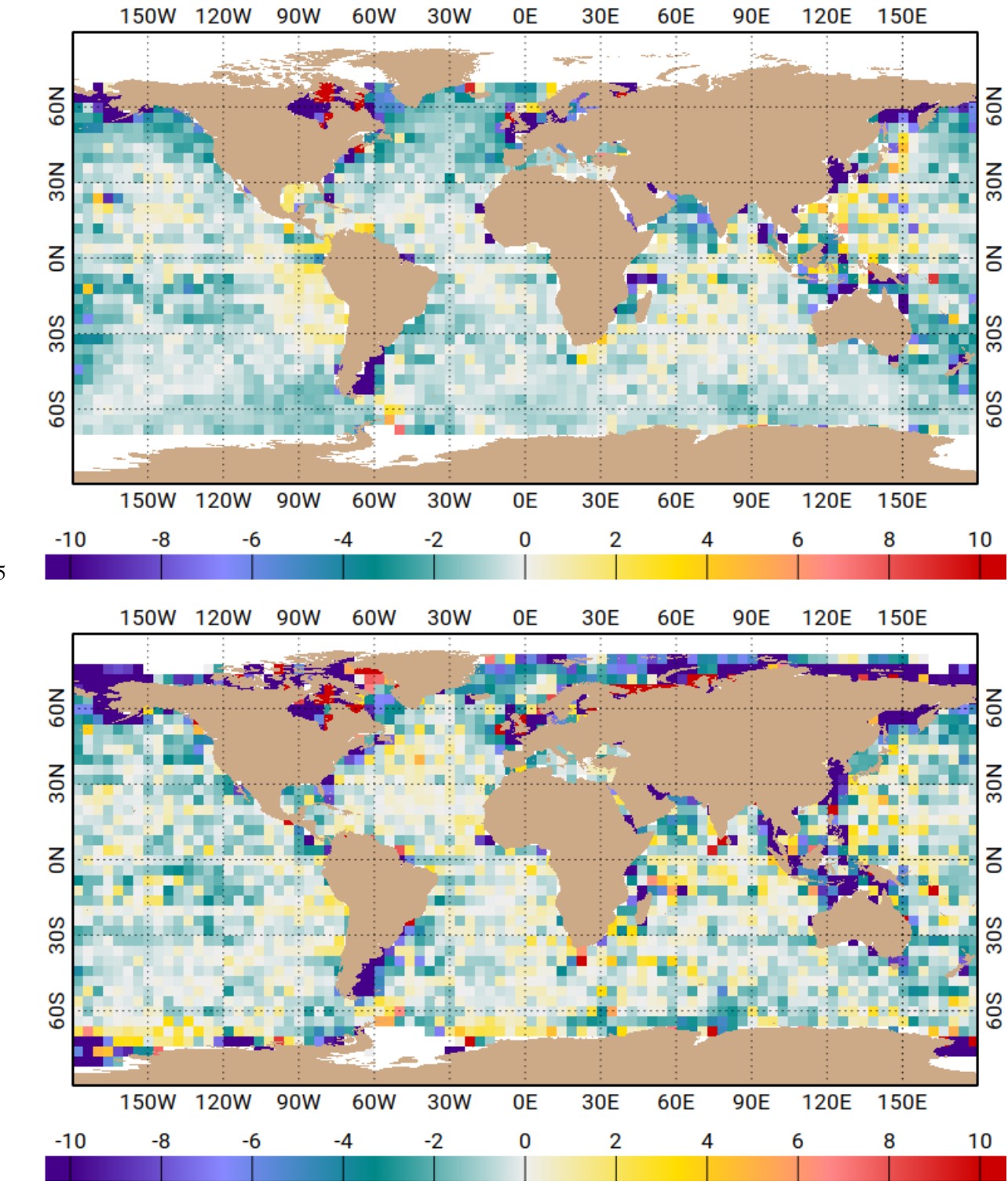

**Figure 15 : Maps of SSH variance differences at crossovers using either the FES2014b tidal atlas or the GOT4v10 atlas in the SSH calculation for the Jason-2 mission (upper panel, J2 cycles 1-281), and for AltiKa (lower panel, AL cycles 1-21, in cm²). Blue colours indicate a higher variance reduction when using FES2014b tidal correction. Tidal corrections made for both models with their native constituents spectrum (i.e. not restricted to their common constituents).**

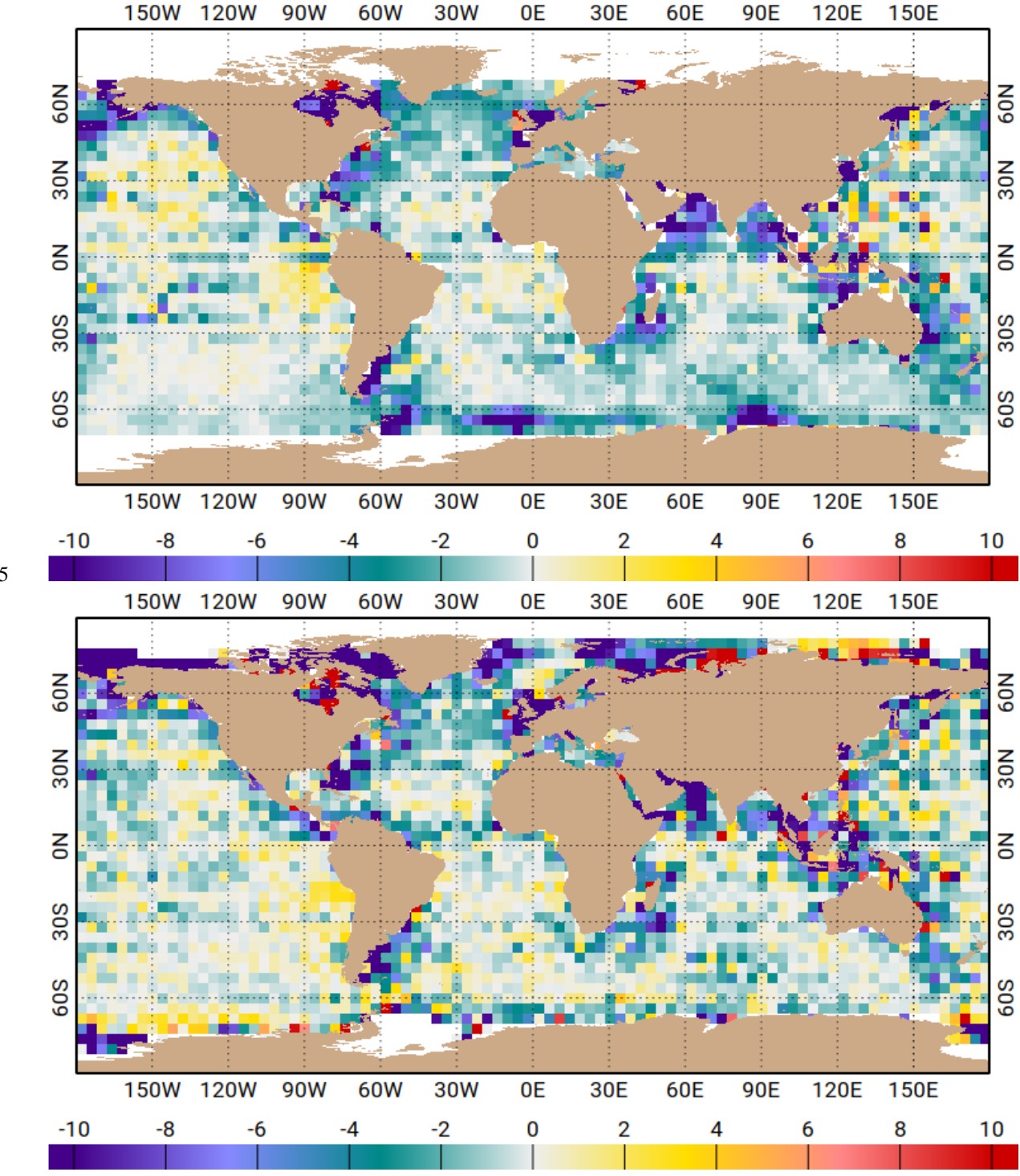

**Figure 16 : Maps of SSH variance differences at crossovers using either the FES2014a tidal atlas and the FES2012 atlas in the SSH calculation for the Jason-1 mission (upper panel, J1 cycles 1-248), and for AltiKa (lower panel, AL cycles 1-14, in cm²). The accuracy improvement between the FES2012 and FES2014 prior solutions is a key ingredient in the accuracy improvement between the FES2012 and FES2014a assimilated solutions. Blue colours indicate a higher variance reduction when using FES2014a tidal correction.**

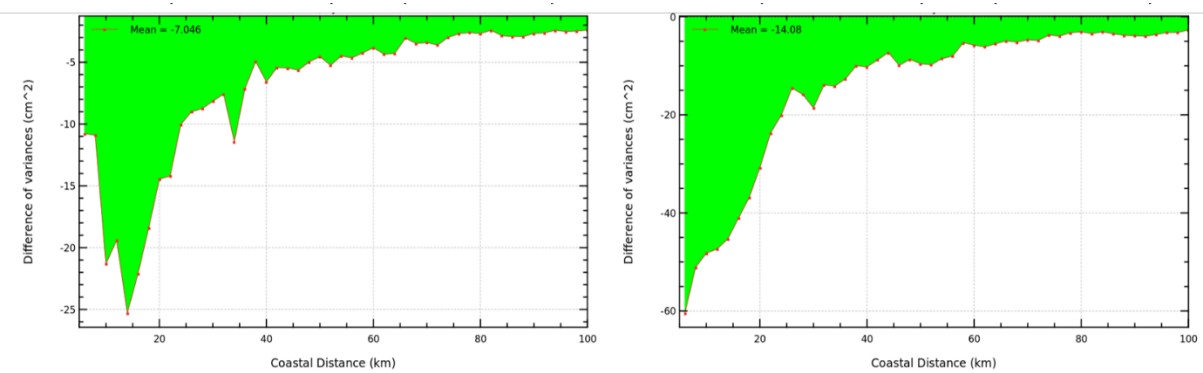

**Figure 17 : Difference of variance of SLA for the Altika (AL) mission as a function of distance to the coast, when using the new FES2014a tide model instead of the FES2012 solution (on left) or instead of the GOT4v10 solution (on right) in the SSH calculation (cm²). The accuracy improvement between the FES2012 and FES2014 prior solutions is a key ingredient in the accuracy improvement between the FES2012 and FES2014a assimilated solutions. AL cycles 1-14 are used. Tidal corrections made for both models with their native constituents spectrum (i.e. not restricted to their common constituents). When necessary, atlas solutions were extended toward the coast (low-order persistence).**

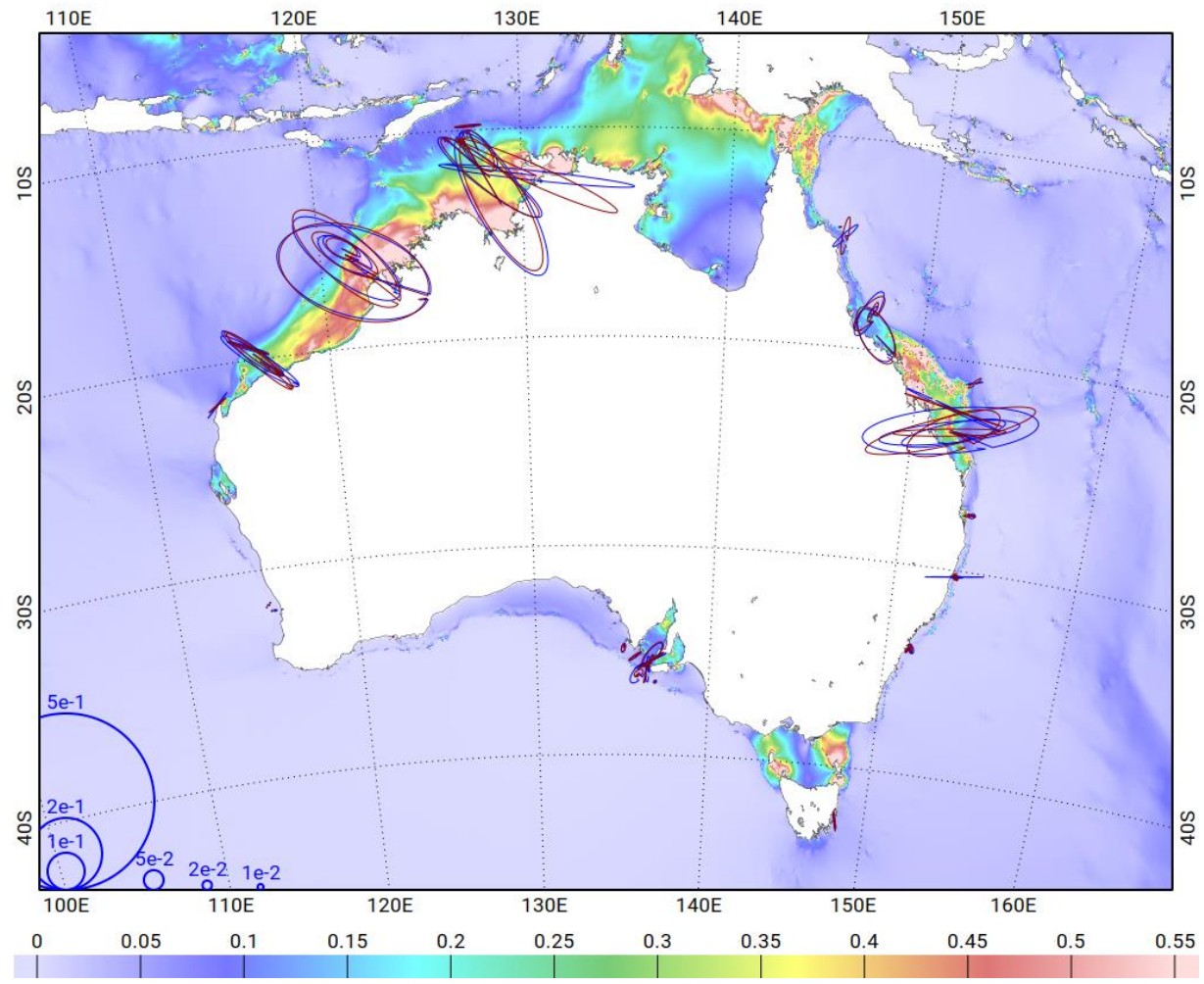

**Figure 18: M2 tidal component, tidal velocity ellipses at the 48 current meter stations around Australia, for the FES2014b tidal model (blue) and the ADCP observations (red). Ellipses scales in m/s. Inside line indicate velocity direction at Greenwich transit time, ellipse rotation from inside line to arrow-terminated ellipse contour.**

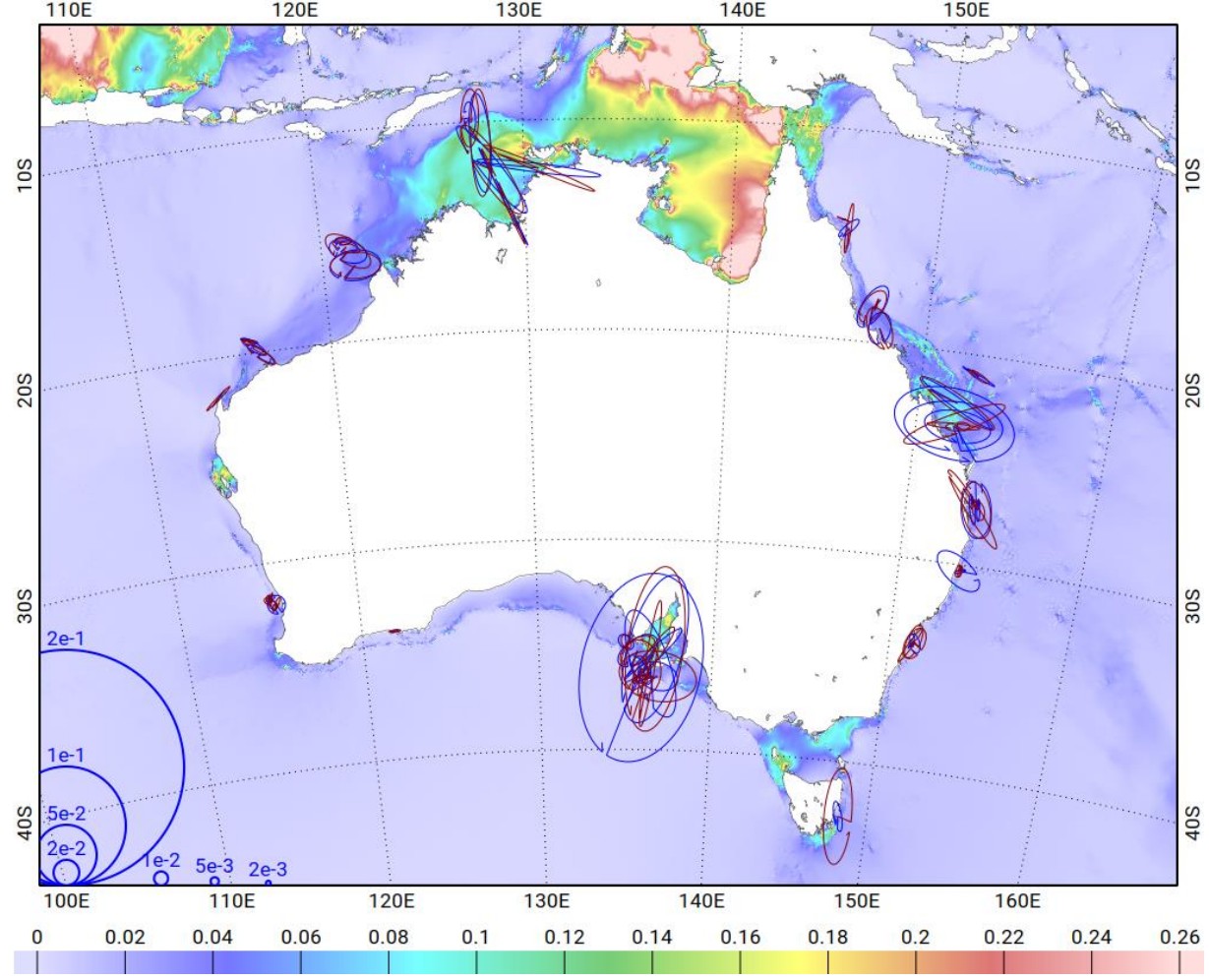

**Figure 19: K1 tidal component, tidal velocity ellipses at the 48 current meter stations around Australia, for the FES2014b tidal model (blue) and the ADCP observations (red). Ellipses scales in m/s. Ellipses scales in m/s. Inside line indicate velocity direction at Greenwich transit time, ellipse rotation from inside line to arrow-terminated ellipse contour.**

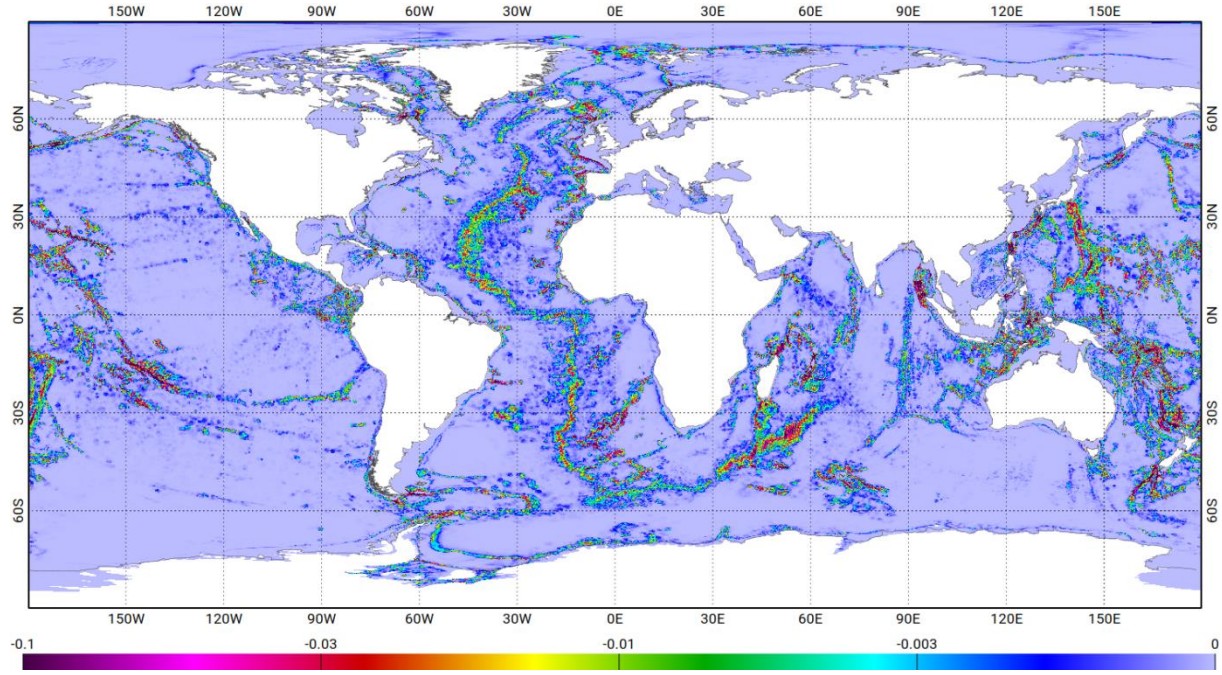

**Figure 20 : M$_2$ barotropic energy conversion rate (W/m²) toward baroclinic internal tides computed from FES2014 hydrodynamic prior.**

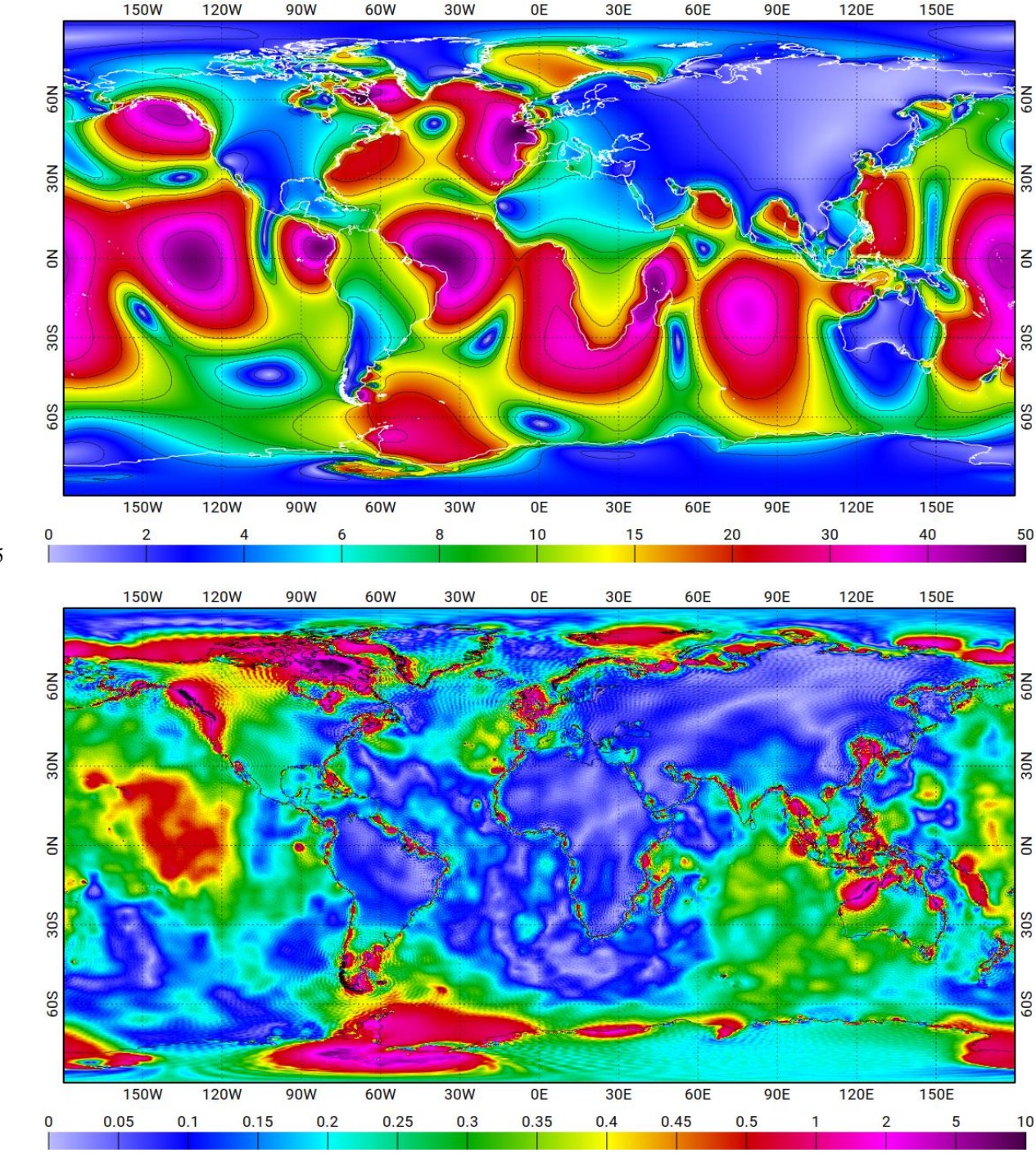

**Figure 21: upper panel, M2 tidal loading, vertical displacement (cm); lower panel, M2 tidal loading vector difference between FES2014b and GOT4.10 (cm). Wave-like patterns visible in some regions are likely due to differences between Green's functions based computation of LSA (as in FES2014-derived atlas) and spherical harmonics based computation (GOT).**

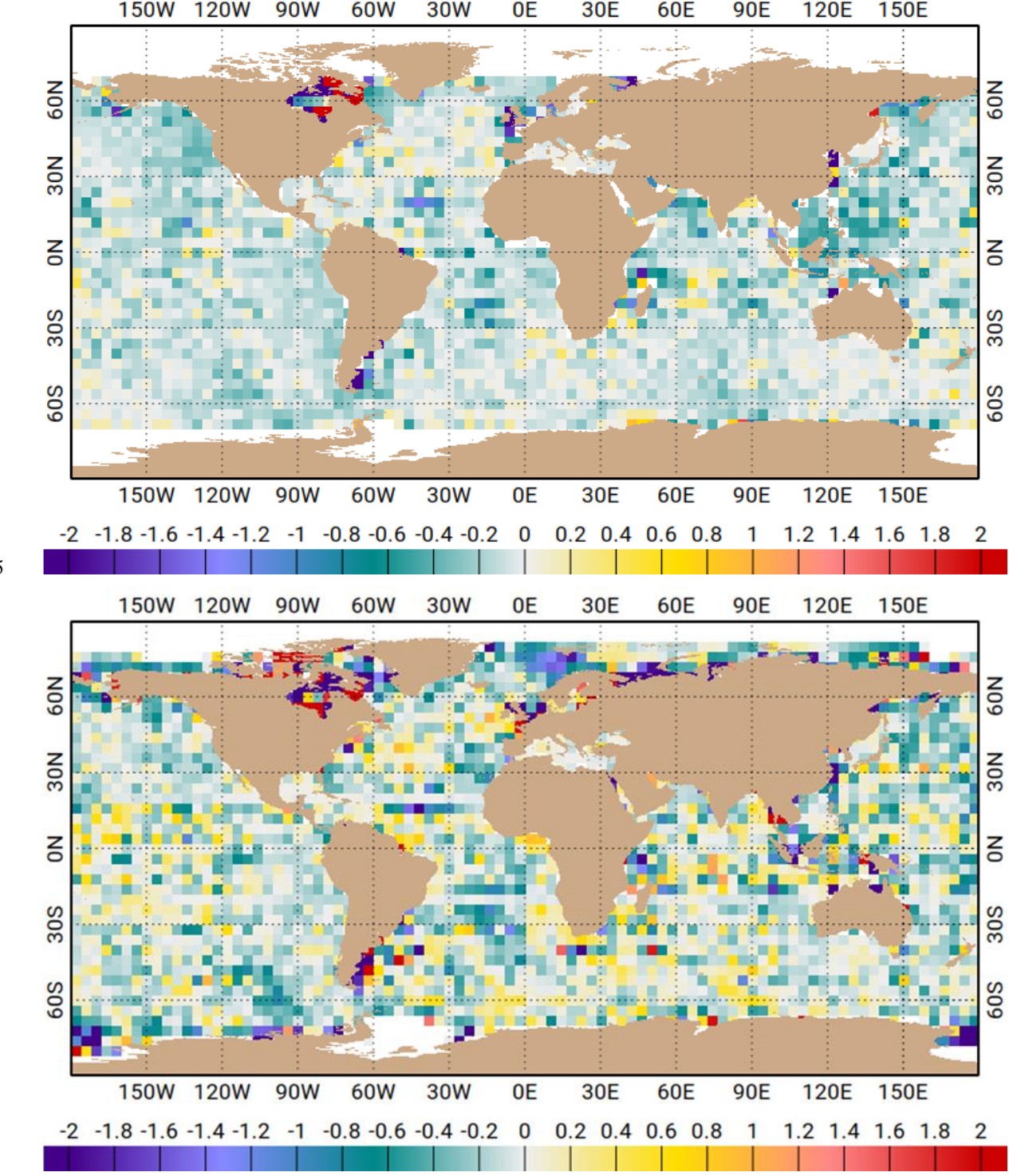

**Figure 22 : Maps of SSH variance differences at crossovers using the new FES2014b tidal model versus the preliminary FES2014a solution for the Jason-2 mission (upper panel), and for AltiKa (lower panel) (cm$^2$). To ensure the best consistency in the ocean and load tide correction (i.e. using the load tide identical to the one used to process the assimilated data), the FES2014b ocean tide is associated with the FES2014a tidal loading, while the FES2014a ocean tide is associated with the GOT4v8ac tidal loading. Blue colours indicate a higher variance reduction when using the FES2014b tidal correction**

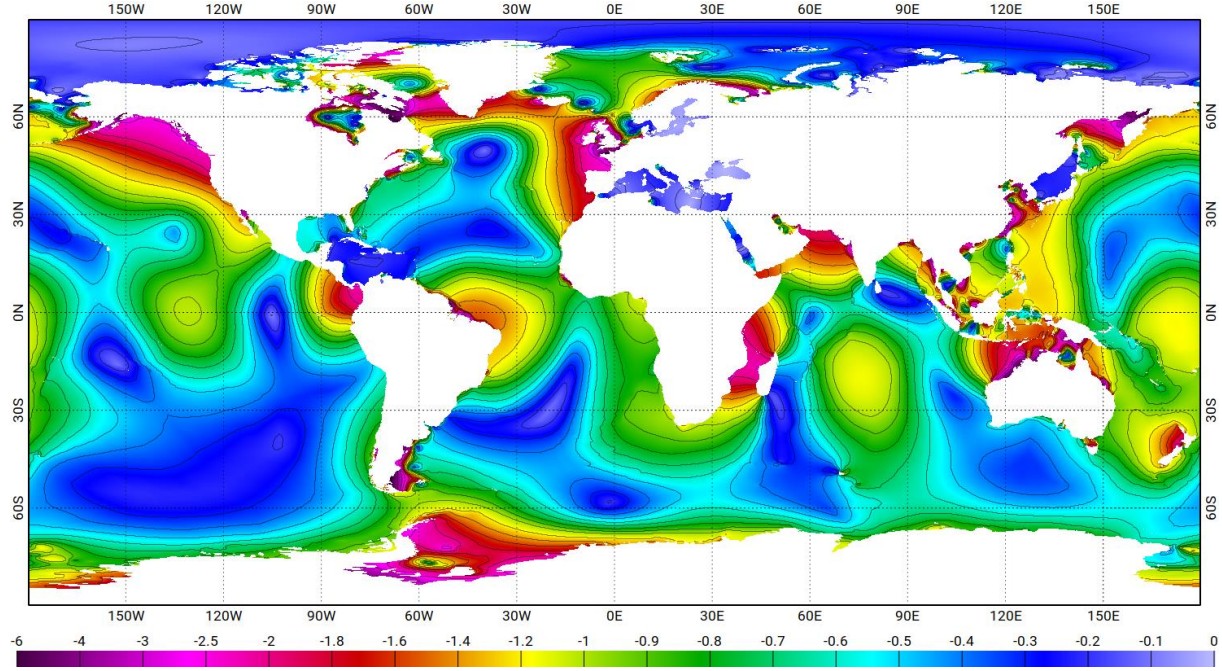

**Figure 23 : Lowest Astronomical Tides (LAT) relative to mean sea level computed from an 20-year FES2014b tidal prediction. Units in metres.**

| | M2 tidal component | | K1 tidal component | |
| --- | --- | --- | --- | --- |
| | Xover TPJ1J2 deep | Xover TPJ1J2 shelf | Xover TPJ1J2 deep | Xover TPJ1J2 shelf |
| **FES2004 hydrodynamic** | 4.56 | 12.32 | 1.45 | 4.19 |
| **FES2012 hydrodynamic** | 2.38 | 9.25 | 1.07 | 2.97 |
| **FES2014 hydrodynamic** | 1.53 | 6.44 | 0.88 | 2.26 |

**Table 1: RMS of the vector differences (in cm) between the purely hydrodynamic solutions of FES2004, FES2012 and FES2014, and the TPJ1J2 altimeter crossover points, for the M2 and K1 tidal components. The accuracy improvement between the FES2012 and FES2014 prior solutions is a key ingredient in the accuracy improvement between the FES2012 and FES2014a/b/c assimilated solutions.**

| | Satellite name | T/P-Jason | GFO | EnviSAT ERS-2 |
|---|---|---|---|---|
| | Satellite cycle (days) | 9.9156 | 17,0505 | 35 |
| | Darwin name | True period (days) | Aliased period (days) | Aliased period (days) | Aliased period (days) |
| Long period tides | Ssa | 182,62 | 182,62 | 182,62 | 182,62 |
| | Mm | 27,554 | 27,554 | 44,727 | 129,53 |
| | Mf | 13,661 | 36,167 | 68,714 | 79,923 |
| Diurnal tides | Q1 | 1,1195 | 69,364 | 74,050 | 132,81 |
| | O1 | 1,0758 | 45,714 | 112,95 | 75,067 |
| | P1 | 1,0027 | 88,891 | 4466,7 | 365,24 |
| | K1 | 0,9972 | 173,19 | 175,45 | 365,24 |
| Semi-diurnal tides | N2 | 0,5274 | 49,528 | 52,072 | 97,393 |
| | M2 | 0,5176 | 62,107 | 317,108 | 94,486 |
| | S2 | 0,5000 | 58,741 | 168,82 | ∞ |
| | K2 | 0,4986 | 85,596 | 87,724 | 182,62 |

**Table 2 : Aliasing periods of main tidal waves for TOPEX-Jason, ERS-EN and GFO altimeter samplings**

| | **TPJ1J2** | **TPNJ1N** | **ERS-EN** |
|---|---|---|---|
| **Min/Max latitude** | +/- 66.14° | +/- 66.14° | 80.25°N / 75.44°S |
| **Cycle duration (days)** | 9.91564 | 9.91564 | 35 |
| **Number of cycles used** | 743 | 223 | 172 |

**Table 3 : Description of altimeter data used**

|  | Area | Resolution | Max error on M2 | Nb data (M2) |
|---|---|---|---|---|
| **TP/J1/J2 crossover points** | Shelves | No decimation | 1 cm | 750 |
| | Open ocean | 200 km | 1 cm | 3677 |
| **TPN/J1N crossover points** | Shelves | No decimation | 2 cm | 278 |
| **E1/E2/EN crossover points** | Arctic Ocean | 100 km | 1 cm | 244 *(except S2)* |
| **TPJ1J2 along track data** | Shelves | 20 km | 1 cm | 6024 |

**Table 4: Selection criteria of the altimetry observations for the data assimilation process, depending on the mission. Deep/shelf limit is the 500m isobaths. "Arctic ocean" denotes seas located over the 60 degrees North limit. The harmonic data error is computed from an estimate of the non-tidal ocean dynamics contamination. It is estimated by looking at sea surface signal spectral energy close to the considered constituent's aliased frequency.**

