# Peer review of "Title page"

_Ocean Science, 2020_

## Referee Comment (RC1) · Jason Otero Torres (Referee) · 19 Nov 2020

Reviewed by: Jason Otero Torres & C.K. Shum November 18, 2020

General Comments:

The manuscript explains improvements achieved in FES2014b compared to its previously released versions. The FES global hydrodynamic ocean tide models have been a 'gold' standard of physics-based finite-element computational regime for the oceanography, geodetic, geophysical and other community, for a variety of applications and scientific research. The methodology for the model development is clearly and concisely described, without overburdening the reading with well-established theoretical background, which is already readily available in the cited literature. However, while

we understand the convenience to synthesize the overall communication, we expected to see additional details in several sections to better assess the presented results. In addition, we urge revising for grammar, presentation and descriptions in figures, and re-working the abstract to include general details found in subsequent sections. In sum, we recommend its publication provided that the authors adequately address the general and specific comments

Specific Comments:

1. Can the authors concisely explain why this new ocean tide model is named FES2014b? We understand that may be there is an upcoming FES2022 (?) model, it would be good if the authors explain the historic, current and future development of this very unique global hydrodynamic ocean tide modeling, for the benefit of interested users. This is also requesting to more clearly delineate FES models 2014, 2014b, 2014c, e.g., for clearly recommending to the users which model should they use?

2. The abstract overemphasizes the improvement of the new model from its previous releases. It is ok to highlight this in one or two lines, but please also include summarized details of each main section.

3. Abstract. Define ITRF; however, the authors may have meant IERS Standards, 2010? If so, please also define IERS and others in the manuscript as appropriate. For example, AVISO+, LEGOS, T-UGOm, CEFMO, LGP1/2, NCP1, GEBCO, ETOPO, RTOPO1, etc.

4. We note that various bathymetry models, including the one inverted using satellite radar altimetry (Sandwell & Smith). Can the authors comment on the applicability of the satellite inverted bathymetric model which would not be sensitive to the gravity signals resulting from coastal sediment compaction and loading? It would be great if the authors could characterize, approximately, the impact of the state of the art bathymetry model on coastal ocean tide modeling. For example, for the FES model is the bathymetry model accuracy still the limiting error, or other errors sources, in the OSD
state of the coastal ocean tide modeling?

5. Along the lines with the good discussions of S2, K1 aliasing (dependent on different altimeter data in the tidal solution), please discuss the issue of S1, and also may be pole tides?

6. Section 3.2: Removing the entire Sa and SSa constituents, or just the non-tidal contribution in Sa and SSa? Please clarify. Also further elaborate on GLORYS-v1.

7. Section 3.4: Explains the ice coverage problem in high latitudes. It seems that this section only applies to unbalanced observations caused by ice coverage and not other phenomena affecting the tidal estimates e.g. ocean circulation. This is explained in section 3.2 and in section 4.3 where fewer crossover data were added for the data assimilation process because of the large contamination of mesoscale processes. Please consider adding these effects (or others), affecting the separability and possibly state their impact on the extended frequencies in the FES2014b model version.

8. Figure 2-3: The details at some regions lost at the coastal regions for both the upper panel and lower panel. Consider an alternate symbology for the presentation of vector differences.

9. Figure 11: Add source for tide gauge database and each of the respective data count.

10. Related to above on pelagic tide gauge data validation of tide models. Can you comment or would you have cases where the tide gauge data used to initialize/constrain FES model are used and not used to evaluate the FES model?

11. Figure 16: A slight rise in variance is visible around Lapdev and Kara Seas for Altika mission altimeter data, when using FES2014 and FES2012 models as tidal corrections. Speculate on the reasons for such a rise in variance.

Please also note the supplement to this comment:

---

## Referee Comment (RC2) · Edward Zaron (Referee) · 20 Nov 2020

**Edward Zaron (Referee)**

edward.d.zaron@oregonstate.edu

Received and published: 20 November 2020

Review of OS-2020-96 by Lyard et al

This paper provides a thorough description and evaluation of the FES2014 tide model, in several variations. The rationale and design of the modeling and assimilation strategy for developing FES versions is explained very clearly, and the detailed description of their ensemble technique will be a welcome addiiton to the literature. This information will help ocean modelers understand the potential strengths and weaknesses of a model like this, and it will also guide others hoping to build on their work in the future. The detailed evaluations of the model and comparisons of its various versions and other tide models will be useful to potential end-users of the tidal atlas. I believe this

article will be of interest to readers of Ocean Science Discussions, and I recommend that it be published after some minor revisions, detailed below.

There are two main areas where the article should be revised to make it easier for a reader to understand and use the provided information:

1) In some places the article lacks clarity regarding the rationale for the comparisons using FES2014a, FES2014b, and FES2014c. Examples are noted in the detailed listing below. The manuscript would be a little easier to read if it clearly stated ahead of time which versions of the atlas are being used and why.

2) The article does not discuss at all the tidal prediction software which is provided with FES2014. I don't think that the article needs to discuss or provide this in detail, but it needs to be clearer about the nature of the tidal predictions used for computing variance reduction statistics. Do these predictions include all the tidal frequencies mentioned in the manuscript, including the ones which are forced with the ERA-interim in the time-stepped model? Do the predictions include smaller tides computed using inference? If so, do the inference formulas rely exclusively on those tides which were computed with assimilation, or do they use some of the other frequencies?

If the comparisons of different models use different constituents, then the manuscript should state this clearly in the tables or figures where the comparisons are shown.

Another aspect of the tide predictions needs to be discussed since the atlas is intended for correcting altimetry: How should FES2014 be used when a separate model is used to provide a DAC?

Detailed comments:

Good introduction which lays out the features of the tidal data assimilation: limitations of L2-norm, hence importance of the prior model and outlier rejection, and relative data sparsity depends on dynamics/wavelength.

Good explanation of FES2014a, b, and c releases.
p5,I7-9: "Initially, ... integrated in ..." I don't understand this sentence.

p5,I33: Does the discrete system satisfy any conservation laws?

p5,l35: Thus, there is no lateral eddy viscosity in the model?

p7,I8-9: "even at regional extents, as earlier ... compensation bias." I don't understand this part of the sentence.

p9,I8-9: Are you making a distinction between the tidal loading (the deformation of the earth surface caused by the tidal changes in ocean bottom pressure) and the solid earth tide here (the deformation of the earth surface caused by the gravitational perturbations of the sun and moon)?

p9,I30: Thanks for describing this in detail.

p9,I36: I would have expected that all the harmonic analyses were based on the longest records available, so this section on S2 can probably be deleted.

p10,119: Please explain this procedure more precisely. Is this equivalent to computing the standard error matrix for the least squares solution, and then recomputing estimates for fewer frequencies when the error correlation was too large? Exactly what criterion was used?

p10,I33: This is a little confusing. Are you saying that there are a total of N\_{tot}=432 ensemble members, and these are computed from sets of indepdently perturbed ensemble components? Do you have, N\_{tot}=N\_{bottom drag} \* N\_{internal tide} \* N\_{bathymetry} \* N\_{LSA} where N\_{LSA} = 2, corresponding to the FES99 and FES2012 load tides?

(I see, yes. Made this table while reading later:)

 $N_{bottom drag} = 8 * 13 = 104$

N\_{internal tide} = 10 \* 7 = 70
 $N_{bathymetry} = 2 * 18 + 6 = 42$

p13,I6: "SpEnOI code is solving the assimilation in the data space" – Are you aware that you could reduce the dimensionality to 432 by solving in the space spanned by the ensemble members, without reducing the quantity of data at all?

p13,I7-22: Can you reduce this text and just state what data you used for assimilation versus validation? I don't understand what data were used.

p13,l30: Did you use an estimate of the data error to weight the datasets that were assimilated, or did you simply assimilate fewer data in the noisier regions?

p14,I23: Could you please make clear whether "FES2014" refers exclusively to FES2014c in this section, or if you are describing aspects of the FES2014a and b solutions, too.

p14,I29: Earlier you stated that S1 belongs to the DAC solution. Please clarify.

p14,I40: I am confused about the use of ERA-INTERIM atmoshperic forcing. Wouldn't you want to compute purely gravitationally-forced versions of these, since the atmospherically-forced component would be provided by a separate DAC model? Since the goal of the paper mentions developing the tidal atlas specifically for dealiasing altimetry data, you should be clearer about how this should be done in practice using FES2014c, since there is the potential to unintentionally duplicate corrections from the tidal atlas and the DAC correction as it is usually applied.

p15,I30: Could yoiu please remind us about why you are validating FES2014b instead of FES2014c here?

p15,l36: Does "TPXO9" refer to the "TPXO9v2" mentioned on page 14? And, does "TPXO9-atlas" also refer to "TPXO9v2"?

p16: I am unclear on why FES2014a and FES2014b and FES2012 are used in some of these comparisons. I understand if this is simply related to the reporting of comparisons
originally conducted for various purposes, but perhaps you could mention the reasons at the beginning of this section.

p17,I31: Can you cite a source for this statement? This certainly depends on depth of the station, stratification, etc.

p18,128: "dynamical quasi-coherence of the covariances" -1 don't know what this means. Please explain or omit.

p18,I30: Can you say anything about the results of Fig 21? If they don't show anything new, then perhaps this section can be omitted. Alternately, maybe they basically confirm but slightly alter previous estimates. Or maybe there is something that can be gleaned from comparisons of recent calculations by DeLavergne et al. Since you previously stated that the currents are not reliable in regions of steep topographic slope (line 13), why would the energy flux divergence or the parameterized baroclinic wave drag be reliable?

p19,I26: "After proper, competitive evaluation procedures" – Without explanation, the reader is not going to know what you are referring to. Could you either omit this phrase, provide a citation, or explain this topic further?

English usage/typos:

p5,I5: omit "about a"

p5,I42: "it is known to allow for" -> "that it allows"

p6,I7: "multi-levels" -> "multi-level"

p6,I10: I am more accustomed to seeing "CFL" rather than "CLF" for this abbreviation.

p6,I16: omit "and"

p6,l27: "Go" -> "GB" ?

p7,I18: "SAL" -> "LSA"; and fix "atlas atlases"
p7,I42: Why repeating S1 in the parens?

p7,I38: Can you reorganize this paragraph for clarity? Something like:

"The hydrodynamic solution for the S2 tide differs from the other tides in that the atmospheric forcing is explicitly included. ... The S1 tide originates mostly from the atmospheric forcing, and it was therefore not computed in the hydrodynamic tidal solution. ..."

p8,I16: omit comma at the end

p9,I14: please fix "Then,18); this"

p9,I20: I thought Parcel's theorem was the equality of the sum of squared Fourier components with the variance. Please reword or explain . p9,23: "guaranty" -> "guarantee"

p10,I11-12: omit "to ease the harmonic system solving"

p11,l8: ", 1982" -> "(1982)"

- p11,I14: "abuse" -> "abuse of nomenclature"?
- p11,I18: "is run" -> "are run"
- p11,l21: "tide drag" -> "internal tide drag"?

p12,I6: "sloppy" -> "sloped"

p12,I9: "efficiency is strongly dependent on" -> "depends on"

p12,l13: "(75)" ?

p12,I19: commas after "GEBCO" and "release"

p14,I6: "repartition" -> "distribution"

p14,I16: "it small" -> "its small"

p14,l17: "ration" -> "ratio"
p16,l9: "if" -> "while" or omit "if"

p16,I14: "none data" -> "none of these data" or "no data"

p16,111: Presumably, you used all the tidal harmonics available from each model in these comparisons, since you mentioned errors of omission previously. You should remind the reader of this here, or later in the Discussion.

p16,I20: Can you redo this comparison of FES2014a with FES2014b, instead? p17,I40: repetitive of line 34; please consider revising this entire section for brevity.

p18,I15: omit "Somehow,"

p18,I37: capitalize "Love"?

p18,I39: omit "then derivation"?

p19,I23: "others" -> "other"

p19,I35: "coastal details grid flexibility" -> "detailed coastal grid"

p19,I38: omit "generation"

p20,I5: "atlases" -> "atlas"; omit "locally strongly" on I6

p20,I9: "eased" -> "open"

p20,l11: "as well in terms of" -> "for both"

Figures and Tables:

p24-25,F2 and F3: The caption mentions panels a, b, and c, but only two panels are shown. Pleas label the panels a, b, and c, and include the Darwin name boldly somewhere on each plot.

F4: capitalize "Maps" in caption. Other captions also need this correction.

F8: The figure shows that the bathymetry of Weddell sea was perturbed, but I thought
the text mentions that, after initial experimentation, it was not.

F12: Please clarify how "deep", "shelf", and "coastal" are defined. Is it a depth criterion, or simply whatever the data source happened to label it as.

F13: I cannot distinguish the dark colors at the ends of the colorscale. Maybe represent this data with different size disks, like your previous comparisons in Fig 2? It is difficult to understand the information on this map.

F14: Consider coloring the land and sea differently (white vs light gray). Does blue mean that FES reduces more variance?

F15: Is this comparing tide predictions made with the same constituents from each model? If not, you should explain to the reader that FES is providing more constituents, and/or cite Stammer et al, or Zaron and Elipot (2020, https://doi.org/10.1175/JPO-D-20-0089.1) where the comparisons are restricted to common constituents.

F17: Once again, it is important to be sure the reader is aware whether the same constituents are being used, especially since FES2014 is using a number of frequencies with substantial atmospheric forcing. Also – GOT is not really intended for use close to the coastline. Did you extrapolate the values landward, or does this comparison only use locations where GOT grid cells happen to overlap the coastline?

F18: I think this figure can be omitted.

F20: Can you use a different scaling to make the current ellipses more visible? Consider omitting this figure if you can't revise it. Or, maybe show a few representative comparisons enlarged, instead.

F23: The caption is hard to understand. Could you describe this with something like: "Map of the SSH variance difference a crossovers using the FES2014a tidal loading versus the GOT4v8ac tidal loading. ..."

p39,T1: fragment at end of caption, "3 Tidal harmonic ..."

OSD
p39,T2: Please fix "#DIV/0!". Also the different vertical alignments and number significant figures displayed for values in the table is disconcerting. Also, unlike the other numeric values in the manuscript (cf., Table 3), the "," is used instead of the "." to indicate the decimal point; please revise for consistency.

p40,T4: What criteria are used to distinguish "shelves", "Open ocean", and "Arctic Ocean"? Is the Max error criterion applied to the least-squares error estimate (from the harmonic analysis), or do you estimate the error in some other manner?

---

## Editor Comment (EC1) · Philip Woodworth (Editor) · 23 Nov 2020

November 2020

Comments on 'FES2014 global ocean tides atlas: design and performances' by Lyard et al. (OSD)

Below are some general comments on the draft with an editor hat on, I leave any more scientific comments to the reviewers. I found the text rather intense and technical but that is probably inevitable. However, there is at least a need to define all acronyms, for example, and have proper referencing. There is also inevitably some french-english.

A second problem is that, while most of the figures might be acceptable (although parts (c) of Figures 2 and 3 are missing), they are a mixture of styles which does not look

good. I make some comments on each below.

Comments on the text, apologies there are so many:

p3, 5, title - tides –> tide, performances –> performance

- with a tidal

- tidal constituent spectrum

- diagnostics and the Lowest and Highest Astronomical Tide and other hydrographic datums.

performance performance accuracy towards the end error covariance data sets.

p4, 5 methodological pretty –> very define CNES

the decision was made performance by the GOT model (reference)

resolution grid on the website (give http)

in 2019 by extending its long-period spectrum to include low-frequency ..

the reader with information a basic accuracy dependence define these acronyms based on the usual ..... with a non-hydrostatic

The ITWD

parameterization, pioneering p5,5 accounts for a significant times smaller solver's parameter

Consequently, we will confine ourselves velocity currents adapted for the global ocean to include near-shore p6, 5 elevations

Reversely –> Conversely, it

Reversely –> On the other hand, minor

What is Go? GBytes?

define acronyms and give proper references give year and add reference for Timmermann et al.

p7, 6 we have always even on a regional level you have defined these FES versions before and later without the hyphens ditto difference RMS reduced by nearly a factor performance such as because of the intrinsic variability of the atmosphere we consider clearly p8, 8 have been used in validation of simulations and in data assimilation steps by means of harmonic how much –> how put 'respectively' at the endd of the sentence time series raise more ... dependence signals dataset, but with larger uncertainties than higher spatial temporal under-sampling

Reversely –> Conversely not only are the S1 and S2 tides projected completed –> complemented p9, 8 targeted applied to the altimeter noted –> denoted ditto what is the funny 18)

aliased to and to the annual analysis by the non-tidal signal is severe virtue of the Parceval Rule (reference)

guarantee portion of the annual and so to tidal harmonic

K1, and will.

The misfits –> Such differences were found to be consistently demonstrating the benefits of the model-based correction

Because this signal was stronger during the TOPEX period [and why was that?]

Jason-1/Jason-2 relatively recent record.

p10, J1-J2. I think these sort of acronyms are asking a lot of the normal reader. I know what this means but I am not sure other people will. Also should not J1-J2 appear in Table 3 for example?

accurately the harmonic separation performance.

[Figure]

gap durations consequently larger errors in the harmonic damaging tide model by essence –> by definition?

internal tide

28-29 I don't understand this sentence. Could you reword it? Maybe it also needs a reference tide becomes shorter.

with the notable substituted by forcing terms as a variational as is the case p11, 7 - a representor

Although the variational poorly able has been constructed to ... error demoniation is a mis-nomer as the error covariances of state vectors are not idealised ... but are justified are run experiences –> experiments dependence in Figure 6

global-average p12, 7 - sloping in Figure 7

global-average regions using either synthetic extracted from what we call 'gridone', of the reference the Weddell Sea region p13, 6 solving an assimilation feasible of the why 20 years? There has been 28 years since T/P launch?

estimate of errors consisted of enables us

Stammer et al should be 2014

what are Kowalik etc. (I know but the reader will not). Please give proper references.
Define acronyms p14, 7 - very peculiar –> particular between the separate the M4

in the M4 analysis kept from –> kept in??

Avon Mouth –> Avonmouth

Bay of Bristol –> Bristol Channel components (twice)

performance is drop 'rather'

- why don't you include third-degree tides, especially M1? Although only a few mm or a centimetre at most places, it will be larger than some of the second-degree tides you have here p15 top - I think, as becomes evident from the figures, you need to make clear that Sa and Ssa come from an ocean model as well as tidal forcing

- Stammer 2014

gauge data

The TPX09 atlas performance in p16, 7 - tide gauge why do you define TG here when 'tide gauge' has been used a lot before. Please defineit first time and use from there.

9-10 I don't understand this sentence. Please reword.

the GLOSS

.. regions, ... no data has the GOT

gauge using the the GOT4v10

in the global the Jason of all the models tested ... variance tidal models respectively.

Statistics for Altika reduces to the GOT

variance when using the for the Altika .. to the coast p17, 10 current maps ... budgets in the global elevations where tides are the major contributor to variability .. validation of tidal currents (as they are based

, and vertical current profiles ellipse dop 'Precisely'

in Figure 19

p18, 7 Globally –> Overall depicts

The barotropic tide energy budget is a valuable diagnostic .. performance proxy for the interaction used to provide additional vertical diffusion information in ocean ..

In the using a spherical harmonic/Love number approach ... Green's

Green's function p19, 8 why is this final as you mention a version 'c' at the start assessment constituents

Mean Lower Low Water (MLLW) and Mean Higher High Water (MHHW)

These are not just used by NOAA. They are two of many rather archaic hydrographic datums. You could maybe refer to one of the annexes of the Pugh and Woodworth (2014) book.

21-22 where the accuracy of tidal atlases .. limited for precise define ITRF and give reference user community were able to accumulate performance in the tidal short-list

- define SWOT and give a reference (probably Morrow et al. in Frontiers a couple of years ago)

emphasis on existing public data release p20, 5 future atlas believe that the

... correction, in terms of surface elevations as well as tidal

The FES2014 project ..

- framework no [superscript to be consistent]

Comments on figures - sorry quite a few here. My main complaint is that their styles are very different, and (although I am not convinced it is necessary) but maps using have Longitude and Latitude axis titles.

Fig 1 - the lon/lats numbers are very small and the colour scales are cramped with them.

line 9 - you haven't defined in the text what resolution ratio means. You should define it here otherwise.

Fig 2 - (a) and (b) annotation needs adding. But figure (c) is missing completely?

Fig 3 - ditto

Fig 4 - say what the units are in the caption or the colour bar

Fig 5, line 1 - signature. What does number of points mean? I couldnn't see that in the text.

You can see this is very different style to Fig 1 for example

Fig 6 - Watt should be W

It seems strange to have negative numbers increasing to the left, but ok.

Are the region numbers used in the paper? If not I would remove them.

line 1 - dissipated by bottom

Figs 7 and 8 - you can see on one page what I mean about different styles

Fig 12 - the fonts are very small

Figs 13 and 14 - please can you remove all the clutter on land with lakes and rivers? It is hard enough as it is to see the coloured dots. And in Fig 14 a simplified coastline might be best as your eye is taken by all the detail.

In the colour bar cm2 should be superscript

Fig 15 line 1 - variance line 2 - for the Jason-2

It would be good to remove the lakes etc. from these maps also. Who cares about them?

Fig 17 - the font is again very small on the axes

Fig 23 line 2 - for the Jason-2

Fig 24 - Lowest Astronomical Tide (LAT) relative to mean sea level ..

Table 2 - Wave period should be (days). Please could you have decimal points and not commas?

---

## Referee Comment (RC3) · Michael Foreman (Referee) · 30 Nov 2020

This is a nice overview of the evolution and present status of the FES series of global tidal models. Brief descriptions of the major components in the model (e.g., finite element/volume approaches, assimilation techniques and data, self-attraction and loading, bottom friction and internal drag coefficients, barotropic to baroclinic conversion) are given along with numerous figures illustrating improvements from earlier versions, and comparisons with other global models. Though I recommend publication, I feel the paper would be stronger if the following, generally minor, issues could be addressed.

General Comments:

1. Though the manuscript title specifies "atlas design and performance" (I suggest

making "performances" singular), I would like to see more information on the atlas itself. The manuscript states (page 4, line 22) that the actual FES tidal constituent harmonics can be found on an "AVISO+" website, but I think the specific URL should be given (here or in an Appendix) (https://www.aviso.altimetry.fr/en/data/products/auxiliary-products/global-tide-fes.html ?) along with a short summary of what (not only data but software?) is available.

2. Though the numerous global maps have lots of information, not all of the details are accessible. Specifically, as a coastal modeler I would not only like to see the M2 and K1 amplitudes in the specific region where I work but also the location of assimilated tide gauge, cross-over, and along-track data. (This information will assist in determining whether: i) FES could be used to provide boundary conditions for a regional model, and ii) the FES de-tided altimetry is likely to have adequately removed smaller scale tidal variations.) For many of the global figures in this manuscript, even when I zoomed-in, it was not possible to see this information clearly. I don't know how this manuscript will be published electronically but urge the authors (and the journal's scientific and technical editors) to provide sufficiently, highly-resolved figures so that when zoomed-in, interested readers can pick out the details they want. On Figures 2, 3, 9, 10, and 13 for example, this may require changing the dot scaling so that as one zooms-in, the dots become smaller so their precise locations become evident. In the case of Figs 2 and 3, smaller dots would no longer obscure the background image but perhaps also allow more amplitude contours to be displayed.

3. Along the lines of comment 2, I also would like to see the actual FES model grid (page 6, lines 14-22), especially its coverage and resolution in shelf and coastal regions. Figure 1 partially addresses the resolution issue but even when zoomed in, it doesn't provide the detail I would like to see. Perhaps an image of this grid (with zoom-in capability) exists somewhere on an AVISO+ or LEGOS/CRNS/CNES website but I couldn't find it in a quick search. If it does exist, then that location should be given. If it doesn't, then I strongly recommend that be done, as again, the information would be

useful for regional and coastal modelers.

Specific Comments:

1. Page 3, line38-40: What other norms can be used? A reference should be given.

2. Page 5, line 5: remove "about"

3. Page 5, line 33: Has NCP1 been defined?

4. Page 6, line 5-8: Give a reference for this statement.

5. Page 6, line 10: CFL vs CLF ?

6. Page 6, lines 19-22: As above in general comments 2 and 3, it appears from zooming-in on Fig 1 that Juan de Fuca and Queen Charlotte Straits (south and north of Vancouver Island) are in the grid (resolution 5-10km?) but the Strait of Georgia (which has a partial TPJ track) and its northern passages are not. Presumably you compensate for this by assimilating crossover, tide gauge and along-track data. Zooming-in on Figs 9 and 11, I can see what crossover and tide gauge data are used but not so for locations of the along-track data in Fig 10. Furthermore, zooming-in on Fig 3 doesn't provide the detail to see, for example, if those data have caused the FES solution to at least partially capture smaller scale features like the surface signature of K1 shelf waves off the Vancouver Island coast.

7. Page 6, line 27: Gb vs Go ?

8. Page 7, line 24: It's not clear if these coefficients are spatially constant over the entire globe or they differ within say, the polygons in Figs 7 and 8.

9. Captions to Figs 2 and 3: These are not clear. The a), b), and c) references suggest there should be 3 panels, yet only 2 are presented. If a zoom-in capability were available for these figures, you could probably include the shallow/coastal crossover locations too.

[Figure]

10. Section 3: I wonder about non-stationarity, especially in coastal regions where seasonal changes in river discharge, winds, and ice cover may interact with the tides. Is this considered?

11. Page 9, line 20: Give a reference for Parceval's rule.

12. Page 10, line 14: In the Canadian Arctic, the additional drag from seasonal ice cover changes the constituent amplitudes and phases at some coastal locations. Can this non-stationarity be accounted for?

13. Page 10, lines 19-22: I'm interested in more details on this. Is there a reference?

14. Page 10, line 29: The text says the filtering wavelength is in km while the Fig 5 caption says it is in number of along-track points.

15. Page 11, lines 28-30: Why is that?

16. Page 11, line 37: Presumably there was a dissipation cutoff to determine these polygons, as not all purple regions warranted one. What was it?

17. Page 12, line 23: The polygons on Fig 8 are not numbered.

18. Page 12, line 32: Does it have to be the same everywhere? You probably need 10m in regions like the Bay of Fundy but it shouldn't have to be that large in many other coastal areas.

19. Page 14, line 18: "used in" vs "kept from" ?

20. Page 15, line 36: It seems that TPXO9 is more accurate than FES2014b in the shelf and coastal regions. Can this be attributed to their generally higher (1/30 degree) spatial resolution?

21. Page 17, line 6: Why is there a peak at about 14km from the coast in the left panel of Figure 17?

22. Page 17, line 28: Give a URL for this portal.

23. Page 17, line 36: Often phase lags are included in these ellipses by showing the current vector position at the time of maximum tidal potential at Greenwich. Also, an arrow is sometimes placed on the ellipse itself to denote the sense of rotation. Were these not done in Figs 19 and 20 because they would make the figures too complex?

24. Page 17, line 42: You should change the scale to make the ellipses larger for K1. Perhaps a zoom-in capability (ellipse size changing as you zoom-in) is needed here in order to decipher which ellipses belong to which dot.

25. Page 18, line 11: I can guess which station this is but maybe you should give the approximate lat/lon to help readers.

26. Figure 21: A zoom-in capability that doesn't blur the color details would be useful here.

27. Page 19, lines 5-6: Why does the lower panel of Fig 22 show wave-like (Gibbs?) patterns, for example radiating eastward and westward from the Canadian Pacific coast?

---

## Author Comment (AC1) · 8 Jan 2021

(for commodity reasons, we have reproduced the reviewer text in black, answers are in light blue, further action to the revised manuscript in bold blue)

The manuscript explains improvements achieved in FES2014b compared to its previously released versions. The FES global hydrodynamic ocean tide models have been a 'gold' standard of physics-based finite-element computational regime for the oceanography, geodetic, geophysical and other community, for a variety of applications and scientific research. The methodology for the model development is clearly and concisely described, without overburdening the reading with well-established theoretical background, which is already readily available in the cited literature. However, whilewe understand the convenience to synthesize the overall communication, we expected to see additional details in several sections to better assess the presented results. In addition, we urge revising for grammar, presentation and descriptions in figures, and re-working the abstract to include general details found in subsequent sections. In sum, we recommend its publication provided that the authors adequately address the general and specific comments

Specific Comments:

1. Can the authors concisely explain why this new ocean tide model is named FES2014b? We understand that may be there is an upcoming FES2022 (?) model, it would be good if the authors explain the historic, current and future development of this very unique global hydrodynamic ocean tide modeling, for the benefit of interested users.
The FES atlas series has started with the FES94 release, quickly followed with the FES95 one, which includes some upgrades and fixes for various issues detected after the FES94 official release. A similar scenario occurred for the FES98 and FES99, FES2002 and FES2004, FES2012 and FES2014 atlases production. Despite intensive quality checking during production phase, any new major version of FES atlas release is followed by an extended verification/validation phase from the FES team and other world-spread specialists through the science applications that used the new atlas. The upgrading/fixing step is limited to issues that do not demand any major changes in the production process (such as unstructured grid modifications) but still will bring valuable improvements for the final user.
This is also requesting to more clearly delineate FES models 2014, 2014b, 2014c, e.g., for clearly recommending to the users which model should they use?
The FES2014a atlas is an intermediary step, which was only released for the need of performances assessments by the tidal community (using GOT LSA). A new loading and self-attraction atlas has been computed from the FES2014a release, and use to generate the FES2014b release (instead of GOT LSA). The FES2014b is the official project release, and should have remained the last one. However, to provide a more comprehensive, coherent tidal spectrum for tidal predictions particularly for the geodesic community, several long period tides constituents were explicitly added (computed from mass-conservative equilibrium solutions) to the FES2014b atlas. It must be noticed that similar long period constituents are implicitly added in tidal prediction if no external solution file provided. The extended atlas has been named as FES2014c to avoid confusion.

2. The abstract overemphasizes the improvement of the new model from its previous releases. It is ok to highlight this in one or two lines, but please also include summarized details of each main section.
**It will be done in the revised manuscript.**

3. Abstract. Define ITRF; however, the authors may have meant IERS Standards, 2010? If so, please also define IERS and others in the manuscript as appropriate. For example, AVISO+, LEGOS, T-UGOm, CEFMO, LGP1/2, NCP1, GEBCO, ETOPO, RTOPO1, etc.
**It will be done in the revised manuscript.**

4. We note that various bathymetry models, including the one inverted using satellite radar altimetry (Sandwell & Smith). Can the authors comment on the applicability of the satellite inverted bathymetric model which would not be sensitive to the gravity signals resulting from coastal sediment compaction and loading?
The authors are fully aware of inverted bathymetry issues on shelves and coastal regions, and consequently were not used in such locations except in some specific areas, in absence of any other more accurate bathymetry. It must be noticed the latest GEBCO distributions now include patches coming from inverted bathymetries, which is a serious issue for using recent GEBCO distributions in FES model bathymetry.
It would be great if the authors could characterize, approximately, the impact of the state of the art bathymetry model on coastal ocean tide modeling. For example, for the FES model is the bathymetry model accuracy still the limiting error, or other errors sources, in the state of the coastal ocean tide modeling?

Bathymetry still remains unfortunately the limiting error to our prior hydrodynamic solutions in most of the global ocean, and also impact the data assimilation accuracy in shallow waters regions. For most of Northern American, European and Japanese waters, this not so much the case. For instance, thanks to the impressively accurate new bathymetry of the European shelf (as available through EMODNET products), most of errors due to bathymetry have dramatically reduced, so we could clearly demonstrate that a wetting/drying scheme is necessary to reach the best tidal accuracy in the North Sea. Using more ancient bathymetry would have totally blurred this point, making any conclusions uncertain.

5. Along the lines with the good discussions of S2, K1 aliasing (dependent on different altimeter data in the tidal solution), please discuss the issue of S1, and also may be pole tides?

The astronomical part of S1 is rather negligible, and it is mostly forced by the atmosphere, which shows significant seasonal and inter-annual variability. So any harmonic S1 solution will be the reflect of the mean of S1 tide over a given time period, and would need to be completed with a consistent residual S1 DAC correction to account for its intrinsic variability. However this intrinsic variability is rather weak, and the accuracy of the S1 DAC solution is still limited by the temporal resolution of the atmospheric forcing used today. So at present, the operational processing of GDRs data is based on a DAC corrected from the mean S1 and S2 atmospheric components, and the S1 S2 variability (both atmospheric and gravitational forced) is then removed by the S1 S2 tidal model. So when the atmospheric forcing will allow sufficient accuracy for the S1 DAC processing, the FES group would be in favor of leaving S1 correction to be accounted for in the high frequency storm surge correction (DAC) instead in the tidal correction.

About pole tides, they are not targeted in the FES tidal atlases (which are mostly focused on higher frequencies), and are available from others sources (such as Wahr, 1985 https://doi.org/0.1029/JB090iB11p09363).

6. Section 3.2: Removing the entire Sa and SSa constituents, or just the non-tidal contribution in Sa and SSa? Please clarify. Also further elaborate on GLORYS-v1.

The correction with GLORYS-v1 SSH (Mercator-Ocean re-analysis) at annual and semi-annual periods aims to remove as much as feasible non-tidal contributions in sea level anomalies that could pollute K1 extraction from altimetry observations.

GLORYS produces and distributes global ocean reanalyses at eddy-permitting (1/4°) resolution that aim to describe the mean and time-varying state of the ocean circulation, including a part of the mesoscale eddy field, over recent past decades with a focus on the period since when satellite altimetry measurements of sea level provide reliable information on ocean eddies (i.e. from 1993 to present). The numerical model used is the NEMO OGCM in the ORCA025 configuration developed within DRAKKAR consortium (global with sea-ice, 1/4° Mercator grid). The model surface boundary conditions are derived from atmospheric ECMWF reanalyses. Assimilated observations are in-situ T&S profiles, satellite SST and along track sea-level anomalies obtained from satellite altimetry. The data assimilation method is based on a reduced order Kalman filter using the SEEK formulation. GLORYS-v1 re-analysis is obtained by assimilating data (including altimetry SLA) into NEMO.

7. Section 3.4: Explains the ice coverage problem in high latitudes. It seems that this section only applies to unbalanced observations caused by ice coverage and not other phenomena affecting the tidal estimates e.g. ocean circulation. This is explained in section 3.2 and in section 4.3 where fewer crossover data were added for the data assimilation process because of the large contamination of mesoscale processes. Please consider adding these effects (or others), affecting the separability and possibly state their impact on the extended frequencies in the FES2014b model version.

The presence of seasonal ice induces a possibly large loss of data in the polar oceans. In this case, the usual Rayleigh criterion, based on observation duration, will fail to accurately estimate frequency separability. The principle of our separation diagnostic method is more direct. Ideally, i.e. in case of quasi-infinite time series, the harmonic matrix will be quasi-diagonal. The shorter the time series, the larger the cross-terms/diagonal-terms ratio in the matrix, which reflects the loss in separation efficiency. In the case of a regularly sampled, continuous time series (no data missing), the usual Rayleigh criterion (at least 1 period difference between two different constituents over the time series duration) is equivalent to a maximum ratio of ~0.15 in any row of the harmonic matrix. In the case where 2 constituents show a ratio larger than 0.15, we check whether admittance can be used to infer the one with the lowest astronomical potential or not. If not the case or if at least one is a non-astronomical constituent, we drop it.

8. Figure 2-3: The details at some regions lost at the coastal regions for both the upper panel and lower panel. Consider an alternate symbology for the presentation of vector differences.

**It will be done in the revised manuscript.**

9. Figure 11: Add source for tide gauge database and each of the respective data count.
**It will be done in the revised manuscript.**

10. Related to above on pelagic tide gauge data validation of tide models. Can you comment or would you have cases where the tide gauge data used to initialize/constrain FES model are used and not used to evaluate the FES model?
Unfortunately, the number of quality tide gauge data is not large enough to allow for a distinct data assimilation and well-balanced validation datasets. In consequence, we made the choice to use the same quality-checked dataset for both use. So there is a bias in terms of accuracy when comparing our final solution with the assimilated dataset. However, because of the ensemble/data representers approach, the assimilation solution will not easily fit the tide gauge data if not consistent with model error covariance or altimetry data. Internally, we also make consistency checks by assimilating altimetry data only, then comparing with tide gauge data and reverse.

11. Figure 16: A slight rise in variance is visible around Lapdev and Kara Seas for Altika mission altimeter data, when using FES2014 and FES2012 models as tidal corrections. Speculate on the reasons for such a rise in variance.
The Altika mission suffers from shorter (in duration) and fewer (in repetitivity) exact repeat observations compared to Jason time series. In consequence, the variance reduction diagnostic is therefore made on a less significant statistical basis, and the overall variance reduction map shows many local, "noisy" outliers (compared to the surrounding general tendency). In addition, seasonal ice in the Arctic Sea is furthermore diminishing the number of available valid observations, hence potentially increasing the uncertainty on the variance reduction estimates. Some independent validation was performed to compare FES2012 and FES2014 (by R. Ray in particular) and showed the clear improvement in FES2014. So we tend to consider that the rise shown in the Altika comparison might not reflect a lower FES2014 accuracy in this region, but simply a statistical issue. Unfortunately, the lack of reliable tide gauge dataset for the Kara and Laptev Sea makes any stronger conclusions quite difficult to draw.

---

## Author Comment (AC2) · 8 Jan 2021

(for commodity reasons, we have reproduced the reviewer text in black, answers are in light blue, further action to the revised manuscript in bold blue)

This paper provides a thorough description and evaluation of the FES2014 tide model, in several variations. The rationale and design of the modeling and assimilation strategy for developing FES versions is explained very clearly, and the detailed description of their ensemble technique will be a welcome addiiton to the literature. This information will help ocean modelers understand the potential strengths and weaknesses of a model like this, and it will also guide others hoping to build on their work in the future.

The detailed evaluations of the model and comparisons of its various versions and other tide models will be useful to potential end-users of the tidal atlas. I believe this article will be of interest to readers of Ocean Science Discussions, and I recommend that it be published after some minor revisions, detailed below.
There are two main areas where the article should be revised to make it easier for a reader to understand and use the provided information:

1) In some places the article lacks clarity regarding the rationale for the comparisons using FES2014a, FES2014b, and FES2014c. Examples are noted in the detailed listing below. The manuscript would be a little easier to read if it clearly stated ahead of time which versions of the atlas are being used and why.
**As stated in the answers to review's comments, we will follow your recommendations in the revised paper.**

2) The article does not discuss at all the tidal prediction software which is provided with FES2014. I don't think that the article needs to discuss or provide this in detail, but it needs to be clearer about the nature of the tidal predictions used for computing variance reduction statistics. Do these predictions include all the tidal frequencies mentioned in the manuscript, including the ones which are forced with the ERA-interim in the time-stepped model? Do the predictions include smaller tides computed using inference? If so, do the inference formulas rely exclusively on those tides which were computed with assimilation, or do they use some of the other frequencies? If the comparisons of different models use different constituents, then the manuscript should state this clearly in the tables or figures where the comparisons are shown. Another aspect of the tide predictions needs to be discussed since the atlas is intended for correcting altimetry: How should FES2014 be used when a separate model is used to provide a DAC?
Yes, the tidal predictions include all tidal frequencies mentioned in the manuscript (either from the FES2014 tidal atlas, or from inference for some smaller tides). The tidal prediction software is available on a bitbucket deposit : https://bitbucket.org/cnes_aviso/fes/src/master/. The inference formulae and the list of the tidal constituents that can be computed by inference are listed in the code.
When comparing FES2014 performances to other models in terms of variance reduction, the FES2014 prediction uses all FES2014 waves available within the FES2014 tidal atlas, and thus it includes more tidal frequencies than other models (GOT4.10 and TPXO respectively). This choice was done in order to take into account both the modeling and the omission error of the models. Concerning GOT4.10 and TPXO predictions all tidal waves available in the respective atlases are taken into account and the same inference algorithms are used.
Yes FES2014 tidal correction can be used even if a new model is used for the DAC correction, at the condition that the new DAC solution does not include the S1 and S2 frequencies which are already contained in the tidal model. When a separate model is used to provide a DAC, S1 should not be used in the prediction. See also our reply to reviewer 1's comment #5.

**Detailed comments:**

Good introduction which lays out the features of the tidal data assimilation: limitations of L2-norm, hence importance of the prior model and outlier rejection, and relative data sparsity depends on dynamics/wavelength.
Good explanation of FES2014a, b, and c releases.

p5,l7-9: "Initially, ... integrated in ..." I don't understand this sentence.
The T-UGOm model has been developed initially has a classical time-stepping code, and the frequency-domain solver (inspired from the CEFMO frequency-domain code used up to FES2004 atlas) is a later addition. As in

many other models, the Flather's method (also called Riemann's invariants method) is often used in regional or coastal configurations to mitigate the prescribed OBCs through a relationship mixing elevation and velocities. However, velocities extracted from a global atlas may not be fully consistent with the regional configuration to run, mostly because of mesh resolution, bathymetry and friction differences, while tidal elevation are less sensitive to those. A frequency-domain simulation, forced at its open boundaries with tidal elevation only, will produce a regional solution with properly "downscaled" velocities at its open limits, that can be re-used for further time-stepping simulations.

p5,l33: Does the discrete system satisfy any conservation laws?
Yes, but in the finite element variational sense (i.e. integral convolution with discretization interpolation functions over the domain). It is a bit tricky compared to usual finite volume (or C-grid) local conservation, but mathematically rigorous.

p5,l35: Thus, there is no lateral eddy viscosity in the model?
There is. We mostly use Smagorinski-tuned Laplacian type of eddy viscosity, but it usually act as a numerical scheme stabilizer, far from what would require a truly physical term. We make use of wave equation formulation to solve for the shallow-water equations, which demands a "diagonalization" of the momentum equations left-hand side. In time-stepping mode, non-diagonal terms (such as viscosity Laplacian, needed for stability reasons) are moved to the right-hand side (at central time in the leapfrog integration for example). It is not directly feasible with the frequency-domain solver, which actually imposes some additional (as for the non-linear terms). As the frequency-domain solver has no stability issue, and as eddy viscosity has a rather small contribution to tidal dynamics, we usually do not perform this step.

p7,l8-9: "even at regional extents, as earlier ... compensation bias." I don't understand this part of the sentence.
The model calibration step (based on semi-empirical trial simulations) aims to set "optimal parameters" (bottom roughness, IWD, etc…) to minimize tidal solution misfits with observations. This calibration helps to define more realistic settings, but also will include some error compensation (such as IWD value not only linked with ocean stratification characteristics, but partly compensating for some local bathymetry error at internal tides generation sites). So it can happen that improving the bathymetry in a limited region of the model will increase models errors, as parameters set for earlier bathymetry error compensation are then irrelevant, and calibration step must be carried out again. Same for change in coastal resolution, LSA atlas, etc… In consequence, we need to keep our model reasonably "light" to allow for a large number of tidal experiments.

p9,l8-9: Are you making a distinction between the tidal loading (the deformation of the earth surface caused by the tidal changes in ocean bottom pressure) and the solid earth tide here (the deformation of the earth surface caused by the gravitational perturbations of the sun and moon)?
Definitely yes, the latter being of course needed for altimetry data corrections, but easily computed from analytical formulas. We concentrate on deformation due to ocean mass re-distribution by the tides.

p9,l30: Thanks for describing this in detail.
The significant, non-tidal ocean sea surface variability around semi-annual and annual frequency will contaminate K1 harmonic analysis because its aliased frequency in TP/Janson (~6 months) and Envisat (~1 year) observations. We use a Glorys-derived SSH harmonic analysis at semi-annual and annual frequency to hopefully remove a part of the non-tidal contamination. The efficiency of the non-tidal ocean signal contamination has been assessed at TP/Jason cross-overs, where the K1 harmonic constants misfits between ascending track and descending track analysis are diminished.

p9,l36: I would have expected that all the harmonic analyses were based on the longest records available, so this section on S2 can probably be deleted.
**We might consider following your recommendations in the revised paper.**

p10,l19: Please explain this procedure more precisely. Is this equivalent to computing the standard error matrix for the least squares solution, and then recomputing estimates for fewer frequencies when the error correlation was too large? Exactly what criterion was used?
The principle of our separation diagnostic method is more direct. Ideally, i.e. in case of quasi-infinite time series, the harmonic matrix will be quasi-diagonal. The shorter the time series, the larger the crossterms/diagonal-terms ratio in the matrix, which reflects the loss in separation efficiency. In the case of a regularly sampled, continuous time series (no data missing), the usual Rayleigh criterion (at least 1 period difference between two different constituents over the time series duration) is equivalent to a maximum ratio of ~0.15 in any row of the harmonic matrix. In the case where 2 constituents show a ratio larger than 0.15, we check whether admittance can be used to infer the one with the lowest astronomical potential or not. If not the case or if at least one is a non-astronomical constituent, we drop it.

p10,l33: This is a little confusing. Are you saying that there are a total of $N_{tot}=432$ ensemble members, and these are computed from sets of independently perturbed ensemble components? Do you have, $N_{tot}=N_{bottom\ drag} * N_{internal\ tide} * N_{bathymetry} * N_{LSA}$ where $N_{LSA} = 2$, corresponding to the FES99 and FES2012 load tides?
(I see, yes. Made this table while reading later:)
$N_{bottom\ drag} = 8 * 13 = 104$
$N_{internal\ tide} = 10 * 7 = 70$
$N_{bathymetry} = 2 * 18 + 6 = 42$
The total number of ensemble members is composed as follows:
$N_{tot} = (N_{bottom\ drag} + N_{internal\ tide} + N_{bathymetry} )* N_{LSA}$
With
$N_{bottom\ drag} = 8 * 13 = 104$
$N_{internal\ tide} = 10 * 7 = 70$
$N_{bathymetry} = 2 * 18 + 6 = 42$
$N_{LSA} = 2$ (FES99 and FES2012 load tides)
**We will modified the text to clarify this section.**

p13,l6: "SpEnOI code is solving the assimilation in the data space" – Are you aware that you could reduce the dimensionality to 432 by solving in the space spanned by the ensemble members, without reducing the quantity of data at all?
We have not investigated this point much, despite it perhaps rings a bell about solution space reduction (G. Egbert et al. publication). As representers are to some extent covariance compute from members, hence not a linear operation, it seems not trivial to me. It would be interesting to discuss this matter with the reviewer#2. Actually, we kept the data representers approach as an heritage from the previous variational data assimilation code (CADOR, used in FES2004), because it was a quick solution (no much change in the software), and second because we wanted to be able to get back to the variational formulation if needed. The main obstacle is not that much the solution of the linear system itself, but the large volume of members vectors to be processed.

p13,l7-22: Can you reduce this text and just state what data you used for assimilation versus validation? I don't understand what data were used.
**We will follow your recommendations in the revised paper**

p13,l30: Did you use an estimate of the data error to weight the datasets that were assimilated, or did you simply assimilate fewer data in the noisier regions?
Yes, an estimate of the data error was used for the selection of all the assimilated observations. For the altimetry data, this data error is based on the least-squares error estimate (from the harmonic analysis). Error thresholds used to select the altimeter data are gathered in table 4 of the manuscript.
For the tide gauge observations, the error estimates were fixed empirically to 3 mm for the deep ocean stations and to 1 cm for the shelf and coastal stations. The idea was to limit the constraint on the model at the tide gauge stations on the shelf and close to the coast in order to avoid drawing the solution to fit some very local tide features observed by the coastal stations that may be inconsistent with the larger scale tidal patterns that can be accurately solved at the resolution of the model.

p14,l23: Could you please make clear whether "FES2014" refers exclusively to FES2014c in this section, or if you are describing aspects of the FES2014a and b solutions, too.
**Clarifications will be made in the revised manuscript.**

p14,l29: Earlier you stated that S1 belongs to the DAC solution. Please clarify.

Some users prefer to deal with a mean S1 tide correction rather than performing a DAC correction. So we provide a S1 harmonic atlas to leave the choice to the community to deal with S1 either as a tidal correction or as a DAC correction. I will clarify this point in the revised paper

p14,l40: I am confused about the use of ERA-INTERIM atmosheric forcing. Wouldn't you want to compute purely gravitationally-forced versions of these, since the atmospherically-forced component would be provided by a separate DAC model? Since the goal of the paper mentions developing the tidal atlas specifically for dealiasing altimetry data, you should be clearer about how this should be done in practice using FES2014c, since there is the potential to unintentionally duplicate corrections from the tidal atlas and the DAC correction as it is usually applied.
As mentioned earlier, we would prefer to leave the atmospherically-forced component in the DAC. Unfortunately, when performing data assimilation, astronomical and atmospheric contributions are definitely mixed in assimilation solution because it is mixed in the data. To avoid duplicated corrections, we consistently filter in the DAC corrections the contributions of (data assimilated) tidal constituents having a significant atmospherically-forced component (S1 and S2).
One could imagine removing the atmospherically-forced component from data using the DAC, but first it is feasible only for data of which original time records are available, and second it would add DAC modeling errors to observations. Finally, purely astronomical tides atlases would be pretty tricky to use by non-altimetric data users.

p15,l30: Could you please remind us about why you are validating FES2014b instead of FES2014c here?
FES2014c is based on FES2014b to which a number of long period tides, computed from mass-conservative equilibrium method, have been added (those long period tides, if not externally provided, are already taken into account in a similar way by the tidal prediction algorithm for the FES2014b prediction). Those long period tides are not considered in this validation section presented here and therefore FES2014b and FES2014c can be equivalently used.
**We agree it can be somehow confusing, and we should clarify this in the revised manuscript.**

p15,l36: Does "TPXO9" refer to the "TPXO9v2" mentioned on page 14? And, does "TPXO9-atlas" also refer to "TPXO9v2"?
Yes, "TPXO9" refers to "TPXO9v2". We have homogenized the names in the revised version of the paper. Following the naming convention used by OSU, "TPXO9-atlas" is the 1/30-degree solution of TPXO9v2 that we use for the comparison.
**We will rephrase the sentence in p15,l37 to make it clearer**.

p16: I am unclear on why FES2014a and FES2014b and FES2012 are used in some of these comparisons. I understand if this is simply related to the reporting of comparisons originally conducted for various purposes, but perhaps you could mention the reasons at the beginning of this section.
**We agree it can be somehow confusing, and we should clarify this in the revised manuscript.**

p17,l31: Can you cite a source for this statement? This certainly depends on depth of the station, stratification, etc.
I will add a source, however it is a common knowledge for people observing or modelling 3D baroclinic tides. Of course IT currents have a complex 3D structure, highly variable in the deep ocean, still they commonly reach much higher values than the barotropic one in the abyssal plain.

p18,l28: "dynamical quasi-coherence of the covariances" – I don't know what this means. Please explain or omit.
As representers (model error covariance) are based on dynamical ensemble statistics, they partially have some true dynamical properties, i.e. the assimilation solution is closer to a possible, particular dynamical solution than if using Gaussian functions as in optimal interpolation instead of representers.
**It will be rephrased this in the revised manuscript.**

p18,l30: Can you say anything about the results of Fig 21? If they don't show anything new, then perhaps this section can be omitted. Alternately, maybe they basically confirm but slightly alter previous estimates. Or maybe there is something that can be gleaned from comparisons of recent calculations by DeLavergne et al.

Since you previously stated that the currents are not reliable in regions of steep topographic slope(line 13), why would the energy flux divergence or the parameterized baroclinic wave drag be reliable?
Figure 21 is complementary to other works as mentioned in the comment and aims to advertise that barotropic tides energy budgets are available for the FES2014b constituents, including energy fluxes, bottom friction rate of work, etc... We believe it is an interesting point as first, energy budget examination is a very valuable way to understand tidal dynamics and validate tidal solutions, and second, internal tide generation budget remains a very open subject.
About the reliability of currents, the issue mentioned in our paper occurs only in a few places where local resolution was accidentally not properly adjusted to topography slopes constraint (which requests resolution to be proportional to H/grad(H)), and do not minder the overall accuracy of the energy budget estimates.

p19,l26: "After proper, competitive evaluation procedures" – Without explanation, the reader is not going to know what you are referring to. Could you either omit this phrase, provide a citation, or explain this topic further?

**English usage/typos:**
Language and typos have been corrected in preparation of the revised manuscript.

p5,l5: omit "about a"
p5,l42: "it is known to allow for" –> "that it allows"
p6,l7: "multi-levels" –> "multi-level"
p6,l10: I am more accustomed to seeing "CFL" rather than "CLF" for this abbreviation.
p6,l16: omit "and"
p6,l27: "Go" –> "GB" ?
p7,l18: "SAL" –> "LSA"; and fix "atlas atlases"
p7,l42: Why repeating S1 in the parens?
p7,l38: Can you reorganize this paragraph for clarity? Something like:
"The hydrodynamic solution for the S2 tide differs from the other tides in that the atmospheric forcing is explicitly included. ... The S1 tide originates mostly from the atmospheric forcing, and it was therefore not computed in the hydrodynamic tidal solution. ..."
p8,l16: omit comma at the end
p9,l14: please fix "Then,18); this"
p9,l20: I thought Parcel's theorem was the equality of the sum of squared Fourier components with the variance. Please reword or explain . p9,23: "guaranty" –> "guarantee"
p10,l11-12: omit "to ease the harmonic system solving"
p11,l8: ", 1982" –> "(1982)"
p11,l14: "abuse" –> "abuse of nomenclature"?
p11,l18: "is run" –> "are run"
p11,l21: "tide drag" –> "internal tide drag"?
p12,l6: "sloppy" –> "sloped"
p12,l9: "efficiency is strongly dependent on" –> "depends on"
p12,l13: "(75)" ?
p12,l19: commas after "GEBCO" and "release"
p14,l6: "repartition" –> "distribution"
p14,l16: "it small" –> "its small"
p14,l17: "ration" –> "ratio"
p16,l9: "if" –> "while" or omit "if"
p16,l14: "none data" –> "none of these data" or "no data"
p16,l11: Presumably, you used all the tidal harmonics available from each model in these comparisons, since you mentioned errors of omission previously. You should remind the reader of this here, or later in the Discussion.
p16,l20: Can you redo this comparison of FES2014a with FES2014b, instead?
p17,l40: repetitive of line 34; please consider revising this entire section for brevity.
p18,l15: omit "Somehow,"
p18,l37: capitalize "Love"?
p18,l39: omit "then derivation"?
p19,l23: "others" –> "other"

p19,l35: "coastal details grid flexibility" –> "detailed coastal grid"
p19,l38: omit "generation"
p20,l5: "atlases" –> "atlas"; omit "locally strongly" on l6
p20,l9: "eased" –> "open"
p20,l11: "as well in terms of" –> "for both"

**Figures and Tables:**
**Most of figures are being re-processed to comply with reviewers comments and improve their graphical quality.**

p24-25,F2 and F3: The caption mentions panels a, b, and c, but only two panels are shown. Pleas label the panels a, b, and c, and include the Darwin name boldly somewhere on each plot.
**This will be fixed in the revised manuscript**

F4: capitalize "Maps" in caption. Other captions also need this correction.
**This will be fixed in the revised manuscript**

F8: The figure shows that the bathymetry of Weddell sea was perturbed, but I thought the text mentions that, after initial experimentation, it was not.
Actually, you're right, and the reason is that ice-shelf seas needs a specific treatment to infer free water column heights from bottom topography and immersed ice thickness, and the number of available databases to perform this operation was found to be too limited to allow for a proper perturbation procedure.

F12: Please clarify how "deep", "shelf", and "coastal" are defined. Is it a depth criterion, or simply whatever the data source happened to label it as.
**This will be fixed in the revised manuscript**

F13: I cannot distinguish the dark colors at the ends of the colorscale. Maybe represent this data with different size disks, like your previous comparisons in Fig 2? It is difficult to understand the information on this map.
**This will be fixed in the revised manuscript**

F14: Consider coloring the land and sea differently (white vs light gray). Does blue mean that FES reduces more variance?
**Yes, and we will clarify it the revised manuscript.**

F15: Is this comparing tide predictions made with the same constituents from each model? If not, you should explain to the reader that FES is providing more constituents, and/or cite Stammer et al, or Zaron and Elipot (2020, https://doi.org/10.1175/JPO-D-20-0089.1) where the comparisons are restricted to common constituents.
No, as mentioned above, the number of constituents differs for each model as the tidal prediction is based on each model tidal atlas, but same tidal prediction software is used. Yes FES2014 atlas includes more waves particularly for non linear tides, which is of great interest in the shallow water regions and makes this model better suited to correct altimeter data in those regions. Notice that the comparison/validation of each individual tidal component is described in the "validation in the frequency domain" section of the manuscript.
**This will be clarified in the revised manuscript**

F17: Once again, it is important to be sure the reader is aware whether the same constituents are being used, especially since FES2014 is using a number of frequencies with substantial atmospheric forcing. Also – GOT is not really intended for use close to the coastline. Did you extrapolate the values landward, or does this comparison only use locations where GOT grid cells happen to overlap the coastline?
The models used are extrapolated on the coasts in order not to miss any altimeter measurement near the coastlines; this processing is the one performed operationally to provide the tidal model solutions to be used in the altimeter GDRs.
**This will be clarified in the revised manuscript**

F18: I think this figure can be omitted.
We agree it finally does not add much value to the paper. **This will be fixed this in the revised manuscript**

F20: Can you use a different scaling to make the current ellipses more visible? Consider omitting this figure if you can't revise it. Or, maybe show a few representative comparisons enlarged, instead.
**This will be fixed in the revised manuscript**

F23: The caption is hard to understand. Could you describe this with something like:
"Map of the SSH variance difference a crossovers using the FES2014a tidal loading versus the GOT4v8ac tidal loading. ..."
**This will be fixed in the revised manuscript**

p39,T1: fragment at end of caption, "3 Tidal harmonic ..."
**This will be fixed in the revised manuscript**
p39,T2: Please fix "#DIV/0!". Also the different vertical alignments and number significant figures displayed for values in the table is disconcerting. Also, unlike the other numeric values in the manuscript (cf., Table 3), the "," is used instead of the "." to indicate the decimal point; please revise for consistency.
**This will be fixed in the revised manuscript**

p40,T4: What criteria are used to distinguish "shelves", "Open ocean", and "Arctic Ocean"? Is the Max error criterion applied to the least-squares error estimate (from the harmonic analysis), or do you estimate the error in some other manner?
The max error is based on the least-squares error estimate. Open and shelf ocean are distinguished following a depth=500m criterion. Arctic Ocean is delimited using a polygon.

---

## Author Comment (AC5) · 8 Jan 2021

Authors answers to RC3 comments/questions/recommandations

(for commodity reasons, we have reproduced the reviewer text in black, answers are in light blue, further action to the revised manuscript in bold blue)

This is a nice overview of the evolution and present status of the FES series of global tidal models. Brief descriptions of the major components in the model (e.g., finite element/volume approaches, assimilation techniques and data, self-attraction and loading, bottom friction and internal drag coefficients, barotropic to baroclinic conversion) are given along with numerous figures illustrating improvements from earlier versions, and comparisons with other global models. Though I recommend publication, I feel the paper would be stronger if the following, generally minor, issues could be addressed.

General Comments:
1. Though the manuscript title specifies "atlas design and performance" (I suggest making "performances" singular), I would like to see more information on the atlas itself. The manuscript states (page 4, line 22) that the actual FES tidal constituent harmonics can be found on an "AVISO+" website, but I think the specific URL should be given (here or in an Appendix) (https://www.aviso.altimetry.fr/en/data/products/auxiliaryproducts/global-tide-fes.html ?) along with a short summary of what (not only data but software?) is available.
**Short description will be added in the revised manuscript**

2. Though the numerous global maps have lots of information, not all of the details are accessible. Specifically, as a coastal modeler I would not only like to see the M2 and K1 amplitudes in the specific region where I work but also the location of assimilated tide gauge, cross-over, and along-track data. (This information will assist in determining whether: i) FES could be used to provide boundary conditions for a regional model, and ii) the FES de-tided altimetry is likely to have adequately removed smaller scale tidal variations.) For many of the global figures in this manuscript, even when I zoomed in, it was not possible to see this information clearly. I don't know how this manuscript will be published electronically but urge the authors (and the journal's scientific and technical editors) to provide sufficiently, highly-resolved figures so that when zoomed in, interested readers can pick out the details they want. On Figures 2, 3, 9, 10, and 13 for example, this may require changing the dot scaling so that as one zooms-in, the dots become smaller so their precise locations become evident. In the case of Figs 2 and 3, smaller dots would no longer obscure the background image but perhaps also allow more amplitude contours to be displayed.
We are now re-generating the figures to improve their graphical quality.
**Your recommendation will be taken into account in the revised manuscript as much as feasible.**

3. Along the lines of comment 2, I also would like to see the actual FES model grid (page 6, lines 14-22), especially its coverage and resolution in shelf and coastal regions. Figure 1 partially addresses the resolution issue but even when zoomed in, it doesn't provide the detail I would like to see. Perhaps an image of this grid (with zoom in capability) exists somewhere on an AVISO+ or LEGOS/CRNS/CNES website but I couldn't find it in a quick search. If it does exist, then that location should be given. If it doesn't, then I strongly recommend that be done, as again, the information would be useful for regional and coastal modelers
We are now re-generating the figures to improve their graphical quality.
**Your recommendation will be taken into account in the revised manuscript as much as feasible.**

Specific Comments:
1. Page 3, line38-40: What other norms can be used? A reference should be given.
There are many other norms than can be used (L1, L-infinite, etc…). The best references I can think about are as following:
Bennett, A. (1992). Inverse Methods in Physical Oceanography (Cambridge Monographs on Mechanics). Cambridge: Cambridge University Press. doi:10.1017/CBO9780511600807
Tarantola, Albert. (2005). Inverse Problem Theory and Methods for Model Parameter Estimation. 10.1137/1.9780898717921.

2. Page 5, line 5: remove "about"
**Fixed in the revised manuscript**

3. Page 5, line 33: Has NCP1 been defined?
Definition added  in the revised manuscript

4. Page 6, line 5-8: Give a reference for this statement
To my knowledge, D. Leroux is one of the head investigator of that type of numerical studies, see for instance:
Le Roux, Daniel & Rostand, Virgile & Pouliot, Benoit. (2007). Analysis of Numerically Induced Oscillations in 2D
Finite-Element Shallow-Water Models Part I: Inertia-Gravity Waves. SIAM J. Scientific Computing. 29. 331-360.
10.1137/060650106.
Reference will be added  in the revised manuscript

5. Page 6, line 10: CFL vs CLF ?
Fixed in the revised manuscript

6. Page 6, lines 19-22: As above in general comments 2 and 3, it appears from zooming-in on Fig 1 that Juan de
Fuca and Queen Charlotte Straits (south and north of Vancouver Island) are in the grid (resolution 5-10km?) but
the Strait of Georgia (which has a partial TPJ track) and its northern passages are not. Presumably you
compensate for this by assimilating crossover, tide gauge and along-track data. Zooming-in on Figs 9 and 11, I
can see what crossover and tide gauge data are used but not so for locations of the along-track data in Fig 10.
Furthermore, zooming-in on Fig 3 doesn't provide the detail to see, for example, if those data have caused the
FES solution to at least partially capture smaller scale features like the surface signature of K1 shelf waves off
the Vancouver Island coast.
The FES2014 mesh resolution is quite poor in all the western Canadian and Alaska coastal regions, and it results
in a loss of details/accuracy in all this area. For computation cost limitation reasons, higher coastal resolution
was deployed only where accurate bathymetry has been found by the project group, and this was not the case
at the time of FES2014 production. Later on, and thanks to some discussions with our Canadian colleagues, this
issue has been identified as quite damaging, but this happened too late to be corrected in FES2014 atlas. It can
be mentioned that a special care will be given to the region in the next FES atlas release.

7. Page 6, line 27: Gb vs Go ?
Fixed in the revised manuscript

8. Page 7, line 24: It's not clear if these coefficients are spatially constant over the entire globe or they differ
within say, the polygons in Figs 7 and 8.
Most of T-UGOM model parameters can be tuned locally using various methods (pre-defined regions, polygons
inclusion, node or element vector, etc…). In the FES2014 stlas simulations, internal wave drag coefficients are
tuned using a global ocean regional partition (distinguishing north, tropical, and south basins in the various
oceans plus Arctic sea and Mediterranean Sea), and bottom frictions coefficients are tuned by using polygons
(focused on large bottom friction dissipation areas). In both cases, a global default value is locally used at
locations not being part of the regions/polygons definition.

9. Captions to Figs 2 and 3: These are not clear. The a), b), and c) references suggest there should be 3 panels,
yet only 2 are presented. If a zoom-in capability were available for these figures, you could probably include the
shallow/coastal crossover locations too.
Legend was obsolete (originally 3 panels were displayed, but reduced to 2 panels during final writing).
Figures are being generated and should support zoom-in capabilities in the revised manuscript.

10. Section 3: I wonder about non-stationarity, especially in coastal regions where seasonal changes in river
discharge, winds, and ice cover may interact with the tides. Is this considered?
Despite the authors are fully aware of, it was not, mostly because the objective was not to produce a
seasonally varying tidal atlas. However it is quite a challenge for the future atlases in the context of SWOT,
which will provide data in estuaries and deltas, and in very high latitude regions. Also tides/storm surges
interactions need to be considered in altimetry high frequency corrections in shelf and coastal seas, but will
require to renew the present correction paradigm (separate tides and storm surges corrections) in the
operational data processing.

11. Page 9, line 20: Give a reference for Parceval's rule.
"Parseval equality", Encyclopedia of Mathematics, EMS Press, 2001 [1994]

**Reference will be added in the revised manuscript**

12. Page 10, line 14: In the Canadian Arctic, the additional drag from seasonal ice cover changes the constituent amplitudes and phases at some coastal locations. Can this non-stationarity be accounted for?
Basically, we have not considered this issue, despite we are aware of (and made several investigation in the past about it). It would lead to a season-dependent atlas, which was not suitable in the usual altimetry data correction framework.

13. Page 10, lines 19-22: I'm interested in more details on this. Is there a reference?
No reference yet, it was imagined and implemented during the FES2014 project , may be I should give more details on it.

14. Page 10, line 29: The text says the filtering wavelength is in km while the Fig 5 caption says it is in number of along-track points.
Wavelength haven been estimated in km, then converted into equidistant along-tracks points
**Comment will be changed to be coherent with the Fig5 caption.**

15. Page 11, lines 28-30: Why is that?
First the FES2014 hydrodynamic configuration has been adjusted (i.e. bottom friction and IWD) in simulations using the FES99 LSA, and part of it is an error compensation story. The most sensitive component in adjustment process is clearly M2. The main reason for that is that bottom friction is truly non-linear for M2, as it has the stronger currents and the dominate the current amplitude in the friction term, and the other constituents have a sort of quasi-linear friction in presence of M2 dominant velocities. So using a more modern, and potentially more accurate LSA, will usually profit to all constituents but M2, as it would require to re-process the adjustments steps to get back at least to a similar or improved accuracy.

16. Page 11, line 37: Presumably there was a dissipation cutoff to determine these polygons, as not all purple regions warranted one. What was it?
The definition of tuning polygons is a compromise to include the most significant sites for tidal dissipation, limited individual polygon area and the number of polygons (to limit too many additional members in our ensembles).

17. Page 12, line 23: The polygons on Fig 8 are not numbered.
**It will be fixed in the revised manuscript**

18. Page 12, line 32: Does it have to be the same everywhere? You probably need 10m in regions like the Bay of Fundy but it shouldn't have to be that large in many other coastal areas.
The depths found in most bathymetry databases in the 0-10m (and probably 20m) range is anything but reliable. In most places, the depths are linearly varying with distance from 0m at coastline to the 10m isobaths, which is not the usual morphology you will find in the true ocean. Such artificial very shallow water patches can have a damaging impact on bottom friction budget in coastal areas. The 10 m limitation is just a safety limit set to model bathymetry, and has been verified to be quite reasonable by experiments in the last 2 decades of tidal modelling. Having said that, in regions were bathymetry databases are highly accurate, it is preferable to keep the true depths (and do wetting/drying). But it represents only a tiny portion of the global ocean coastal regions.

19. Page 14, line 18: "used in" vs "kept from" ?

20. Page 15, line 36: It seems that TPXO9 is more accurate than FES2014b in the shelf and coastal regions. Can this be attributed to their generally higher (1/30 degree) spatial resolution?
To be fair with TPXO atlases, we have chosen to compare with the very last release, and not the release available at FES2014 production time (TPXO8), which shows less accuracy than FES2014. So probably the new TPX09 atlas has taken profit of longer time series for altimetry data, and possibly improved bathymetry for its prior simulations (or any others improvements in the regional hydrodynamics models configurations).

21. Page 17, line 6: Why is there a peak at about 14km from the coast in the left panel of Figure 17?

Difficult to tell with full certitude, the most likely explanation is that near-shore performance in FES2012 and FES2014 is limited by local bathymetry accuracy and coastal detail discretization, and ensemble/representers being less able to properly describe local errors statistics, so data assimilation improvements in FES2014 propagate only partially toward nearshore zones.

22. Page 17, line 28: Give a URL for this portal.
https://imos.org.au
**It will be added in the revised manuscript**

23. Page 17, line 36: Often phase lags are included in these ellipses by showing the current vector position at the time of maximum tidal potential at Greenwich. Also, an arrow is sometimes placed on the ellipse itself to denote the sense of rotation. Were these not done in Figs 19 and 20 because they would make the figures too complex?
We are now re-generating the figures to improve their graphical quality.
**Your recommendation will be taken into account in the revised manuscript as much as feasible.**

24. Page 17, line 42: You should change the scale to make the ellipses larger for K1. Perhaps a zoom-in capability (ellipse size changing as you zoom-in) is needed here in order to decipher which ellipses belong to which dot.
We are now re-generating the figures to improve their graphical quality.
**Your recommendation will be taken into account in the revised manuscript as much as feasible.**

25. Page 18, line 11: I can guess which station this is but maybe you should give the approximate lat/lon to help readers.
**It will be added in the revised manuscript**

26. Figure 21: A zoom-in capability that doesn't blur the color details would be useful here.
We are now re-generating the figures to improve their graphical quality.
**Your recommendation will be taken into account in the revised manuscript as much as feasible.**

27. Page 19, lines 5-6: Why does the lower panel of Fig 22 show wave-like (Gibbs?) patterns, for example radiating eastward and westward from the Canadian Pacific coast?
This is likely due to differences between Green's functions based computation of LSA (as in FES2014 atlas) and spherical harmonics based computation (GOT).

---

## Author Response (AR2)

**Authors answers to EC2 comments/questions/recommandations**

(for commodity reasons, we have reproduced the reviewer text in black, changes are in light blue, further action/comment to the revised manuscript in bold blue)

**Topic Editor Decision: Publish subject to minor revisions (review by editor)** (14 Feb 2021) by Philip Woodworth Comments to the Author: 14/2/2021

Comments on the revision of this paper.

This paper has been much improved following the comments of the reviewers, and the figures have also been improved. I simply read the paper again and have a remaining large number of small edits which should not take the authors long. There are also a couple of more important issues.

If the authors disagree with any of these comments then I suggest they mail me privately. Otherwise I assume they will be all attended to in a final revision which I will not have read in detail again.

All comments accepted and fixed in revised version. I just reported correction that you might check before I upload the revised paper.

General comment - please see lower down but I got confused why you made comparisons with FES2014a and not FES2014b which is the final product, isn't it? This needs explicitly explaining.

I added some words to explain FES2014a and FES2012 in some inter-comparison diags.

**5 Atlas assessment and validation**

The validation of the FES2014 tidal atlas is based on a frequency-domain (harmonic) validation of the ocean tide components plus a temporal validation of the total geocentric tide components (i.e. ocean tide plus loading tide). The FES2014b performance is compared to state-of-the-art global tidal models available at the time of the study, namely GOT4v8/GOT4v10, DTU10, TPXO9v2, EOT11A and FES2012 (please note that FES2014c and FES2014b have identical main long period, diurnal, semi-diurnal and sub-harmonics solutions, and the FES2014c long period extension is identical to the one implicitly made inside the prediction software, so the following validations will mention FES2014b only and will hold for FES2014c as well). The FES2012 and Fes2014a atlas have been included in some performance inter-comparison assessments to demonstrate the beneficial impact of the following evolutions: FES2014c prior hydrodynamic solution in the assimilated solutions, while FES2014a/b differences mostly illustrate the improvement coming from the significantly higher accuracy of the FES2014a prior hydrodynamic solution in the assimilated altimetry data processing (tidal loading correction)...

**I also added in figure captions:**

The accuracy improvement between the FES2012 and FES2014 prior solutions is a key ingredient in the accuracy improvement between the FES2012 and FES2014a assimilated solutions

Detailed comments:

pages 1-2. I ignored these as the title and abstract are slightly different from those on page 3 which I took to be the official versions.

**I messed up (duplicated) abstract in last revised version, sorry. Now fixed**

31 Lynch and Gray (1977), and continuously developed since, the approach has evolved from application to the deep [?] global ocean, now up to the inclusion of near-shore ..

Inspired from Lynch and Gray (1977), and continuously developed since, the approach has evolved from application to the global ocean, now up to the inclusion of near-shore and estuarine numerical applications, with wetting/drying and non-hydrostatic (surface wave dynamics) capabilities.

15 - you refer to the model grids here, but there is no reference in the text to the Supplementary Figure and its grid. It needs including either here or somewhere.

The targeted resolution for coastal areas is typically 10 kilometres or less in terms of triangle side-length (shown in Figure 1; the mesh details would not be visible on a printed global ocean figure, the authors have provided a zoomable supplementary pdf file available on Ocean Science website https://www.ocean-science.net)

**(not yet sure how to refer to the pdf, will check with Anna)**

38 what does 'non-free' mean here? Please reword

(actually, this is the only one model ingredient which depends upon a pre-existing ocean tides information in our hydrodynamic simulations)

I feel this paragraph is not complete. Or perhaps it needs a pointer to where the S1 and S2 issue is discussed lower down.

The numerous difficulties arising from the atmospheric pressure forcing at tidal frequencies (impacting tidal hydrodynamic solutions, de-aliasing corrections and data processing), so additional discussions on S1 and S2 constituent issues are given in the following sections.

18 Because this signal was strongest during the TOPEX-POSEIDON mission - this needs a reference also.

I think somewhere you have to make it clear what you mean by 'altimetry' i.e that it has a routine IB correction, unlike tide gauges

lines 18-25 - I found these lines very hard to understand. Please could you look at and reword?

In altimetry mission observations, the S2 tidal constituent is challenging as it is aliased on the infinite period and thus is not observable by the ERS/EnviSat sun-synchronous orbit as mentioned before. TP-Jason orbit is adequate to the observation of most of the main tidal constituents, however, because of its 58.74-day aliased period, the S2 tide sea surface signal is mixed with the residual Mean Sea Level (MSL) signal visible at the same frequency in the TP-Jason time series, which is linked to the inaccurate account of the  $\beta'$  angle in one or several standards used in MSL computation (Ablain et al. 2010; Zawadzki et al. 2016). Consequently S2 harmonic analysis will be contaminated by this GDR processing-dependent signal (with a possible feed-back through the tidal corrections in the GDRs, making this issue even more intricate). As it is stronger in TOPEX-POSEIDON mission GDRs (as reported in Zawadzki et al., 2016), several analyses have been performed using either the entire TOPEX-Jason time series or only the Jason-1/Jason-2 relatively recent records. But due to the much shorter duration of the latter, the estimation error is larger for the J1-J2 only analysis, and the assimilated solution proves finally to be more accurate (using TG data as sea truth) using the analysis from the entire altimeter series. Notice that thanks to its primarily approach based on an accurate hydrodynamic modelling, further moderately tuned by data assimilation (thus allowing a reduced weight of the data and data errors in the global FES solution), the FES2014 S2 solution is less affected by this residual GDR processing signal than empirical models, with in addition a beneficial effect on reducing the residual MSL error if used for tidal corrections in GDR processing (Zawadzki et al. 2016).

**p13, 6 - what does point-by-point clearing mean?**

Neither high-latitude data set manual editing nor entire data set rejection were an option, the former being a gigantic task and the latter an extremely damaging loss of data in already poorly documented regions.

**34 - 'error compensation story' sounds odd. Can you reword?**

first the FES2014 hydrodynamic configuration has been adjusted (i.e. bottom friction and internal wave drag due to barotropic to baroclinic energy conversion, denoted IWD) in simulations using the FES99 LSA, and includes clearly an error compensation contribution, i.e. configuration adjustments compensate for the FES99 LSA defects. Consequently, considering the high level of accuracy of the hydrodynamic solutions and thus the sensitivity to any minor changes, they are not fully appropriate for a simulation forced with another LSA atlas

34-35 sentence 'It might'. I don't understand this. I would drop it. (not dropped but I changed words)

However, the implementation, inside the prediction software, of the inference method to increase the prediction spectrum efficiently compensates for the impact of missing astronomical constituents in the GOT4v10 atlas, so most of the differences in the actual prediction spectrum will be limited to the differences in the availability of compound tide constituents.

Fig 19 - remove the stray script top right in the figure

why does the big ellipse near Adelaide have a gap? Looks like a plotting error

**Plots have been reprocessed and captions are updated**

Ellipses scales in m/s. Inside line indicate velocity direction at Greenwich transit time, ellipse rotation from inside line to arrow-terminated ellipse contour.

**Additional corrections (after iterating with P. Woodworth, see below). All comments and corrections were accepted and corrections were made in the revised pdf.**

Hi Florent – I have made some edits below in red (either use or don't use them if you agree or not). Things in [red brackets] means delete.

I didn't understand the (tidal loading correction) mentioned at the end of the first paragraph below.

All the other things were ok.

Thanks for doing all this again.

Phil

**5 Atlas assessment and validation**

The validation of the FES2014 tidal atlas is based on a frequency-domain (harmonic) validation of the ocean tide components plus a temporal validation of the total geocentric tide components (i.e. ocean tide plus loading tide). The FES2014b performance is compared to state-of-the-art global tidal models available at the time of the study, namely GOT4v8/GOT4v10, DTU10, TPXO9v2, EOT11A and FES2012 (please note that FES2014c and FES2014b have identical main long period, diurnal, semi-diurnal and sub-harmonics solutions, and the FES2014c long period extension is identical to the one implicitly made inside the prediction software, so the following validations will mention FES2014b only and will hold for FES2014c as well). The FES2012 and FES2014a atlases have been included in [some] performance inter-comparison assessments to demonstrate the beneficial impact of the following evolutions: FES2014a prior hydrodynamic solution in the assimilated solutions, while FES2014a/b differences mostly illustrate the improvement coming from the significantly higher accuracy of the FES2014a prior hydrodynamic solution in the assimilated altimetry data processing (tidal loading correction)...

31 Lynch and Gray (1977), and continuously developed since, the approach has evolved from application to the deep [?] global ocean, now up to the inclusion of near-shore ..

Inspired by Lynch and Gray (1977), and continuously developed since, the approach has evolved from application to the global ocean, now up to the inclusion of near-shore and estuarine numerical applications, with wetting/drying and non-hydrostatic (surface wave dynamics) capabilities.

15 - you refer to the model grids here, but there is no reference in the text to the Supplementary Figure and its grid. It needs including either here or somewhere.

The targeted resolution for coastal areas is typically 10 kilometres or less in terms of triangle side-length (shown in Figure 1; the mesh details will not be visible on a printed global ocean figure, the authors have provided a zoomable supplementary pdf file available on the Ocean Science website <a href="https://www.ocean-science.net">https://www.ocean-science.net</a>)

**(not yet sure how to refer to the pdf, will check with Anna)**

38 what does 'non-free' mean here? Please reword

(actually, this is the only [one] model ingredient which depends upon [a] pre-existing ocean tide[s] information in our hydrodynamic simulations)

I feel this paragraph is not complete. Or perhaps it needs a pointer to where the S1 and S2 issue is discussed lower down.

There are numerous difficulties arising from the atmospheric pressure forcing at tidal frequencies (impacting tidal hydrodynamic solutions, de-aliasing corrections and data processing), so additional discussions of the S1 and S2 constituent issues are given in the following sections.

lines 18-25 - I found these lines very hard to understand. Please could you look at and reword?

In altimetry mission observations, the S2 tidal constituent is challenging as it is aliased [on the] to infinite period and thus is not observable by the ERS/EnviSat sun-synchronous orbit as mentioned before. The TP-Jason orbit is adequate for [to] the observation of most of the main tidal constituents. However, because of its 58.74-day aliased period, the S2 tide sea surface signal is mixed with the residual Mean Sea Level (MSL) signal visible at the same frequency in the TP-Jason time series, which in turn is linked to inaccuracy in [the inaccurate value account of] the  $\beta'$  angle in [one or several standards used in] MSL computations (Ablain et al. 2010; Zawadzki et al. 2016). Consequently, S2 harmonic analysis will be contaminated by this GDR processing-dependent signal (with a possible feed-back through the tidal corrections in the GDRs, making this issue even more complicated [intricate]). As this problem is larger for the [it is stronger in] TOPEX-POSEIDON mission GDRs (as reported in Zawadzki et al., 2016), several analyses have been performed using either the entire TOPEX-Jason time series or only the Jason-1/Jason-2 relatively recent records. But due to the much shorter duration of the latter, the estimation error is larger for the J1-J2 only analysis, and the assimilated solution proved[s] finally to be more accurate (using TG data as sea truth) using the analysis from the entire altimeter series. Notice that thanks to its primary emphasis [primarily approach based] on [an] accurate hydrodynamic modelling, further moderately tuned by data assimilation (thus allowing a reduced weight of the data and data errors in the global FES solution), the FES2014 S2 solution is less affected by this residual GDR processing signal than empirical models, with in addition a beneficial effect on reducing the residual MSL error if used for tidal corrections in GDR processing (Zawadzki et al., 2016).

**p13, 6 - what does point-by-point clearing mean?**

Neither high-latitude data set manual editing nor entire data set rejection were [an] options, the former being a gigantic task and the latter an extremely damaging loss of data in already poorly documented regions.

**34 - 'error compensation story' sounds odd. Can you reword?**

First, the FES2014 hydrodynamic configuration has been adjusted (i.e. bottom friction and internal wave drag due to barotropic to baroclinic energy conversion, denoted IWD) in simulations using the FES99 LSA, and including clearly an error compensation contribution, i.e. configuration adjustments compensate for the FES99 LSA defects. Consequently, considering the high level of accuracy of the hydrodynamic solutions and thus the sensitivity to any minor changes, they are not fully appropriate for a simulation forced with another LSA atlas

---

## Author Response (AR3)

(for commodity reasons, we have reproduced the reviewer text in black, changes are in light blue, further action/comment to the revised manuscript in bold blue)

P12, line 25 – in turn

P19, line 5 – atlas ◊ atlases

P29, line 29 – there is a stray javascript mention here

P30, line 24 – there is a stray Wikipedia reference here

**All comments accepted and fixed in revised version.**